# Evaluating the design space of diffusion-based generative models

**Yuqing Wang**
Simons Institute
University of California, Berkeley
yq.wang@berkeley.edu

**Ye He**
School of Mathematics
Georgia Institute of Technology
yhe367@gatech.edu

**Molei Tao**
School of Mathematics
Georgia Institute of Technology
mtao@gatech.edu

## Abstract

Most existing theoretical investigations of the accuracy of diffusion models, albeit significant, assume the score function has been approximated to a certain accuracy, and then use this a priori bound to control the error of generation. This article instead provides a first quantitative understanding of the whole generation process, i.e., both training and sampling. More precisely, it conducts a non-asymptotic convergence analysis of denoising score matching under gradient descent. In addition, a refined sampling error analysis for variance exploding models is also provided. The combination of these two results yields a full error analysis, which elucidates (again, but this time theoretically) how to design the training and sampling processes for effective generation. For instance, our theory implies a preference toward noise distribution and loss weighting in training that qualitatively agree with the ones used in Karras et al. [30]. It also provides perspectives on the choices of time and variance schedules in sampling: when the score is well trained, the design in Song et al. [46] is more preferable, but when it is less trained, the design in Karras et al. [30] becomes more preferable.

## 1 Introduction

Diffusion models became a very popular generative modeling approach in various domains, including computer vision [20, 7, 27, 28, 38, 51], natural language processing [6, 34, 37], various modeling tasks [15, 41, 55], and medical, biological, chemical and physical applications [3, 17, 43, 49, 23, 56] (see more surveys in [53, 11, 14]). Karras et al. [30] provided a unified empirical understanding of the derivations of model parameters, leading to new state-of-the-art performance. Karras et al. [31] further upgraded the model design by revamping the network architectures and replacing the weights of the network with an exponential moving average. As diffusion models gain wider usage, efforts to understand and enhance their generation capability become increasingly meaningful.

In fact, a rapidly increasing number of theoretical works already analyzed various aspects of diffusion models [32, 19, 52, 16, 12, 8, 18, 9, 13, 39, 44, 25]. Among them, a majority [32, 19, 52, 16, 12, 8, 18] focus on sampling/inference; more precisely, they assume the score error is within a certain accuracy threshold (i.e. the score function is well trained in some sense), and analyze the discrepancy between the distribution of the generated samples and the true one. Meanwhile, there are a handful of results [44, 25] that aim at understanding different facets of the training process. See more detailed discussions of existing theoretical works in Section 1.1.

38th Conference on Neural Information Processing Systems (NeurIPS 2024).

However, as indicated in Karras et al. [30], the performance of diffusion models also relies on the interaction between design components in both training and sampling, such as the noise distribution, weighting, time and variance schedules, etc. While focusing individually on either the training or generation process provides valuable insights, a holistic quantification of the actual generation capability can only be obtained when both processes are considered altogether. Therefore, motivated by obtaining *deeper theoretical understanding of how to maximize the performance of diffusion models*, this paper aims at establishing a full generation error analysis, combining both the optimization and sampling processes, to partially investigate the design space of diffusion models.

More precisely, we focus on the variance exploding setting [46], which is also the foundation of continuous forward dynamics in Karras et al. [30]. Our main contributions are summarized as follows:

- For denoising score matching objective, we establish the exponential convergence of its gradient descent training dynamics (Theorem 1). We develop a new method for proving a key lower bound of gradient under the semi-smoothness framework [1, 35, 57, 58].
- We extend the sampling error analysis in [8] to the variance exploding case (Theorem 2), under only the finite second moment assumption (Assumption 3) of the data distribution. Our result applies to various variance and time schedules, and implies a sharp almost linear complexity in terms of data dimension under optimal time schedule.
- We conduct a full error analysis of diffusion models, combining training and sampling (Theorem 3).
- We qualitatively derive the theory for choosing the noise distribution and weighting in the training objective, which coincides with Karras et al. [30] (Section 4.1). More precisely, our theory implies that the optimal rate is obtained when the total weighting exhibits a similar "bell-shaped" pattern used in Karras et al. [30].
- We develop a theory of choosing time and variance schedules based on both training and sampling (Section 4.2). Indeed, when the score error dominates, i.e., the neural network is less trained and not very close to the true score, polynomial schedule [30] ensures smaller error; when sampling error dominates, i.e., the score function is well approximated, exponential schedule [46] is preferred.

Conclusions and limitations are in Appendix A.

## 1.1 Related works

**Sampling.** There has been significant progress in quantifying the sampling error of the generation process of diffusion models, assuming the score function is already approximated within certain accuracy. Most existing works [e.g., 16, 12, 8] focused on the variance preserving

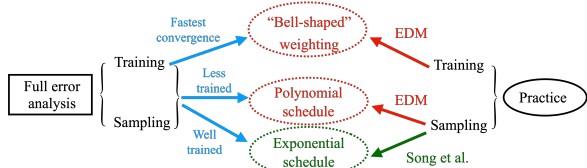

Figure 1: Structure of this paper.

(VP) SDEs, whose discretizations correspond to DDPM. For example, Benton et al. [8] is one of the latest results for the VPSDE-based diffusion models, and it only needs a very mild assumption: the data distribution has finite second moment. The iteration complexity is shown to be almost linear in the data dimension and polynomial in the inverse accuracy, under exponential time schedule. However, a limited amount of works [32, 24, 54] analyzed the variance exploding (VE) SDEs, whose discretizations correspond to Score Matching with Langevin dynamics (SMLD) [45, 46]. To our best knowledge, Yang et al. [54] obtained the best result so far for VE assuming the data distribution has bounded support: the iteration complexity is polynomial in the data dimension and the inverse accuracy, under the uniform time schedule. In contrast, our work only assumed that the data distribution has finite second moment, and by extending the stochastic localization approach in [8] to VESDE, we obtain an iteration complexity that is polynomial in the data dimension and the inverse accuracy, under more general time schedules as well. Note the improved complexity in terms of the inverse accuracy and the data dimension dependencies; in fact, under the exponential time schedule, our complexity is almost linear in the data dimension, which recovers the state-of-the-art result for VPSDE-based diffusion models.

**Training.** To our best knowledge, the only works that quantify the training process of the diffusion models are Shah et al. [44] and Han et al. [25]. Shah et al. [44] employed the DDPM formulation and considered data distributions as mixtures of two spherical Gaussians with various scales of separation, together with $K$ spherical Gaussians with a warm start. Then the score function can be analytically solved, and they modeled it in a teacher-student framework solved by gradient descent. They also provided the sample complexity bound under these specific settings. In contrast, our results work for general data distributions for which the true score is unknown, and training analysis is combined with

sampling analysis. Han et al. [25] considered the GD training of a two-layer ReLU neural network with the last layer fixed, and used the neural tangent kernel (NTK) approach to establish a first result on generalization error. They uniformly sampled the time points in the training objective, assumed that the Gram matrix of the kernel is away from 0 (implying a lower bound on the gradient), and lacked a detailed non-asymptotic characterization of the training process. In contrast, we use the deep ReLU network with $L$ layers trained by GD and prove instead of assuming that the gradient is lower bounded by the objective function. Moreover, we obtain a non-asymptotic bound for the optimization error, and our bound is valid for general time and variance schedules, which allows us to obtain a full error analysis.

**Convergence of neural networks training.** The convergence analysis of neural networks under gradient descent has been a longstanding challenge and has been developed into an extensive field. Here we will only focus on results mostly related to the techniques used in this paper. One line of them is approaches directly based on neural tangent kernel (NTK) [22, 21, 5, 47, 36]. However, existing works in this direction focus more on either scalar output, or vector output but with only one layer trained under two-layer networks, which is insufficient for diffusion models. Another line of research also considers overparameterized models in a regime analogous to NTK, though not necessarily explicitly resorting to kernels. Instead, it directly quantifies the lower bound of the gradient [1, 35, 2, 57, 58] and uses a semi-smoothness property to prove exponential convergence. Our results align with the latter line, but we develop a new method for proving the lower bound of the gradient and adopt assumptions that are closer to the setting of diffusion models. See more discussions in Section 3.1.

## 1.2 Notations

We denote $\|\cdot\|$ to be the $\ell^2$ norm for both vectors and matrices, and $\|\cdot\|_F$ to be the Frobenius norm. For the discrete time points, we use $t_i$ to denote the time point for forward dynamics and $t_i^\leftarrow$ for backward dynamics. For the order of terms, we follow the theoretical computer science convention to use $\mathcal{O}(\cdot), \Theta(\cdot), \Omega(\cdot)$. We also denote $f \lesssim g$ if $f \leq Cg$ for some universal constant $C$.

## 2 Basics of diffusion-based generative models

In this section, we will introduce the basic forward and backward dynamics of diffusion models, as well as the denoising score matching setting under which a model is trained.

### 2.1 Forward and backward processes

Consider a forward diffusion process that pushes an initial distribution $P_0$ to Gaussian
$$dX_t = -f_t X_t \, dt + \sqrt{2\sigma_t^2} \, dW_t, \tag{1}$$

where $dW_t$ is the Brownian motion, $X_t$ is a $d$-dim. random variable, and $X_t \sim P_t$. Under mild assumptions, the process can be reversed and the backward process is defined as follows

$$dY_t = \left(f_{T-t} Y_t + 2\sigma_{T-t}^2 \nabla \log p_{T-t}(Y_t)\right) dt + \sqrt{2\sigma_{T-t}^2} \, d\tilde{W}_t, \tag{2}$$

where $Y_0 \sim P_T$, and $p_t$ is the density of $P_t$. Then $Y_{T-t}$ and $X_t$ have the same distribution with density $p_t$ [4], which means the dynamics (2) will push (near) Gaussian distribution back to (nearly) the initial distribution $P_0$. To apply the backward dynamics for generative modeling, the main challenge lies in approximating the term $\nabla \log p_{T-t}(Y_t)$ which is called *score function*. It is common to use a neural network to approximate this score function and learn it via the forward dynamics (1); then, samples can be generated by simulating the backward dynamics (2).

### 2.2 The training of score function via denoising score matching

In order to learn the score function, a natural starting point is to consider the following score matching objective [e.g., 29]
$$\mathcal{L}_{\text{conti}}(\theta) = \frac{1}{2} \int_{t_0}^{T} w(t) \mathbb{E}_{X_t \sim P_t} \|S(\theta; t, X_t) - \nabla_x \log p_t(X_t)\|^2 \, dt \tag{3}$$

where $S(\theta; t, X_t)$ is a $\theta$-parametrized neural network, $w(t)$ is some *weighting function*, and the subscript means this is the continuous setup. Ideally one would like to minimize this objective function to obtain $\theta$; however, $p_t$ in general is unknown, and so is the true score function $\nabla_x \log p_t$. One of the solutions is denoising score matching proposed by Vincent [48], where one, instead of directly matching the true score, leverages conditional score for which initial condition is fixed so that $p_{t|0}$ is analytically known.

More precisely, given the linearity of forward dynamics (1), its exact solution is explicitly known: Let $\mu_t = \int_0^t f_s \, ds$, and $\bar{\sigma}_t^2 = 2 \int_0^t e^{2\mu_s - 2\mu_t} \sigma_s^2 \, ds$. Then the solution is $X_t = e^{-\mu_t} X_0 + \bar{\sigma}_t \xi$, where $\xi \sim \mathcal{N}(0, I)$. We also have $X_t | X_0 \sim \mathcal{N}\left(e^{-\mu_t} X_0, \bar{\sigma}_t^2 I\right)$ and $g_t(x|y) = (2\pi\bar{\sigma}_t^2)^{-d/2} \exp(-\|x - e^{-\mu_t} y\|^2 / (2\bar{\sigma}_t^2))$, which is the density of $X_t | X_0$. Then the objective can be rewritten as

$$\mathcal{L}_{\text{conti}}(\theta) = \frac{1}{2} \int_{t_0}^T w(t) \mathbb{E}_{X_0} \mathbb{E}_{X_t | X_0} \|S(\theta; t, X_t) - \nabla \log g_t(X_t | X_0)\|^2 dt + \frac{1}{2} \int_{t_0}^T w(t) C_t dt$$

$$= \frac{1}{2} \int_{t_0}^T w(t) \frac{1}{\bar{\sigma}_t} \mathbb{E}_{X_0} \mathbb{E}_\xi \|\bar{\sigma}_t S(\theta; t, X_t) + \xi\|^2 dt + \frac{1}{2} \int_{t_0}^T w(t) C_t dt \qquad (4)$$

where $C_t = \mathbb{E}_{X_t} \|\nabla \log p_t\|^2 - \mathbb{E}_{X_0} \mathbb{E}_{X_t | X_0} \|\nabla \log g_t(X_t | X_0)\|^2$. For completeness, we will provide a detailed derivation of these results in Appendix C and emphasize that it is just a review of existing results in our notation.

Throughout this paper, we adopt the variance exploding (VESDE) setting [46], where $f_t = 0$ and hence $\mu_t = 0$, which also aligns with the setup in the classic of EDM [30].

## 3 Error analysis for diffusion-based generative models

In this section, we will quantify both the training and sampling processes, and then integrate them into a more comprehensive generation error analysis.

### 3.1 Training

In this section, we consider a practical implementation of denoising score matching objective, represent the score by a deep ReLU network, and establish the exponential convergence of GD training dynamics.

**Training objective function.** Consider a quadrature discretization of the time integral in (4) based on deterministic[1] collocation points $0 < t_0 < t_1 < t_2 < \cdots < t_N = T$. Then

$$\mathcal{L}_{\text{conti}}(\theta) \approx \bar{\mathcal{L}}(\theta) + \bar{C}, \qquad (5)$$

where $\bar{C} = \frac{1}{2} \sum_{j=1}^N w(t_j)(t_j - t_{j-1}) C_{t_j}$, and

$$\bar{\mathcal{L}}(\theta) = \frac{1}{2} \sum_{j=1}^N w(t_j)(t_j - t_{j-1}) \frac{1}{\bar{\sigma}_{t_j}} \mathbb{E}_{X_0} \mathbb{E}_\xi \|\bar{\sigma}_{t_j} S(\theta; t_j, X_{t_j}) + \xi\|^2. \qquad (6)$$

Define $\beta_j = w(t_j)(t_j - t_{j-1}) \frac{1}{\bar{\sigma}_{t_j}}$ to be the *total weighting*. Consider the empirical version of $\bar{\mathcal{L}}$ (6). Denote the initial data to be $\{x_i\}_{i=1}^n$ with $x_i \sim P_0$, and the noise to be $\{\xi_{ij}\}_{j=1}^N$ with $\xi_{ij} \sim \mathcal{N}(0, I_d)$. Then the input data of the neural network is $\{t_j, X_{ij}\}_{i=1, j=1}^{n,N} = \{t_j, x_i + \bar{\sigma}_{t_j} \xi_{ij}\}_{i=1, j=1}^{n,N}$ and the output data is $\{\xi_{ij}/\bar{\sigma}_{t_j}\}_{i=1, j=1}^{n,N}$ if $\bar{\sigma}_{t_j} \neq 0$. Consequently, $\bar{\mathcal{L}}(\theta)$ (6) can be approximated by the following

$$\bar{\mathcal{L}}_{em}(\theta) = \frac{1}{2n} \sum_{i=1}^n \sum_{j=1}^N \beta_j \|\bar{\sigma}_{t_j} S(\theta; t_j, x_i + \bar{\sigma}_{t_j} \xi_{ij}) + \xi_{ij}\|^2. \qquad (7)$$

We will use (7) as the training objective function in our analysis. For simplicity, we also denote $f(\theta; i, j) = \beta_j \|\bar{\sigma}_{t_j} S(\theta; t_j, x_i + \bar{\sigma}_{t_j} \xi_{ij}) + \xi_{ij}\|^2$ and then $\bar{\mathcal{L}}_{em}(\theta) = \frac{1}{2n} \sum_{i=1}^n \sum_{j=1}^N f(\theta; i, j)$. Note the time dependence can be absorbed into the $X$ dependence. More precisely, because $\bar{\sigma}_t$ is a monotonically increasing function of $t$, we can replace $t_j$ in the inputs by $\bar{\sigma}_{t_j}$ to indicate the time dependence. This is then equivalent to augmenting $X_{ij}$ to be $d + 1$ dimensional with $(x_i)_{d+1} := 0$ and $(\xi_{ij})_{d+1} := 1$. For simplified presentation, we will slightly abuse notation and still use $d$ as the input dimension rather than $d + 1$.

**Architecture.** The analysis of diffusion model training is in general very challenging. One obvious factor is the complex score parameterizations used in practice such as U-Net [42] and transformers [40, 34]. In this paper, we simplify the architecture and consider deep feedforward networks. Although it is still far from practical usage, note this simple structure can already provide insights about the design space, as shown in later sections, and is more complicated than existing works [25, 44] related to the training of diffusion models (see Section 1.1). More precisely, we consider the standard deep ReLU network with bias absorbed:

$$S(\theta; t_j, X_{ij}) = W_{L+1} \sigma(W_L \cdots W_1 \sigma(W_0 X_{ij})), \qquad (8)$$

---

[1] Otherwise it is no longer GD training but stochastic GD.

where $\theta = (W_0, \cdots, W_{L+1})$, $W_0 \in \mathbb{R}^{m \times d}$, $W_{L+1} \in \mathbb{R}^{d \times m}$, $W_\ell \in \mathbb{R}^{m \times m}$ for $\ell = 1, \cdots, L$, and $\sigma(\cdot)$ is the ReLU activation.

**Algorithm.** Let $\theta^{(k)} = (W_0, W_1^{(k)}, \cdots, W_L^{(k)}, W_{L+1})$. We consider the gradient descent (GD) algorithm as follows

$$\theta^{(k+1)} = \theta^{(k)} - h \nabla \bar{\mathcal{L}}_{em}(\theta^{(k)}), \tag{9}$$

where $h > 0$ is the learning rate. We fix $W_0$ and $W_{L+1}$ throughout the training process and only update $W_1, \cdots, W_L$, which is a commonly used setting in the convergence analysis of neural networks [1, 10, 25].

**Initialization.** We employ the same initialization as in Allen-Zhu et al. [1], which is to set $(W_\ell^{(0)})_{ij} \sim \mathcal{N}(0, \frac{2}{m})$ for $\ell = 0, \cdots, L$, $i, j = 1, \cdots, m$, and $(W_{L+1}^{(0)})_{ij} \sim \mathcal{N}(0, \frac{1}{d})$ for $i = 1 \cdots, d$, $j = 1 \cdots, m$.

For this setup, the main challenge in our convergence analysis for denoising score matching lies in the nature of the data. 1) The output data that neural network tries to match is an unbounded Gaussian random vector, and cannot be rescaled as assumed in many theoretical works (for example, Allen-Zhu et al. [1] assumed the output data to be of order $o(1)$). 2) The input data $X_{ij}$ is the sum of two parts: $x_i$ which follows the initial distribution $P_0$, and a Gaussian noise $\bar{\sigma}_{t_j} \xi_{ij}$. Therefore, any assumption on the input data needs to agree with this noisy and unbounded nature, and commonly used assumptions like data separability [1, 35] can no longer be used.

To deal with the above issues, we instead make the following assumptions.

**Assumption 1** (On network hyperparameters and initial data of the forward dynamics). *We assume the following holds:*

1. *Data scaling:* $\|x_i\| = \Theta(d^{1/2})$ *for all $i$.*
2. *Input dimension:* $d = \Omega(\text{poly}(\log(nN)))$.

We remark that the first assumption focuses only on the initial data $x_i$ instead of the whole solution of the forward dynamics $X_{ij}$ which incorporates the Gaussian noise. Also, this assumption is indeed not far away from reality; for example, it holds with high (at least $1 - \mathcal{O}(\exp(-\Omega(d)))$) probability for standard Gaussian random vectors. The requirement for input dimension $d$ is to ensure that $d$ is not too small, or equivalently the number of data points is not exponential in $d$.

We also make the following assumptions on the hyperparameters of the denoising score matching.

**Assumption 2** (On the design of diffusion models). *We assume the following holds:*

1. *Weighting:* $\sum_{j=1}^N w(t_j)(t_j - t_{j-1})\bar{\sigma}_{t_j} = \mathcal{O}(N)$.
2. *Variance:* $\bar{\sigma}_{t_0} > 0$ and $\bar{\sigma}_{t_N} = \Theta(1)$.

The first assumption is to guarantee that the weighting function $w(t)$ is properly scaled. This expression $w(t_j)(t_j - t_{j-1})\bar{\sigma}_{t_j}$ is obtained from proving the upper and lower bounds of the gradient of (7), and is different from the total weighting $\beta_i$ defined above. In the second assumption, $\bar{\sigma}_{t_0} > 0$ ensures the output $\xi_{ij}/\bar{\sigma}_{t_j}$ is well-defined. The $\bar{\sigma}_{t_N} = \Theta(1)$ guarantees that the scales of the noise $\bar{\sigma}_{t_j} \xi_{ij}$ and the initial data $x_i$ are of the same order at the end of the forward process, namely, the initial data $x_i$ is eventually push-forwarded to near Gaussian with the proper variance. Therefore, Assumption 2 aligns with what has been used in practice (see Section 4 and Karras et al. [30], Song et al. [46] for examples).

The following theorem summarizes our convergence result for the training of the score function.

**Theorem 1** (Convergence of GD). *Define a set of indices to be $\mathcal{G}^{(s)} = \{(i,j) | f(\theta^{(s)}; i, j) \geq f(\theta^{(s)}; i', j')$ for all $i', j'\}$. Then given Assumption 1 and 2, for any $\epsilon_{\text{train}} > 0$, there exists some $M(\epsilon_{\text{train}}) = \Omega\left(\text{poly}\left(n, N, d, L, T/t_0, \log(\frac{1}{\epsilon_{\text{train}}})\right)\right)$, s.t., when $m \geq M(\epsilon_{\text{train}})$, $h = \Theta(\frac{nN}{m \min_j w(t_j)(t_j - t_{j-1})\bar{\sigma}_{t_j}})$, and $k = \mathcal{O}(d^{\frac{1-a_0}{2}} n^2 N \log(\frac{d}{\epsilon_{\text{train}}}))$, with probability at least $1 - \mathcal{O}(nN)\exp(-\Omega(d^{2a_0 - 1}))$, we have*

$$\bar{\mathcal{L}}_{em}(\theta^{(k)}) \leq \prod_{s=0}^{k-1} \left(1 - C_5 h \, w(t_{j^*(s)})(t_{j^*(s)} - t_{j^*(s)-1})\bar{\sigma}_{t_{j^*(s)}} \left(\frac{m d^{\frac{a_0 - 1}{2}}}{n^3 N^2}\right)\right) \bar{\mathcal{L}}_{em}(\theta^{(0)})$$

*where the universal constant $C_5 > 0$, $a_0 \in (\frac{1}{2}, 1)$, and $(i^*(s), j^*(s)) = \arg\max_{(i,j) \in \mathcal{G}^{(s)}} w(t_j)(t_j - t_{j-1})\bar{\sigma}_{t_j}$. Moreover, when $K = \Theta(d^{\frac{1-a_0}{2}} n^2 N \log(\frac{d}{\epsilon_{\text{train}}}))$,*

$$\bar{\mathcal{L}}_{em}(\theta^{(K)}) \leq \epsilon_{\text{train}}.$$

The above theorem implies that for denoising score matching objective $\bar{\mathcal{L}}_{em}(\theta)$, GD has exponential convergence. For example, if we simply take $j^* = \min_j w(t_j)(t_j - t_{j-1})\bar{\sigma}_{t_j}$, then $\bar{\mathcal{L}}_{em}(\theta^{(k+1)})$ is further upper bounded by $\left(1 - C_6 h \ w(t_{j^*})(t_{j^*} - t_{j^*-1})\bar{\sigma}_{t_{j^*}} \left(\frac{md^{\frac{a_0-1}{2}}}{n^3 N^2}\right)\right)^{k+1} \bar{\mathcal{L}}_{em}(\theta^{(0)})$. The rate of convergence can be interpreted in the following way: 1) at the $k$th iteration, we collect all the indices of the time points into $\mathcal{G}^{(k)}$ where $f(\theta; i, j)$ has the maximum value; 2) we then choose the maximum of $w(t_j)(t_j - t_{j-1})\bar{\sigma}_{t_j}$ among all such indices and denote the index to be $j^*(k)$, and obtain the decay ratio bound for the next iteration as $1 - C_6 h \ w(t_{j^*(k)})(t_{j^*(k)} - t_{j^*(k)-1})\bar{\sigma}_{t_{j^*(k)}} \left(\frac{md^{\frac{a_0-1}{2}}}{n^3 N^2}\right)$.

**Remark 1** (Can $\epsilon_{\text{train}}$ be arbitrarily small? Some ramifications of the denoising setting)**.** *Let us first see some facts about $\bar{\mathcal{L}}_{em}$ and $\bar{\mathcal{L}}$. Under minimal assumption of the existence of score function and in the zero-time-discretization-error limit, the score matching objective can be made zero and therefore the denoising score matching objective is bounded below by $-\bar{C}$, which is nonnegative and zero only when the data distribution is extremely special (we thus write $-\bar{C} > 0$ from hereon unless confusion arises). That is, $\min_\theta \bar{\mathcal{L}}(\theta) \geq \min_{any\ function\ S} \bar{\mathcal{L}} = -\bar{C} > 0$ according to (4). Since $\bar{\mathcal{L}}_{em} \to \bar{\mathcal{L}}$ as the sample size of the training data set $n \to \infty$, we have $\bar{\mathcal{L}}_{em} \geq -\bar{C} - c_n > 0$ for some constant $c_n > 0$ and $c_n \to 0$ as $n \to \infty$.*

*However, Theorem 1 seems to imply $\bar{\mathcal{L}}_{em}(\theta^{(k)}) \to 0$ as $k \to \infty$ since $\bar{\mathcal{L}}_{em}(\theta^{(K)}) \leq \epsilon_{\text{train}}$ and $\epsilon_{\text{train}}$ is arbitrary, and it seems to contradict the $-\bar{C} > 0$ lower bound. However, there is no contradiction due to the combination of two facts. First, the theorem states that for arbitrary $\epsilon_{\text{train}} > 0$, there exists a critical size, such that for overparameterized network beyond this size, GD can render the loss $\bar{\mathcal{L}}_{em}(\theta)$ eventually no greater than $\epsilon_{\text{train}}$. If we fix the network size, i.e., with $m, L, d$ given, then $K$ is given, and Theorem 1 says nothing about GD's behavior after $K$ iterations. That is, we do not know whether $\limsup_{k \to \infty} \bar{\mathcal{L}}_{em}(\theta^{(k)}) = 0$. Second, our optimization setting requires the sample size $n$ to be smaller than the network width $m$ (Assumption 1). Thus, when $m$ is fixed, the sample size $n$ is upper bounded.*

*The above discussion implies, within the validity of our theory, for any fixed network width $m$, if $\epsilon_{\text{train}}$ is small, the sample size $n$ cannot be too large, meaning $\bar{\mathcal{L}}_{em}(\theta) - \bar{\mathcal{L}}(\theta)$ may not be small. Therefore, we can simultaneously have $\bar{\mathcal{L}}_{em}(\theta)$ close to 0 and $\bar{\mathcal{L}}(\theta)$ close to $-\bar{C} > 0$.*

**Main technical steps for proving Theorem 1.** The proof of Theorem 1 is in Appendix D, where the analysis framework is adapted from Allen-Zhu et al. [1]. Roughly speaking, the key proof in this framework is to establish the lower bound of the gradient. Then by integrating it into the semi-smoothness property of the neural network, we can obtain the exponential rate of convergence of gradient descent. For the lower bound of gradient, we develop a new method to deal with the difficulties in the denoising score matching setting (see the discussions earlier in this section).

Our new proof technique adopts a different decoupling of the gradient and leverages a high probability bound based on a high-dimensional geometric idea. See Appendix D.1 for a proof sketch and more details.

### 3.2 Sampling

In this section, we establish a nonasymptotic error bound of the backward process in the variance exploding setting, which is an extension to Benton et al. [8]. For simplified notations, denote the backward time schedule as $\{t_j^\leftarrow\}_{0 \leq j \leq N}$ such that $0 = t_0^\leftarrow < t_1^\leftarrow < \cdots < t_N^\leftarrow = T - \delta$.

**Generation algorithm.** We consider the exponential integrator scheme for simulating the backward SDE (2) with $f_t \equiv 0$[2]. The generation algorithm can be piecewise expressed as a continuous-time SDE: for any $t \in [t_j^\leftarrow, t_{j+1}^\leftarrow)$,

$$d\bar{Y}_t = 2\sigma_{T-t}^2 S(\theta; T - t_j^\leftarrow, \bar{Y}_{t_j^\leftarrow})dt + \sqrt{2\sigma_{T-t}^2}d\bar{W}_t. \tag{10}$$

**Initialization.** Denote $q_t := \text{Law}(\bar{Y}_t)$ for all $t \in [0, T - \delta]$. We choose the Gaussian initialization, $q_0 = \mathcal{N}(0, \bar{\sigma}_T^2)$.

Our convergence result relies on the following assumption.

**Assumption 3.** *The distribution $P_0$ has a finite second moment:* $\mathbb{E}_{x \sim P_0}[\|x\|^2] = m_2^2 < \infty$.

Next we state the main convergence result, whose proof is provided in Appendix E.

---

[2]The exponential integrator scheme is degenerate since $f_t \equiv 0$. Time discretization is applied when we evaluate the score approximations $\{S(\theta; t, \bar{Y}_t)\}$.

**Theorem 2.** *Under Assumption 3, for any $\delta \in (0,1)$ and $T > 1$, we have*

$$KL(p_\delta|q_{T-\delta}) \lesssim \underbrace{\frac{\mathrm{m}_2^2}{\bar{\sigma}_T^2}}_{E_I} + \underbrace{\sum_{j=0}^{N-1} \gamma_j \sigma_{T-t_j^\leftarrow}^2 \mathbb{E}_{Y_{t_j^\leftarrow} \sim p_{T-t_j^\leftarrow}}[\|S(\theta; T-t_j^\leftarrow, Y_{t_j^\leftarrow}) - \nabla \log p_{T-t_j^\leftarrow}(Y_{t_j^\leftarrow})\|^2]}_{E_S}$$

$$+ \underbrace{d \sum_{j=0}^{N-1} \gamma_j \int_{t_j^\leftarrow}^{t_{j+1}^\leftarrow} \frac{\sigma_{T-t}^4}{\bar{\sigma}_{T-t}^4} dt + \mathrm{m}_2^2 \frac{\int_0^{t_1^\leftarrow} \sigma_{T-t}^2 dt}{\bar{\sigma}_T^4} + (\mathrm{m}_2^2 + d) \sum_{j=1}^{N-1} (1 - e^{-\bar{\sigma}_{T-t_j^\leftarrow}^2}) \frac{\bar{\sigma}_{T-t_j^\leftarrow}^4 - \bar{\sigma}_{T-t_{j+1}^\leftarrow}^2 \bar{\sigma}_{T-t_{j-1}^\leftarrow}^2}{\bar{\sigma}_{T-t_{j-1}^\leftarrow}^2 \bar{\sigma}_{T-t_j^\leftarrow}^4}}_{E_D} \cdot$$

(11)

*where $\gamma_j := t_{j+1}^\leftarrow - t_j^\leftarrow$ for all $j = 0, 1, \cdots, N-1$ is the stepsize of the generation algorithm in* (10).

Theorem 2 is a VESDE-based diffusion model's analogy of what's proved in Benton et al. [8] for VPSDE-based diffusion model, only requiring the data distributions to have finite second moments, and it achieves the sharp almost linear data dimension dependence under the exponential time schedule. The major differences from [8] are (1) the initialization error in the VESDE case is handled differently (see Lemma 10); (2) Theorem 2 applies to varies choices of time schedules, which enables to investigate the design space of the diffusion model, as we will discuss in Section 4. Worth mentioning is, Yang et al. [54] also obtained polynomial complexity results for VESDE-based diffusion models with uniform stepsize, but under stronger data assumption (assuming compact support). Compared to their result, complexity implied by Theorem 2 has better accuracy and data dimension dependencies. A detailed discussion on complexities is given in Appendix I.1.

Terms $E_I, E_D, E_S$ in (11) represent the three types of errors: initialization error, discretization error, and score estimation error, respectively. Term $E_I$ quantifies the error between the initial density of the sampling algorithm $q_0$ and the ideal initialization $p_T$, which is the density when the forward process stops at time $T$. Term $E_D$ is the error stemming from the discretization of the backward dynamics. Term $E_S$ characterizes the error of the estimated score function and the true score, and is related to the optimization error of $\bar{\mathcal{L}}_{em}$. Important to note is, in Theorem 2, population loss is needed instead of the empirical version $\bar{\mathcal{L}}_{em}$ (7). Besides this, the weighting $\gamma_j \sigma_{T-t_j^\leftarrow}^2$ is not necessarily the same as the total weighting in $\bar{\mathcal{L}}_{em}$ (7) $\beta_j$, depending on choices of $w(t_j)$ and time and variance schedules (see more discussion in Section 4). We will later on integrate the optimization error (Theorem 1) into this score error $E_S$ to obtain a full error analysis in Section 3.3.

**Remark 2** (sharpness of dependence in $d$ and $\mathrm{m}_2^2$)**.** *In one of the simplest cases, when the data distribution is Gaussian, the score function is explicitly known. Hence $KL(p_\delta|q_{T-\delta})$ can be explicitly computed as well, which verifies that the dependence of parameters $d$ and $\mathrm{m}_2^2$ is sharp in $E_I$ and $E_D$.*

### 3.3 Full error analysis

In this section, we combine the analyses from the previous two sections to obtain an end-to-end generation error bound.

Before providing the main result of this section, let us first clarify some terminologies.

**Time schedule, variance schedule, and total weighting.** The terms *time schedule* and *variance schedule* respectively refer to the choice of $t_j^\leftarrow$ and $\bar{\sigma}_{t_j}$ in sampling. Meanwhile, note both the training and sampling processes require the proper choices of time and variance, and these choices are not necessarily the same for both processes. For training, the effect of these two is integrated into the *total weighting* $\beta_j$, which is also influenced by an additional weighting parameter $w(t_j)$. In this theoretical paper, when studying the generation error, we aim to apply the optimization result to better understand the effect of optimization on sampling. Therefore, to simplify the analysis and discussions in Section 4, we choose the same time and variance schedules for both training and sampling.

The main result is stated in the following.

**Theorem 3.** *Under the same conditions as Theorem 1,2, and that $K$ is such that GD reaches $\epsilon_{train}$ in at most $K$th iterations, we have*

$$KL(p_\delta|q_{T-\delta}) \lesssim E_I + E_D + \max_{1 \leq j \leq N} \frac{\sigma_{t_{N-j}}^2}{w(t_{N-j})}(\epsilon_{\text{train}} + \epsilon_n + \epsilon_{\text{est}} + \epsilon_{\text{approx}})$$

*where $E_I, E_D$ are defined in Theorem 2, $\epsilon_{\text{train}}$ is defined in Theorem 1, $\epsilon_n = |\bar{\mathcal{L}}(\theta^{(K)}) - \bar{\mathcal{L}}_{em}(\theta^{(K)}) + \bar{\mathcal{L}}_{em}(\theta^*) - \bar{\mathcal{L}}(\theta^*)|$, $\epsilon_{\text{est}} = |\bar{\mathcal{L}}(\theta^*) - \bar{\mathcal{L}}(\theta_\mathcal{F})|$, $\epsilon_{\text{approx}} = |\bar{\mathcal{L}}(\theta_\mathcal{F}) + \bar{C}|$. In these terms, $\bar{C}$ is defined in* (5),

$\theta^* = \arg\min_{\theta, s.t., \bar{\mathcal{L}}_{em}(\theta)=0} \bar{\mathcal{L}}(\theta)$ *and* $\theta_{\mathcal{F}} = \arg\inf_{\{\theta:S(\theta)\in\mathcal{F}\}} |\bar{\mathcal{L}}(\theta) + \bar{C}|$ *with* $\mathcal{F} = \{$*ReLU network function defined in* (8)*, with* $d = \Omega(\text{poly}(\log(nN))), m = \Omega\left(poly\left(n, N, d, L, T/t_0\right)\right)\}$.

In this theorem, the discretization error $E_D$ and initialization error $E_I$ are the same as Theorem 2. For the score error $E_S$, our optimization result is valid for general time schedules and therefore can directly fit into the sampling error analysis, which is in contrast to existing works [25, 44] (see more discussions in Section 1.1). The coefficient $\max_j \sigma_{t_{N-j}}^2 / w(t_{N-j})$ results from different weightings in $E_S$ and $\bar{\mathcal{L}}_{em}$, i.e., $\gamma_j \sigma_{T-t_j^{\leftarrow}}^2$ and $\beta_j$. We will discuss the effect of $\max_j \sigma_{t_{N-j}}^2 / w(t_{N-j})$ under different time and variance schedules in Section 4.

The way we bound $\mathbb{E}_{Y_{t_j^{\leftarrow}} \sim p_{T-t_j^{\leftarrow}}}[\|S(\theta; T - t_j^{\leftarrow}, Y_{t_j^{\leftarrow}}) - \nabla \log p_{T-t_j^{\leftarrow}}(Y_{t_j^{\leftarrow}})\|^2]$ in $E_S$ (see Theorem 2) is to decompose it into the optimization error $\epsilon_{\text{train}}$, statistical error $\epsilon_n$, estimation error $\epsilon_{\text{est}}$, and approximation error $\epsilon_{\text{approx}}$. This gives clear intuition to results, but we also note it may not give a tight bound. In fact, we have

$$\epsilon_n + \epsilon_{\text{train}} = |\bar{\mathcal{L}}(\theta^{(K)}) - \bar{\mathcal{L}}_{em}(\theta^{(K)}) + \bar{\mathcal{L}}_{em}(\theta^*) - \bar{\mathcal{L}}(\theta^*)| + |\bar{\mathcal{L}}_{em}(\theta^{(K)}) - \bar{\mathcal{L}}_{em}(\theta^*)|$$
$$\geq \bar{\mathcal{L}}(\theta^{(K)}) + \bar{\mathcal{L}}(\theta^*) \geq 2\min_\theta \bar{\mathcal{L}}(\theta) \geq -2\bar{C} > 0.$$

$\epsilon_n$ can still be small if we take $n \to \infty$, but that means $\epsilon_{\text{train}}$ has to be large, and our generation error bound cannot be made 0. It is unclear yet whether this is due to limitation of our analysis or intrinsic, and will be left for future investigation.

Another related note is, in this paper, we focus on $\epsilon_{\text{train}}$ and the effect of optimization, but the analyses of $\epsilon_n$, $\epsilon_{\text{est}}$, and $\epsilon_{\text{approx}}$ are also important and possible [13, 39, 25, 50]. On the other hand, again, whether it is optimal to decompose the full error into these four is unclear.

To better see the parameter dependence of the error bound in Theorem 3, the following is an example with simplified results, where we employ the schedules in EDM [30].

**Corollary 1** (Full error analysis under EDM [30] designs)**.** *Under the same conditions as Theorem 3, we have*

$$KL(p_\delta | q_{T-\delta}) \lesssim \frac{\mathrm{m}_2^2}{T^2} + \frac{da^2 T^{\frac{1}{a}}}{\delta^{\frac{1}{a}} N} + (\mathrm{m}_2^2 + d)\left(\frac{a^2 T^{\frac{1}{a}}}{\delta^{\frac{1}{a}} N} + \frac{a^3 T^{\frac{2}{a}}}{\delta^{\frac{2}{a}} N^2}\right) + \frac{1}{N}\left(C_9 + \left(1 - C_8 h\left(\frac{md^{\frac{a_0-1}{2}}}{n^3 N^2}\right)\right)^K\right),$$

*where* $C_8, C_9 > 0$ *and* $a = 7$ *in [30].*

## 4 Theory-based understanding of the design space and its relation to existing empirical counterparts

This section theoretically explores preferable choices of parameters in both training and sampling, and shows that they agree with the ones used in EDM [30] and Song et al. [46] in different circumstances.

### 4.1 Choice of total weighting for training

This section develops the optimal total weighting $\beta_j$ for training objective (7). We qualitatively show in two steps that "bell-shaped" weighting, which is the one used in EDM [30], will lead to the optimal rate of convergence: Step 1) $\|\bar{\sigma}_{t_j} S(\theta; t_j, X_{ij}) + \xi_{ij}\|$ as a function of $j$ is inversely "bell-shaped"; Step 2) $f(\theta; i, j) = \beta_j \|\bar{\sigma}_{t_j} S(\theta; t_j, X_{ij}) + \xi_{ij}\|$ should be close to each other for any $i, j$.

#### 4.1.1 Inversely "bell-shaped" loss $\|\bar{\sigma}_{t_j} S(\theta; t_j, X_{ij}) + \xi_{ij}\|$ as a function of time index $j$
**Proposition 1.** *Under the same assumptions as Theorem 1, for any $\theta$ and $i = 1, \cdots, n$, we have*

*1.* $\forall \epsilon_1 > 0$, $\exists \delta_1 > 0$, *s.t., when* $0 \leq \bar{\sigma}_{t_j} < \delta_1$, $\|\bar{\sigma}_{t_j} S(\theta; t_j, x_i + \bar{\sigma}_{t_j} \xi_{ij}) + \xi_{ij}\| > \|\xi_{ij}\| - \epsilon_1$.

*2.* $\forall \epsilon_2 > 0$, $\exists M > 0$, *s.t., when* $\bar{\sigma}_{t_j} > M$, $\|\bar{\sigma}_{t_j} S(\theta; t_j, x_i + \bar{\sigma}_{t_j} \xi_{ij}) + \xi_{ij}\| \geq M^2(\|S(\theta; t_j, \xi_{ij})\| - \epsilon_2)$.

The above proposition can be interpreted in the following way. Given any network $S$, when $\bar{\sigma}_{t_j}$ is very small, 1 implies that $\|\bar{\sigma}_{t_j} S(\theta; t_j, x_i + \bar{\sigma}_{t_j} \xi) + \xi_{ij}\|$ is away from 0 by approximately $\|\xi_{ij}\|$ which is of order $\sqrt{d}$ with high probability, i.e., it cannot be small. When $\bar{\sigma}_{t_j}$ is large, 2 shows that as it becomes larger and larger, i.e., as $M$ increases, $\|\bar{\sigma}_{t_j} S(\theta; t_j, x_i + \bar{\sigma}_{t_j} \xi) + \xi_{ij}\|$ will also increase. Therefore, the function $\|\bar{\sigma}_{t_j} S(\theta; t_j, X_{ij}) + \xi_{ij}\|$ has most likely an inversely "bell-shaped" curve in terms of $j$ dependence.

#### 4.1.2 Ensuring comparable values of $f(\theta; i, j)$ for optimal rate of convergence

**Corollary 2.** *Under the same conditions as Theorem 1, for some large $K' > 0$, if $|f(\theta^{(k+K')}; i, j) - f(\theta^{(k+K')}; l, s)| \le \epsilon$ holds for all $k > 0$ and all $(i, j), (l, s)$, with some small universal constant $\epsilon > 0$, then we have, for some constant $C_7 > 0$,*

$$\bar{\mathcal{L}}_{em}(\theta^{(k+K')}) \le \left( 1 - C_7 h \max_{j=1,\cdots,N} w(t_j)(t_j - t_{j-1})\bar{\sigma}_{t_j} \left( \frac{md^{\frac{a_0-1}{2}}}{n^3 N^2} \right) \right)^k \bar{\mathcal{L}}_{em}(\theta^{(K')}).$$

The above corollary shows that if $f(\theta^{(k)}; i, j)$'s are almost the same for any $i, j$, then the decay ratio of the next iteration is minimized. More precisely, the index set $\mathcal{G}^{(k)}$ defined in Theorem 1 is roughly the whole set $\{1, \cdots, N\}$, and therefore $w(t_{j^*})(t_{j^*} - t_{j^*-1})\bar{\sigma}_{t_{j^*}}$ can be taken as the maximum value over all $j$, which consequently leads to the optimal rate.

#### 4.1.3 "Bell-shaped" weighting: our theory and EDM

Combining the above two aspects, the optimal rate of convergence leads to the choice of total weighting $\beta_j$ such that $f(\theta; i, j) = \beta_j \|\bar{\sigma}_{t_j} S(\theta; t_j, X_{t_j}) + \xi_{ij}\|$ is close to each other; as a result, the total weighting should be chosen as a "bell-shaped" curve as a function of $j$ according to the shape of the curve for $\|\bar{\sigma}_{t_j} S(\theta; t_j, X_{t_j}) + \xi_{ij}\|$.

Figure 2: Weighting choice $\beta_{\mathrm{EDM}}$ in EDM.

Before comparing the preferable weighting predicted by our theory and the intuition-and-empirics-based one in EDM [30], let us first recall that the EDM training objective[3] can be written as $\mathbb{E}_{\bar{\sigma} \sim p_{\text{train}}} \mathbb{E}_{y,n} \lambda(\bar{\sigma}) \|D_\theta(y + n; \bar{\sigma}) - y\|^2$

$$= \frac{1}{Z_1} \int e^{-\frac{(\log \bar{\sigma} - P_{\text{mean}})^2}{2P_{\text{std}}^2}} \frac{\bar{\sigma}^2 + \sigma_{\text{data}}^2}{\bar{\sigma}\sigma_{\text{data}}^2} \mathbb{E}_{X_0,\xi} \|\bar{\sigma}s(\theta; t, X_t) + \xi\|^2 \, d\bar{\sigma}, \tag{12}$$

where $Z_1$ is a normalization constant, and we denote $\beta_{\mathrm{EDM}}(\bar{\sigma}) = e^{-\frac{(\log \bar{\sigma} - P_{\text{mean}})^2}{2P_{\text{std}}^2}} \frac{\bar{\sigma}^2 + \sigma_{\text{data}}^2}{\bar{\sigma}\sigma_{\text{data}}^2}$ to be the *total weighting* of EDM. Note the dependence on $\bar{\sigma}$ and time $j$ can be freely switched due to their 1-to-1 correspondence.

Figure 2 plots the total weighting of EDM $\beta_{\mathrm{EDM}}$ as a function of $\bar{\sigma}$. As is shown in the picture, this is a "bell-shaped" curve[4], which coincides with our choice of total weighting in the above theory. When $\bar{\sigma}$ is very small or very large, according to Proposition 1, the lower bound of $\|\bar{\sigma}_{t_j} S(\theta; t_j, X_{t_j}) + \xi_{ij}\|$ cannot vanish and therefore needs the smallest weighting over all $\bar{\sigma}$. When $\bar{\sigma}$ takes the middle value, the scale of the output data $\xi_{ij}/\bar{\sigma}_j$ is roughly the same as the input data $X_{ij}$ and therefore makes it easier for the neural network to fit the data, which admits larger weighting.

### 4.2 Choice of time and variance schedules

This section will discuss the choice of time and variance schedules based on the three errors $E_S, E_D, E_I$ in the error analysis of Section 3.3. Two situations will be considered based on how well the score function is approximated in training: when the network is less trained, $E_S$ dominates and polynomial schedule [30] is preferable; when the score function is well approximated, $E_D + E_I$ dominates and exponential schedule [46] is better.

#### 4.2.1 When score error $E_S$ dominates

As is shown in Theorem 3, the main impact of different time and variance schedules on score error $E_S$ appears in the term $\max_j \sigma_{t_{N-j}}^2 / w(t_{N-j})$, when the score function is approximated to a certain accuracy. It remains to compute $w(t)$ under various choices of schedules.

**General rule of constructing $w(t)$.** To ensure fair comparisons between different time and variance schedules, we maintain a fixed total weighting in the training objective. Additionally, to facilitate comparisons with practical usage, we adopt the total weighting in EDM, i.e., $\beta_j = C_3 \beta_{\mathrm{EDM}}(\bar{\sigma}_{t_j})$, for some universal constant $C_3 > 0$. The reason for using the EDM total weighting is that according to Section 4.1, our total weighting $\beta_j$ should be "bell-shaped" as a function of $j$, which agrees qualitatively with the one used in EDM.

---

[3] In EDM [30], they use $P_{\text{mean}} = -1.2$, $P_{\text{std}} = 1.2$, $\sigma_{\text{data}} = 0.5$, $\bar{\sigma}_{\min} = 0.002$, $\bar{\sigma}_{\max} = 80$.

[4] This horizontal axis is in $\log$-scale and the plot in regular scale is a little bit skewed, not precisely a "bell" shape. However, we remark that the trend of the curve still matches our theory.

**Polynomial schedule [30] vs exponential schedule [46].** We fix $\epsilon_n, \epsilon_{\text{train}}$ and apply the two schedules (Table 1) separately to the above total weighting $\beta$ (hence $w$). Then, compute $\max_j \sigma_{t_{N-j}}^2 / w(t_{N-j})$ which is a factor in score error $E_S$ (Thm.3) in Table 2. The Exp.'s result $\frac{1}{2}\left(\bar{\sigma}_{\max} - \bar{\sigma}_{\max}\left(\frac{\bar{\sigma}_{\min}^2}{\bar{\sigma}_{\max}^2}\right)^{1/N}\right)$ is larger[5] than the Poly.'s result $\left(\bar{\sigma}_{\max} - \left(\bar{\sigma}_{\max}^{1/\rho} - \frac{\bar{\sigma}_{\max}^{1/\rho} - \bar{\sigma}_{\min}^{1/\rho}}{N}\right)^{\rho}\right)$ for large $N$, meaning the poly. time schedule in EDM is better than the exp. schedule in [46]. Note these two terms are both of order $1/N$ as $N \to \infty$ and therefore the difference lies in their prefactors.

Table 1: Polynomial and exponential (time) schedules.

| | Variance schedule $\bar{\sigma}_t$ | Time schedule $t_j$ |
|---|---|---|
| Poly. [30] | $t$ | $\left(\bar{\sigma}_{\max}^{1/\rho} - (\bar{\sigma}_{\max}^{1/\rho} - \bar{\sigma}_{\min}^{1/\rho})\frac{N-j}{N}\right)^{\rho}$ |
| Exp. [46] | $\sqrt{t}$ | $\bar{\sigma}_{\max}^2 \left(\frac{\bar{\sigma}_{\min}^2}{\bar{\sigma}_{\max}^2}\right)^{\frac{N-j}{N}}$ |

Table 2: Comparisons between different schedules.

| | $E_S$ (score error) dominates | | $E_D + E_I$ (sampling error) dominates | |
|---|---|---|---|---|
| | $\max_j \sigma_{t_j}^2 / w(t_j)$ | Choice | $N$ | Choice |
| Poly. [30] | $C_4\left(\bar{\sigma}_{\max} - \left(\bar{\sigma}_{\max}^{1/\rho} - \frac{\bar{\sigma}_{\max}^{1/\rho} - \bar{\sigma}_{\min}^{1/\rho}}{N}\right)^{\rho}\right)$ | ✓ | $\Omega\left(\frac{m_2^2 \vee d}{d}\rho^2\left(\frac{\bar{\sigma}_{\max}}{\bar{\sigma}_{\min}}\right)^{1/\rho}\bar{\sigma}_{\max}^2\right)$ | |
| Exp. [46] | $C_4 \cdot \frac{1}{2}\left(\bar{\sigma}_{\max} - \bar{\sigma}_{\max}\left(\frac{\bar{\sigma}_{\min}^2}{\bar{\sigma}_{\max}^2}\right)^{1/N}\right)$ | | $\Omega\left(\frac{m_2^2 \vee d}{d}\ln\left(\frac{\bar{\sigma}_{\max}}{\bar{\sigma}_{\min}}\right)^2\bar{\sigma}_{\max}^2\right)$ | ✓ |

#### 4.2.2 When discretization error $E_D$ and initialization error $E_I$ dominate

In this section, we compare the two different schedules in Table 1 by studying the iteration complexity of the sampling algorithm, i.e., number of time points $N$, when $E_D + E_I$ dominates.

**General rules of comparison.** We consider the case when the discretization and initialization errors are bounded by the same quantity $\epsilon$, i.e., $E_I + E_D \lesssim \varepsilon$. Then according to Theorem 2 and Theorem 3, we compute the iteration complexity for achieving this error using the two schedules in Table 1. To make the comparison more straightforward, we adopt $T = t_N = \Theta(\text{poly}(\varepsilon^{-1}))$ and therefore $\bar{\sigma}_{\max} = \Theta(\varepsilon^{-1/2})$. More details are provided in Appendix I.1.

**Polynomial schedule [30] vs exponential schedule [46].** As is shown in the last column of Table 2, the iteration complexity under exponential schedule [46] has the poly-logarithmic dependence on the ratio between maximal and minimal variance $(\bar{\sigma}_{\max}/\bar{\sigma}_{\min})$[6], which is better than the complexity under polynomial schedule [30], which is polynomially dependent on $\bar{\sigma}_{\max}/\bar{\sigma}_{\min}$. Both complexities are derived from Theorem 2 by choosing different parameters.

**Remark 3** (The existence of optimal $\rho$ in the polynomial schedule [30])**.** *For fixed $\bar{\sigma}_{\max}$ and $\bar{\sigma}_{\min}$, the optimal $\rho$ that minimizes the iteration complexity is $\rho = \frac{1}{2}\ln\left(\frac{\bar{\sigma}_{\max}}{\bar{\sigma}_{\min}}\right)$. In [30], it was empirically observed that with fixed iteration complexity, there is an optimal value of $\rho$ that minimizes the FID. Our result indicates that, for fixed $\bar{\sigma}_{\max}$ and $\bar{\sigma}_{\min}$, hence the desired accuracy in KL divergence being fixed, there is an optimal value of $\rho$ that minimizes the iteration complexity to reach the fixed accuracy. Even though we consider a different metric/divergence instead of FID, our result still provides a quantitative support to the existence of optimal $\rho$ observed in [30].*

## Acknowledgments and Disclosure of Funding

The authors are grateful for the partially support by NSF DMS-1847802, Cullen-Peck Scholarship, and GT-Emory Humanity.AI Award. We thank the anonymous reviewers for their helpful comments.

---

[5]This holds under parameters used in either Song et al. [46] or Karras et al. [30].

[6]The exponential time schedule under the variance schedule in [30] also has the poly-logarithmic dependence on $\bar{\sigma}_{\max}/\bar{\sigma}_{\min}$. Under both variance schedules in [30] and [46], it can be shown that exponential time schedule is optimal. Details are provided in Appendix I.1.

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

# Appendix

## A    Conclusions and limitations

**Conclusions.** In this paper, we provide a first full error analysis incorporating both optimization and sampling processes. For the training process, we provide a first result under a deep neural network and prove the exponential convergence into a neighborhood of minima. At the same time, we extend the current analysis to the variance exploding case for sampling. Moreover, based on the full error analysis, we establish a quantitative understanding of the error bound under the two schedules. Consequently, we conclude with a qualitative illustration of the "bell-shaped" weighting and the choices of schedules under well-trained and less-trained cases.

**Limitations.** The network architecture we used in the model is a deep ReLU network. Although being so far the most complicated architecture for theoretical results, it is still far from what is used in practice like U-Nets and transformers. Moreover, regarding the full error analysis, we only focus on the optimization and sampling error and do not dissect the generalization error. When bridging the theoretical results with practical designs of diffusion models, our results are mostly qualitative and we only compare two existing schedules under two extreme cases, when the network is well-trained and less-trained. Thus, theoretical implications on practical designs remain to be explored. We will leave these perspectives for future exploration.

## B    Notations

| | |
|---|---|
| $X_t$ | Solution of forward dynamics (1) |
| $Y_t$ | Solution of backward dynamics (2) |
| $\bar{Y}_t$ | Solution of generation algorithm (10) |
| $\sigma_t$ | Diffusion coefficient of (1) and (2) |
| $\bar{\sigma}_t$ | Standard deviation of $X_t$ (4) |
| $\mathcal{L}_{\text{conti}}$ | Continuous-time score-matching objective (3) |
| $\bar{\mathcal{L}}$ | Discrete-time denoising score-matching objective (population version) |
| $\bar{\mathcal{L}}_{em}$ | Discrete-time denoising score-matching objective (empirical version) (7) |
| $C_t$ | Constant between score-matching and denoising score-matching loss at time $t$ (4) |
| $\bar{C}$ | Constant between score-matching and denoising score-matching loss over all discrete times (5) |
| $x_i$ | Sample from the initial data distribution $P_0$ (7) |
| $X_{ij}$ | Sample from the distribution $P_t$ at time $t$ (7) |
| $t_j$ | The $j$th time point for forward process (6) |
| $t_j^{\leftarrow}$ | The $j$th time point for backward process (10) |
| $\delta$ | The first (last) time point of the forward (backward) dynamics, i.e., $t_0$ (11) |
| $T$ | Stopping time of the forward dynamics (11) |
| $\gamma_j$ | Difference between backward time points, $t_{j+1}^{\leftarrow} - t_j^{\leftarrow}$ (11) |
| $p_t$ | Density of the solution of forward dynamics at time $t$ (and backward dynamics at time $T-t$) (11) |
| $q_t$ | Density of the solution of the generation algorithm at time $t$ (11) |
| $w(t)$ | Weighting function (3) |
| $\beta_j$ | Total weighting, i.e. $w(t_j)(t_j - t_{j-1})/\bar{\sigma}_{t_j}$ (7) |
| $\beta_{\text{EDM}}$ | Total weighting used in EDM [30] (12) |
| $\bar{\sigma}_{\max}\ (\bar{\sigma}_{\min})$ | Maximum (minimum) of $\bar{\sigma}_{t_j}$ (Table 2) |
| $n$ | Number of samples from the initial distribution $P_0$ (7) |
| $N$ | Number of time steps when discretizing the forward and backward dynamics (6) |
| $d$ | Dimension of input, output data, and the solutions of the dynamics (1) and (2) |
| $S$ | Deep ReLU network (parameterization of score function) |
| $\theta$ | All the parameters in the network $S$ (8) |
| $W_\ell$ | The weight in the $i$th layer of the network $S$ (8) |
| $\theta^{(k)}$ | The $k$th iteration of the weights $\theta$ through GD (9) |
| $W_\ell^{(k)}$ | The $k$th iteration of the weights $W_\ell$ through GD (9) |
| $m$ | Width of the network (8) |

| | |
|---|---|
| $L$ | Depth of the network (8) |
| $(i^*(s), j^*(s))$ | Index of the largest loss and $w(t_j)(t_j - t_{j-1})\bar\sigma_{t_j}$ at the $s$th iteration (Theorem 1) |
| $\mathrm{m}_2^2$ | Second moment of the initial distribution $P_0$ (11) |
| $E_I$ | Initialization error (11) |
| $E_D$ | Discretization error (11) |
| $E_S$ | Score error (11) |
| $\epsilon_{\text{train}}$ | Optimization error (Theorem 1) |
| $\epsilon_n$ | Statistical error (Theorem 3) |
| $\epsilon_{\text{est}}$ | Estimation error (Theorem 3) |
| $\epsilon_{\text{approx}}$ | Approximation error (Theorem 3) |
| $\theta^*$ | Minimum point of $\bar{\mathcal{L}}$ when $\bar{\mathcal{L}}_{em} = 0$ (Theorem 3) |
| $\theta_{\mathcal{F}}$ | Optimal parameter in the function class (Theorem 3) |

## C  Derivation of denoising score matching objective

In this section, we will derive the denoising score matching objective, i.e. show the equivalence of (3) and (4). For simplicity, we denote $S_\theta$ to be the neural network we use $S(\theta; t, X_t)$.

Consider

$$\mathbb{E}_{X_t \sim P_t} \|S(\theta; t, X_t) - \nabla \log p_t\|^2 = \mathbb{E}_{X_t} \left[ \|S_\theta\|^2 - 2\langle S_\theta, \nabla \log p_t \rangle \right] + \mathbb{E}_{X_t} \|\nabla \log p_t\|^2, \qquad (13)$$

where $p_t$ is the density of $X_t$.

Since $p_t(x) = \int p_0(y) q_t(x|y) dy$, where $q_t(\cdot)$ is the density of $X_t|X_0$, then we have

$$\begin{aligned}
\mathbb{E}_{X_t} \langle S_\theta, \nabla \log p_t \rangle &= \int S_\theta^\intercal \nabla \log p_t \cdot p_t \, dx_t \\
&= \int S_\theta^\intercal \nabla p_t \, dx_t \\
&= \iint S_\theta^\intercal \nabla q_t(x|y) p_0(y) \, dxdy \\
&= \iint S_\theta^\intercal \nabla \log q_t(x|y) p_0(y) q_t(x|y) \, dxdy \\
&= \mathbb{E}_{X_0 \sim P_0} \mathbb{E}_{X_t|X_0 \sim Q_t} \langle S_\theta, \nabla \log q_t(x_t|x_0) \rangle .
\end{aligned}$$

Then

$$\begin{aligned}
(13) &= \mathbb{E}_{X_0 \sim P_0} \mathbb{E}_{X_t|X_0 \sim Q_t} \left[ \|S_\theta\|^2 - 2\langle S_\theta, \nabla \log q_t(x_t|x_0) \rangle \right] + \mathbb{E}_{X_t} \|\nabla \log p_t\|^2 \\
&= \underbrace{\mathbb{E}_{X_0} \mathbb{E}_{X_t|X_0} \|S_\theta - \nabla \log q_t\|^2}_{(\Delta)} + C,
\end{aligned}$$

where $C = \mathbb{E}_{X_t} \|\nabla \log p_t\|^2 - \mathbb{E}_{X_0} \mathbb{E}_{X_t|X_0} \|\nabla \log q_t(x_t|x_0)\|^2$.

Moreover, $X_t|X_0 \sim \mathcal{N}(e^{-\mu_t} X_0, \bar\sigma_t^2 I)$, and its density function is

$$q_t(x|y) = (2\pi\bar\sigma_t^2)^{-d/2} \exp\left( -\frac{\|x - e^{-\mu_t}y\|^2}{2\bar\sigma_t^2} \right).$$

Then

$$\begin{aligned}
(\Delta) &= \mathbb{E}_{X_0} \mathbb{E}_{X_t|X_0} \|S_\theta - \nabla \log q_t\|^2 \\
&= \mathbb{E}_{X_0} \mathbb{E}_{X_t|X_0} \left\| S_\theta - \nabla_x \left( -\frac{\|X_t - e^{-\mu_t} X_0\|^2}{2\bar\sigma_t^2} \right) \right\|^2 \\
&= \mathbb{E}_{X_0} \mathbb{E}_{X_t|X_0} \left\| S_\theta + \frac{X_t - e^{-\mu_t} X_0}{\bar\sigma_t^2} \right\|^2 \\
&= \mathbb{E}_{X_0} \mathbb{E}_{\epsilon_t} \left\| S_\theta + \frac{\epsilon_t}{\bar\sigma_t^2} \right\|^2 .
\end{aligned}$$

Let $\xi = \frac{\epsilon_t}{\bar{\sigma}_t} \sim \mathcal{N}(0, I)$. Then

$$(\Delta) = \mathbb{E}_{X_0} \mathbb{E}_\xi \bar{\sigma}_t \cdot \frac{1}{\bar{\sigma}_t^2} \| \bar{\sigma}_t S_\theta + \xi \|^2$$

$$= \frac{1}{\bar{\sigma}_t} \mathbb{E}_{X_0} \mathbb{E}_\xi \| \bar{\sigma}_t S_\theta + \xi \|^2$$

## D    Proofs for training

In this section, we will prove Theorem 1.

Before introducing the concrete proof, we first redefine the deep fully connected feedforward network

$$r_{ij,0} = W_0 X_{ij}, q_{ij,0} = \sigma(r_{ij,0}),$$
$$r_{ij,\ell} = W_\ell q_{ij,\ell-1}, q_{ij,\ell} = \sigma(r_{ij,\ell}), \text{ for } \ell = 1, \cdots, L$$
$$S(\theta; t_j, X_{ij}) = W_{L+1} q_{ij,L}$$

where $W_0 \in \mathbb{R}^{m \times d}, W_{L+1} \in \mathbb{R}^{d \times m}$ and $W_\ell \in \mathbb{R}^{m \times m}$; $\sigma$ is the ReLU activation. We also denote $q_{ij,-1}$ to be $X_{ij}$.

We also follow the notation in Allen-Zhu et al. [1] and denote $D_{i,\ell} \in \mathbb{R}^{m \times m}$ to be a diagonal matrix and $(D_{i,\ell})_{kk} = \mathbb{1}_{(W_\ell q_{ij,\ell-1})_k > 0}$ for $k = 1, \cdots, m$. Then

$$q_{ij,\ell} = D_{ij,\ell} W_\ell q_{ij,\ell-1}$$

For the objective (7), the gradient w.r.t. to the $k$th row of $W_\ell$ for $\ell = 1, \cdots, L$ is the following

$$\nabla_{(W_\ell)_k} \bar{\mathcal{L}}_{em}(\theta) = \frac{1}{n} \sum_{i=1}^n \sum_{j=1}^N w(t_j)(t_j - t_{j-1})$$
$$[(\underbrace{W_{L+1} D_{ij,L} W_L \cdots D_{ij,\ell} W_{\ell+1}}_{R_{ij,\ell+1}})^\top (\bar{\sigma}_{t_j} W_{L+1} q_{ij,L} + \xi_{ij})]_k q_{ij,\ell-1} \mathbb{1}_{(W_\ell q_{ij,\ell-1})_k > 0}$$

Throughout the proof, we use both $\bar{\mathcal{L}}_{em}(\theta)$ and $\bar{\mathcal{L}}_{em}(W)$ to represent the same value of the loss function, where $W = (W_1, \cdots, W_L)$, and we let $a = b = \frac{1}{2}$.

Next, we will prove Theorem 1.

*Proof of Theorem 1.* First by Lemma 4,

$$\bar{\mathcal{L}}_{em}(W^{(0)}) = \mathcal{O}(d^{2a} \sum_j w(t_j)(t_j - t_{j-1})/\bar{\sigma}_{t_j})$$

Also, $\| \nabla \bar{\mathcal{L}}_{em}(\theta) \| \leq \sqrt{L} \max_\ell \| \nabla_{W_\ell} \bar{\mathcal{L}}_{em}(\theta) \|$. Then we have

$$\| W^{(k)} - W^{(0)} \| \leq \sum_{i=0}^{k-1} h \| \nabla \bar{\mathcal{L}}_{em}(W^{(i)}) \|$$

$$\leq \mathcal{O}(\sqrt{md^{2a-1} NL \max_j w(t_j)(t_j - t_{j-1})\bar{\sigma}_{t_j}}) hk \max_i \sqrt{\bar{\mathcal{L}}_{em}(W^{(i)})}$$

$$\leq \mathcal{O}(\sqrt{md^{2a-1} NL \max_j w(t_j)(t_j - t_{j-1})\bar{\sigma}_{t_j}} d^a) hk \sqrt{\sum_j w(t_j)(t_j - t_{j-1})/\bar{\sigma}_{t_j}} := \omega$$

Let $h = \Theta(\frac{nN}{m \min_j w(t_j)(t_j - t_{j-1})\bar{\sigma}_{t_j}})$ and $k = \mathcal{O}(d^{\frac{1-a_0}{2}} n^2 N \log(\frac{d}{\epsilon_{\text{train}}}))$, where $\epsilon_{\text{train}} > 0$ is some small constant. Then $\omega = \mathcal{O}(\log(\frac{d}{\epsilon_{\text{train}}}) \frac{d^{1-\frac{a_0}{2}} n^3 N^{5/2} L^{1/2}}{\sqrt{m}} \frac{\sqrt{\max_j w(t_j)(t_j - t_{j-1})\bar{\sigma}_{t_j} \sum_k w(t_j)(t_j - t_{j-1})/\bar{\sigma}_{t_j}}}{\min_j w(t_j)(t_j - t_{j-1})\bar{\sigma}_{t_j}})$ and by Lemma 8,

with probability at least $1 - \mathcal{O}(nN)\exp(-\Omega(d^{2a_0-1}))$,

$$
\begin{aligned}
&\bar{\mathcal{L}}_{em}(W^{(k+1)})\\
&\le \bar{\mathcal{L}}_{em}(W^{(k)}) - h\|\nabla\bar{\mathcal{L}}_{em}(W^{(k)})\|^2\\
&\quad + h\sqrt{\bar{\mathcal{L}}_{em}}\sqrt{\sum_j w(t_j)(t_j-t_{j-1})\bar{\sigma}_{t_j}}\,\mathcal{O}(\omega^{1/3}L^2\sqrt{m\log m}d^{a/2})\|\nabla\bar{\mathcal{L}}_{em}(W^{(k)})\|\\
&\quad + h^2\sqrt{\bar{\mathcal{L}}_{em}}\sqrt{\sum_j w(t_j)(t_j-t_{j-1})\bar{\sigma}_{t_j}}\,\mathcal{O}(L^2\sqrt{m}d^a)\|\nabla\bar{\mathcal{L}}_{em}(W^{(k)})\|^2\\
&\le \left(1 - hw(t_{j^*})(t_{j^*}-t_{j^*-1})\bar{\sigma}_{t_{j^*}}\cdot\Omega\left(\frac{md^{\frac{a_0-1}{2}}}{n^3N^2}\right)\right)\bar{\mathcal{L}}_{em}(W^{(k)})\\
&\quad + hC\frac{m^{5/6}d^{7/12-a_0/6}}{N^{1/6}n^{2/3}(\log m)^{1/6}\sqrt{L}}\frac{\sqrt{\sum_j w(t_j)(t_j-t_{j-1})\bar{\sigma}_{t_j}}\min_j w(t_j)(t_j-t_{j-1})\bar{\sigma}_{t_j}}{\max_j w(t_j)(t_j-t_{j-1})\bar{\sigma}_{t_j}\sum_k w(t_j)(t_j-t_{j-1})/\bar{\sigma}_{t_j}}\bar{\mathcal{L}}_{em}(W^{(k)})\\
&\le \left(1 - hw(t_{j^*})(t_{j^*}-t_{j^*-1})\bar{\sigma}_{t_{j^*}}\cdot\Omega\left(\frac{md^{\frac{a_0-1}{2}}}{n^3N^2}\right)\right)\bar{\mathcal{L}}_{em}(W^{(k)})
\end{aligned}
$$

where $C > 0$ is some constant, $a_0 \in (1/2, 1)$; the second inequality follows from Lemma 7 with

$$
\|\nabla_{W_L}\bar{\mathcal{L}}_{em}(\theta^{(k)})\|^2 = \Omega\left(\frac{md^{\frac{a_0-1}{2}}}{n^3N^2}\,w(t_{j^*})(t_{j^*}-t_{j^*-1})\bar{\sigma}_{t_{j^*}}\right)\bar{\mathcal{L}}_{em}(\theta^{(k)}),
$$

which is obtained inductively; the last inequality follows from $m = \Omega\left(d^{13/2-2a_0/3}n^{14/3}N^{11}L^3(\log m)\left(\frac{\max_j w(t_j)(t_j-t_{j-1})\bar{\sigma}_{t_j}\sum_k w(t_j)(t_j-t_{j-1})/\bar{\sigma}_{t_j}}{\min_j w(t_j)(t_j-t_{j-1})\bar{\sigma}_{t_j}\sqrt{\sum_j w(t_j)(t_j-t_{j-1})\bar{\sigma}_{t_j}}}\right)^6\right)$

$\square$

### D.1 Proof of lower bound of the gradient at the initialization

In this section, we will show the main part of the convergence analysis, which is the following lower bound of the gradient.

**Lemma 1** (Lower bound). *With probability $1 - \mathcal{O}(nN)\exp(-\Omega(d^{2a_0-1}))$, we have*

$$
\|\nabla\bar{\mathcal{L}}_{em}(\theta^{(0)})\|^2 \ge C_6\left(\frac{md^{\frac{a_0-1}{2}}}{n^3N^2}\,w(t_{j^*})(t_{j^*}-t_{j^*-1})\bar{\sigma}_{t_{j^*}}\right)\bar{\mathcal{L}}_{em}(\theta^{(0)})
$$

*where $(i^*, j^*) = \arg\max\left\|\sqrt{\frac{w(t_j)(t_j-t_{j-1})}{\bar{\sigma}_{t_j}}}(\bar{\sigma}_{t_j}W_{L+1}q_{ij,L}+\xi_{ij})\right\|$, $\frac{1}{2} < a_0 < 1$, and $C_6 > 0$ is some universal constant.*

Below is the proof sketch of Lemma 1.

*Proof sketch.* We first decompose the gradient of the $k$th row of $W_L$ $\nabla_{(W_L)_k}\bar{\mathcal{L}}_{em}(\theta) = \underbrace{\frac{1}{n}w(t_{j^*})(t_{j^*}-t_{j^*-1})(W_{L+1})^{k^\top}(\bar{\sigma}_{t_{j^*}}W_{L+1}q_{i^*j^*,L}+\xi_{i^*j^*})q_{i^*j^*,L-1}1_{(W_Lq_{i^*j^*,L-1})_k>0}}_{\nabla_1}$

$+ \underbrace{\frac{1}{n}\sum_{(i,j)\ne(i^*,j^*)}w(t_j)(t_j-t_{j-1})(W_{L+1})^{k^\top}(\bar{\sigma}_{t_j}W_{L+1}q_{ij,L}+\xi_{ij})q_{ij,L-1}1_{(W_Lq_{ij,L-1})_k>0}}_{\nabla_2}$

where $(i^*, j^*)$ indicates the sample index with the largest loss value.

Then we first fix $(q_{ij,L-1})_s = 1$, and prove that the index set of both $(q_{i^*j^*,L})_s > 0$ and $\sum_{(i,j)\ne(i^*,j^*)}w(t_j)(t_j-t_{j-1})\bar{\sigma}_{t_j}1_{(W_Lq_{ij,L-1})_k>0}(q_{ij,L})_s > 0$ is order $m$ with high probability.

Next, we conditioned on the index set we've found, then we can decouple each element of $\nabla_{(W_L)_k}\bar{\mathcal{L}}_{\text{em}}$ with high probability. We prove that with high probability

$$\angle\left(W_{L+1}\bar{\sigma}_{t_{j^*}}q_{i^*j^*,L}+\xi_{i^*j^*}, W_{L+1}\sum_{(i,j)\neq(i^*,j^*)}\alpha_{ij}q_{ij,L}+\sum_{(i,j)\neq(i^*,j^*)}\bar{\alpha}_{ij}\xi_{ij}\right)\leq\pi-cd^{\frac{a_0-1}{2}},$$

for some constant $c>0$ and $\frac{1}{2}<a_0<1$. Based on this, we show that with probability at least $1-\mathcal{O}(nN)\exp(-\Omega(d))$,

$$\mathbb{P}\left((\nabla_1)_s>0,(\nabla_2)_s>0\right)\geq cd^{\frac{a_0-1}{2}},$$

for some $c>0$. Then we prove that with probability at least $1-\exp(-\Omega(md^{\frac{a_0-1}{2}}))$

$$|\{k:(W_{L+1}^k)^\top v\geq 0,(W_{L+1}^k)^\top(u+\xi)\geq 0\}|=\Theta(md^{\frac{a_0-1}{2}}).$$

with high probability, the event $(\nabla_1)_s>0$ and $(\nabla_2)_s>0$ has probability at least of order $d^{(a_0-1)/2}$ where $a_0\in(1/2,1)$.

Now, we deal with $(q_{ij,L-1})_s$ and prove that if the above results hold for $(q_{ij,L-1})_s=1$, then there exists an index set with cardinality of order $m/(nN)$ such that $(\nabla_1)_s>0$ and $(\nabla_2)_s>0$ also hold in this index set.

In the end, combining all the steps above yields the lower bound. $\qquad\square$

Here is the complete proof.

*Proof.* The main idea of the proof of lower bound is to decouple the elements in the gradient and incorporate geometric view. We focus on $\nabla_{W_L}\bar{\mathcal{L}}_{em}(\theta)$.

**Step 1**: Rewrite $\nabla_{(W_L)_k}\bar{\mathcal{L}}_{em}(\theta)$ to be the $(i^*,j^*)$th term $g_1$ plus the rest $nN-1$ terms $g_2$.

Let $(i^*,j^*)=\arg\max\left\|\sqrt{\frac{w(t_j)(t_j-t_{j-1})}{\sigma_{t_j}}}(\bar{\sigma}_{t_j}W_{L+1}q_{ij,L}+\xi_{ij})\right\|$. Let

$$g_{ij,L}=w(t_j)(t_j-t_{j-1})(W_{L+1})^{k^\top}(\bar{\sigma}_{t_j}W_{L+1}q_{ij,L}+\xi_{ij})\,q_{ij,L-1}.$$

Then

$$\nabla_{(W_L)_k}\bar{\mathcal{L}}_{em}(\theta)$$

$$=\underbrace{\frac{1}{n}w(t_{j^*})(t_{j^*}-t_{j^*-1})(W_{L+1})^{k^\top}(\bar{\sigma}_{t_{j^*}}W_{L+1}q_{i^*j^*,L}+\xi_{i^*j^*})\,q_{i^*j^*,L-1}\mathbb{1}_{(W_Lq_{i^*j^*,L-1})_k>0}}_{\nabla_1}$$

$$+\underbrace{\frac{1}{n}\sum_{(i,j)\neq(i^*,j^*)}w(t_j)(t_j-t_{j-1})(W_{L+1})^{k^\top}(\bar{\sigma}_{t_j}W_{L+1}q_{ij,L}+\xi_{ij})\,q_{ij,L-1}\mathbb{1}_{(W_Lq_{ij,L-1})_k>0}}_{\nabla_2}$$

Also define

$$\nabla_{1,s}=\underbrace{\frac{1}{n}w(t_{j^*})(t_{j^*}-t_{j^*-1})\bar{\sigma}_{t_{j^*}}(W_{L+1})^{k^\top}W_{L+1}q_{i^*j^*,L}\,(q_{i^*j^*,L-1})_s\mathbb{1}_{(W_Lq_{i^*j^*,L-1})_k>0}}_{\nabla_{11,s}}$$

$$+\underbrace{\frac{1}{n}w(t_{j^*})(t_{j^*}-t_{j^*-1})(W_{L+1})^{k^\top}\xi_{i^*j^*}\,(q_{i^*j^*,L-1})_s\mathbb{1}_{(W_Lq_{i^*j^*,L-1})_k>0}}_{\nabla_{12,s}}$$

$$\nabla_{2,s}=\underbrace{\frac{1}{n}\sum_{(i,j)\neq(i^*,j^*)}w(t_j)(t_j-t_{j-1})\bar{\sigma}_{t_j}(W_{L+1})^{k^\top}W_{L+1}q_{ij,L}\,(q_{ij,L-1})_s\mathbb{1}_{(W_Lq_{ij,L-1})_k>0}}_{\nabla_{21,s}}$$

$$+\underbrace{\frac{1}{n}\sum_{(i,j)\neq(i^*,j^*)}w(t_j)(t_j-t_{j-1})(W_{L+1})^{k^\top}\xi_{ij}\,q_{ij,L-1}(q_{ij,L-1})_s\mathbb{1}_{(W_Lq_{ij,L-1})_k>0}}_{\nabla_{22,s}}$$

Our goal is to show that with high probability, there are at least $\mathcal{O}(\frac{md^{\frac{a_0-1}{2}}}{nN})$ number of rows $k$ such that $\nabla_{11,s} \geq 0, \nabla_{12,s} \geq 0, \nabla_{21,s} \geq 0, \nabla_{22,s} \geq 0$. Then we can lower bound $\|\nabla_{(W_L)_k}\bar{\mathcal{L}}_{em}(\theta)\|^2$ by $\|\nabla_1\|^2$, which can be eventually lower bounded by $\bar{\mathcal{L}}_{em}(\theta)$.

**Step 2**: Consider $[\nabla_{(W_L)_k}\bar{\mathcal{L}}_{em}(\theta)]_s$. For $(g_2)_s$, first take $(q_{ij,L-1})_s = 1$ for all $(i,j) \neq (i^*,j^*)$. Then we only need to consider

$$\nabla'_{2,s} = \frac{1}{n} \sum_{(i,j)\neq(i^*,j^*)} w(t_j)(t_j - t_{j-1})(W_{L+1})^{k\top}(\bar{\sigma}_{t_j}W_{L+1}q_{ij,L} + \xi_{ij})\mathbb{1}_{(W_L q_{ij,L-1})_k>0}$$

which is independent of $s$. For $\nabla_1$, since $q_{i^*,j^*,L-1} \geq 0$ which does not affect the sign of this term, we can also first take $(q_{i^*,j^*,L-1})_s = 1$ for all $s$.

**Step 3**: We focus on $\nabla_{11}$ and $\nabla_{21}$ and we would like to pick the non-zero elements in this two terms. More precisely, let

$$N_1 = \{s \mid (q_{i^*j^*,L})_s > 0, s = 1, \cdots, m\},$$

$$N_2 = \left\{s \mid \sum_{(i,j)\neq(i^*,j^*)} w(t_j)(t_j - t_{j-1})\bar{\sigma}_{t_j}\mathbb{1}_{(W_L q_{ij,L-1})_k>0}(q_{ij,L})_s > 0, s = 1, \cdots, m\right\}$$

Let $\alpha_{ij} = w(t_j)(t_j - t_{j-1})\bar{\sigma}_{t_j}\mathbb{1}_{(W_L q_{ij,L-1})_k>0} \geq 0$. Then

$$\sum_{(i,j)\neq(i^*,j^*)} w(t_j)(t_j - t_{j-1})\bar{\sigma}_{t_j}\mathbb{1}_{(W_L q_{ij,L-1})_k>0}(q_{ij,L})_s$$

$$= \sum_{(i,j)\neq(i^*,j^*)} \alpha_{ij}(q_{ij,L})_s = \sum_{(i,j)\neq(i^*,j^*)} \alpha_{ij}\sigma(W_L q_{ij,L-1})_s.$$

If

$$\sum_{(i,j)\neq(i^*,j^*)} \alpha_{ij}(W_L q_{ij,L-1})_s = (W_L)_s \sum_{(i,j)\neq(i^*,j^*)} \alpha_{ij}q_{ij,L-1} > 0,$$

then there must be at least one pair of $(i,j)$ s.t. $\alpha_{ij}(W_L q_{ij,L-1})_s = \alpha_{ij}\sigma(W_L q_{ij,L-1})_s > 0$, which implies $\sum_{(i,j)\neq(i^*,j^*)} \alpha_{ij}(q_{ij,L})_s > 0$. Therefore, it suffices to consider

$$N_1 = \{s \mid (q_{i^*j^*,L})_s = (W_L)_s q_{i^*j^*,L-1} > 0, s = 1, \cdots, m\},$$

$$N'_2 = \left\{s \mid (W_L)_s \sum_{(i,j)\neq(i^*,j^*)} \alpha_{ij}q_{ij,L-1} > 0\right\}.$$

Since $(q_{ij,L-1})_s \geq 0$, we have

$$\left\langle q_{i^*j^*,L-1}, \sum_{(i,j)\neq(i^*,j^*)} \alpha_{ij}q_{ij,L-1} \right\rangle \geq 0,$$

$$i.e., \angle\left(q_{i^*j^*,L-1}, \sum_{(i,j)\neq(i^*,j^*)} \alpha_{ij}q_{ij,L-1}\right) \leq \frac{\pi}{2}$$

By Lemma 2 and Proposition 2, we have

$$\mathbb{P}\left((W_L)_s q_{i^*j^*,L-1} > 0, (W_L)_s \sum_{(i,j)\neq(i^*,j^*)} \alpha_{ij}q_{ij,L-1} > 0\right)$$

$$= \mathbb{P}\left(\frac{(W_L)_s}{\|(W_L)_s\|} q_{i^*j^*,L-1} > 0, \frac{(W_L)_s}{\|(W_L)_s\|} \sum_{(i,j)\neq(i^*,j^*)} \alpha_{ij}q_{ij,L-1} > 0\right)$$

$$\geq \frac{1}{4}.$$

Also $(W_L)_s$'s are $i.i.d.$ multivariate Gaussian. By Chernoff bound,

$$\mathbb{P}\left(|N_1 \cap N'_2| \in (\delta_1 \frac{m}{4}, \delta_2 \frac{m}{4})\right) \leq 1 - 2e^{-\Omega(m)}$$

for some small $\delta_1 \leq \frac{1}{4}$ and $\delta_2 \leq 4$, i.e., $|N_1 \cap N_2'| = \Theta(m)$ with probability at least $1 - 2e^{-\Omega(m)}$.

**Step 4**: Next we condition on $N_1 \cap N_2'$ and consider $(W_{L+1})^{k^\top} W_{L+1} \bar{\sigma}_{t_{j^*}} q_{i^* j^*, L} + (W_{L+1})^{k^\top} \xi_{i^* j^*}$ and $(W_{L+1})^{k^\top} W_{L+1} \sum_{(i,j) \neq (i^*,j^*)} \alpha_{ij} q_{ij,L} + (W_{L+1})^{k^\top} \sum_{(i,j) \neq (i^*,j^*)} \bar{\alpha}_{ij} \xi_{ij}$, where $\bar{\alpha}_{ij} = \alpha_{ij}/\bar{\sigma}_{ij}$. We would like to prove that with high probability

$$\angle \left( W_{L+1} \bar{\sigma}_{t_{j^*}} q_{i^* j^*, L} + \xi_{i^* j^*}, W_{L+1} \sum_{(i,j) \neq (i^*,j^*)} \alpha_{ij} q_{ij,L} + \sum_{(i,j) \neq (i^*,j^*)} \bar{\alpha}_{ij} \xi_{ij} \right) \leq \pi - c d^{\frac{a_0 - 1}{2}},$$

for some constant $c > 0$ and $\frac{1}{2} < a_0 < 1$.

First, since $\xi_{ij} \sim \mathcal{N}(0, I_d)$, by Bernstein's inequality, with probability at least $1 - \exp(-\Omega(d))$, we have $\|\xi_{ij}\|^2 = \Theta(d)$. Similarly, since $(W_{L+1} q_{ij,L})_s \sim \mathcal{N}(0, \frac{2\|q_{ij,L}\|^2}{m})$, by Berstein's inequality and Lemma 4, with probability at least $1 - \exp(-\Omega(d))$, we have $\|W_{L+1} q_{ij,L}\|^2 = \Theta(d)$. By union bounds, the above holds for all $i, j$ with probability at least $1 - 2nN \exp(-\Omega(d))$.

Let $v = \frac{W_{L+1} \sum_{(i,j) \neq (i^*,j^*)} \alpha_{ij} q_{ij,L} + \sum_{(i,j) \neq (i^*,j^*)} \bar{\alpha}_{ij} \xi_{ij}}{\|W_{L+1} \sum_{(i,j) \neq (i^*,j^*)} \alpha_{ij} q_{ij,L} + \sum_{(i,j) \neq (i^*,j^*)} \bar{\alpha}_{ij} \xi_{ij}\|}$ and $u = W_{L+1} \bar{\sigma}_{t_{j^*}} q_{i^* j^*, L}$. For notational simplicity, we use $\xi$ to denote $\xi_{i^* j^*}$. Fix $v, u$ and consider the probability of event $A$

$$A = \{ v^\top (u + \xi) \leq -\sqrt{1 - c_0 d^{a_0 - 1}} \|u + \xi\| \}$$

for some $c_0 > 0$ and $\frac{1}{2} < a_0 < 1$.

Then consider the following event that has larger probability than $A$

$$(v^\top (u + \xi))^2 \geq (1 - c_0 d^{a_0 - 1}) \|u + \xi\|^2 \tag{14}$$
$$\iff (v^\top u)^2 - (1 - c_0 d^{a_0 - 1}) \|u\|^2 + 2(v^\top u \, v - (1 - c_0 d^{a_0 - 1}) u)^\top \xi + (v^\top \xi)^2 \geq (1 - c_0 d^{a_0 - 1}) \|\xi\|^2 \tag{15}$$

Since $(v^\top u)^2 \leq \|u\|^2$ where the equality holds when $v = \frac{u}{\|u\|}$, we have

$$\text{LHS} \leq c_0 d^{a_0 - 1} \|u\|^2 + 2(v^\top u \, v - (1 - c_0 d^{a_0 - 1}) u)^\top \xi + (v^\top \xi)^2$$

Also, since $\|u\|^2 = \mathcal{O}(d)$ with probability at least $1 - 2nN \exp(-\Omega(d))$, we have

$$\mathbb{P} \left( |2(v^\top u \, v - (1 - c_0 d^{a_0 - 1}) u)^\top \xi| \geq d^{a_0} \right) \leq 2 \exp \left( -c \frac{d^{2a_0}}{\|u\|^2} \right) = 2 \exp(-\Omega(d^{2a_0 - 1}))$$

for some constant $c > 0$.

Therefore, with probability at least $1 - \mathcal{O}(nN) \exp(-\Omega(d^{2a_0 - 1}))$

$$\text{LHS of (15)} \leq c d^{a_0} + (v^\top \xi)^2$$

for some constant $c > 0$.

Then

$$\mathbb{P}(v^\top (u + \xi) \leq -\sqrt{1 - c_0 d^{a_0 - 1}} \|u + \xi\|)$$
$$\leq \mathbb{P}((v^\top (u + \xi))^2 \geq (1 - c_0 d^{a_0 - 1}) \|u + \xi\|^2)$$
$$\leq \mathbb{P}((v^\top \xi)^2 \geq (1 - c' d^{a_0 - 1}) \|\xi\|^2)$$
$$= \mathbb{P} \left( v^\top \frac{\xi}{\|\xi\|} \geq \sqrt{(1 - c' d^{a_0 - 1})} \right) + \mathbb{P} \left( -v^\top \frac{\xi}{\|\xi\|} \geq \sqrt{(1 - c' d^{a_0 - 1})} \right)$$
$$= \mathbb{P} \left( \angle \left( v, \frac{\xi}{\|\xi\|} \right) \geq \arccos(-\sqrt{(1 - c' d^{a_0 - 1})}) \right) + \mathbb{P} \left( \angle \left( -v, \frac{\xi}{\|\xi\|} \right) \geq \arccos(-\sqrt{(1 - c' d^{a_0 - 1})}) \right)$$
$$= 2\mathbb{P} \left( \angle \left( v, \frac{\xi}{\|\xi\|} \right) \geq \pi - c'' d^{\frac{a_0 - 1}{2}} \right)$$
$$= \frac{C}{(d^{\frac{1 - a_0}{2}})^{d - 1} \sqrt{d}}$$

where the second equality follows from Lemma 2; the third equality follows from series expansion; the forth equality follows from (16); $c', c'', C > 0$ are some constants.

Thus with probability at least $1 - \frac{C}{(d^{\frac{1-a_0}{2}})^{d-1}\sqrt{d}}$,

$$v^\top(u + \xi) \geq -\sqrt{1 - c_0 d^{a_0-1}}\|u + \xi\|$$

$$i.e. \angle(v, u + \xi) \leq \pi - cd^{\frac{a_0-1}{2}}$$

for some $c > 0$.

Then by Lemma 2, with probability at least $1 - \mathcal{O}(nN)\exp(-\Omega(d))$,

$$\mathbb{P}\left((W_{L+1}^k)^\top v \geq 0, (W_{L+1}^k)^\top(u + \xi) \geq 0\right) \geq cd^{\frac{a_0-1}{2}},$$

for some $c > 0$.

Since $(W_{L+1}^k)$ are iid Guassian vectors for $k = 1, \cdots, m$, by Chernoff bound on Bernoulli variable $\mathbb{1}_{\{(W_{L+1}^k)^\top v \geq 0, (W_{L+1}^k)^\top(u+\xi) \geq 0\}}$, we have, with probability at least $1 - \exp(-\Omega(md^{\frac{a_0-1}{2}}))$

$$|\{k : (W_{L+1}^k)^\top v \geq 0, (W_{L+1}^k)^\top(u + \xi) \geq 0\}| = \Theta(md^{\frac{a_0-1}{2}}).$$

**Step 5:** Combining the above 4 steps, we would like to obtain the lower bound of the gradient.

For each $k$, consider $(q_{ij,L-1})_s$ for $(i, j) \neq (i^*, j^*)$ and denote $q_s = \{(q_{ij,L-1})_s\}_{(i,j)\neq(i^*,j^*)} = \{\sigma((W_{L-1})_s q_{ij,L-2})\}_{(i,j)\neq(i^*,j^*)}$. Let $\bar{q}_s = \{(W_{L-1})_s q_{ij,L-2}\}_{(i,j)\neq(i^*,j^*)}$ and $\bar{q}_s \sim \mathcal{N}(0, QQ^\top)$, where each row of $Q$ is $q_{ij,L-2}^\top$ for $(i, j) \neq (i^*, j^*)$. Thus, $q_s$ is $\bar{q}_s$ projected to the nonnegative orthant.

Let $\mathbf{1} = (1, 1, \cdots, 1) \in \mathbb{R}^{nN-1}$. Therefore, if $\langle \beta_k, \mathbf{1} \rangle \geq 0$ for some $\beta_k \in \mathbb{R}^{nN-1}$, then at least half of the nonnegative orthant is contained in $\{v \in \mathbb{R}^{nN-1} : \langle \beta_k, v \rangle \geq 0\}$, i.e., there exists a constant $c_k > 0$, s.t.

$$\mathbb{P}(\langle \beta_k, q_s \rangle \geq 0) \geq c_k \geq \min_{k=1,\cdots,m} c_k > 0, \text{ for all } s = 1, \cdots, m$$

Then since $\beta_k \in \mathbb{R}^{nN-1}$ for $k = 1, \cdots, m$ and $nN \ll m$, there exists a set of indices $\mathcal{K} \subseteq \{1, \cdots, m\}$ with $|\mathcal{K}| = \Theta(\frac{m}{nN})$ and a set of indices $\mathcal{S} \subseteq \{1, \cdots, m\}$ with $|\mathcal{S}| = \Theta(m)$, s.t., $\langle \beta_k, q_s \rangle \geq 0$, for $k \in \mathcal{K}, s \in \mathcal{S}$.

Let $q_{ij,\ell}^{\mathcal{K}} = (q_{ij,\ell})_{k \in \mathcal{K}}$. Then by Bernstein's inequality, we can also obtain that $\|q_{ij,\ell}^{\mathcal{K}}\|^2 = \Theta(d)$ with probability at least $1 - nN\exp(-\Omega(d))$.

Combine all of the above and apply the Claim 9.5 in Allen-Zhu et al. [1], we obtain, with probability at least $1 - \mathcal{O}(nN)\exp(-\Omega(d^{2a_0-1}))$,

$$\|\nabla_{W_L}\bar{\mathcal{L}}_{em}(\theta^{(0)})\|_F^2 \geq \frac{1}{n^2}C_6 w(t_{j^*})^2(t_{j^*} - t_{j^*-1})^2 \frac{1}{d}\|\bar{\sigma}_{t_{j^*}}W_{L+1}q_{i^*j^*,L} + \xi_{i^*j^*}\|^2 \|q_{i^*j^*,L-1}^{\mathcal{K}}\|^2 \frac{1}{nN}md^{\frac{a_0-1}{2}}$$

$$\geq C_6 md^{\frac{a_0-1}{2}}w(t_{j^*})(t_{j^*} - t_{j^*-1})\bar{\sigma}_{t_{j^*}}\frac{1}{n^3 N^2}\bar{\mathcal{L}}_{em}(\theta^{(0)}),$$

where $C_6 > 0$ is some universal constant, $\frac{1}{2} < a_0 < 1$, and the second inequality follows from the definition of $i^*, j^*$.

$\square$

### D.1.1 Geometric ideas used in the proof

**Proposition 2.** *Consider* $w \sim \mathcal{N}(0, I)$, *where* $w \in \mathbb{R}^n$. *Then* $\|w\|$ *and* $\frac{w}{\|w\|}$ *are independent random variables and* $\frac{w}{\|w\|} \sim Unif(\mathbb{S}^{n-1})$.

**Lemma 2.** *Let* $w \sim Unif(\mathbb{S}^{n-1})$, *where* $\mathbb{S}^{n-1} = \{x \in \mathbb{R}^n | \|x\| = 1\}$. *Then for two vectors* $v_1, v_2 \in \mathbb{R}^n$,

$$\mathbb{P}(w^\top v_1 \geq 0, w^\top v_2 \geq 0) = \frac{\pi - \angle(v_1, v_2)}{2\pi}.$$

*Proof.* Since $w \sim Unif(\mathbb{S}^{n-1})$, we only need to consider the area of the event. It is obvious that the set $\{w \in \mathbb{S}^{n-1}|w^\top v_i\}$ is a semi-hypersphere. Therefore, we only need to consider the intersection of two semi-hypersphere, i.e.,

$$\mathbb{P}(w^\top v_1 \geq 0, w^\top v_2 \geq 0) = \frac{\text{area of } \{w \in \mathbb{S}^{n-1}|w^\top v_1 \geq 0\} \cap \text{ area of } \{w \in \mathbb{S}^{n-1}|w^\top v_2 \geq 0\}}{\text{area of the hypersphere}}$$

$$= \frac{\pi - \angle(v_1, v_2)}{2\pi}.$$

$\square$

Next we follow the notations and definitions in Lee and Kim [33]. Consider the unit hypersphere in $\mathbb{R}^d$, $\mathbb{S}^{d-1} = \{x \in \mathbb{R}^d \mid \|x\| = 1\}$. The area of $\mathbb{S}^{d-1}$ is

$$A_d(1) = \frac{2\pi^{d/2}}{\Gamma(d/2)}.$$

**Lemma 3.** *Fix $\xi_1 \in \mathbb{S}^{d-1}$ and let $\xi_2 \sim Unif(\mathbb{S}^{d-1})$, where $\mathbb{S}^{d-1} = \{x \in \mathbb{R}^d\|x\| = 1\}$. Then with probability at least $1 - \exp(-\Omega(d))$, we have $\angle(\xi_1, \xi_2) \leq \frac{3\pi}{4}$.*

*Proof.* For any fixed $\xi_1$, all the $\xi_2$'s that satisfy $\angle(\xi_1, \xi_2) \geq \pi - \theta$ are on a hyperspherical cap. By Lee and Kim [33], the area of the hypersperical cap is

$$A_d^\theta(1) = \frac{1}{2}A_d(1)I_{\sin^2\theta}\left(\frac{d-1}{2}, \frac{1}{2}\right).$$

Then

$$\mathbb{P}(\angle(\xi_1, \xi_2) \geq \pi - \theta) = \frac{A_d^\theta(1)}{A_d(1)} = \frac{1}{2}I_{\sin^2\theta}\left(\frac{d-1}{2}, \frac{1}{2}\right) \propto \frac{1}{2}\frac{\theta^{d-1}}{\sqrt{\pi}\sqrt{\frac{d-1}{2}}}. \tag{16}$$

Let $\theta = \frac{\pi}{4} < 1$. Then with probability at least $1 - \exp(-\Omega(d))$, we have $\angle(\xi_1, \xi_2) \leq \frac{3\pi}{4}$. $\square$

### D.2 Proofs related to random initialization

Consider $W_i = W_i^{(0)}$ in this section.

**Lemma 4.** *If $\epsilon \in (0,1)$, with probability at least $1 - \mathcal{O}(nN)e^{-\Omega(\min(\epsilon^2 d^{4b-1}, \epsilon d^{2b}))}$, $\|X_{ij}\|^2 \in [\|e^{-\mu_{t_j}}x_i\|^2 + \bar{\sigma}_{t_j}^2 d - \epsilon\bar{\sigma}_{t_j}^2 d^{2b}, \|e^{-\mu_{t_j}}x_i\|^2 + \bar{\sigma}_{t_j}^2 d + \epsilon\bar{\sigma}_{t_j}^2 d^{2b}]$ for all $i = 1, \cdots, n$ and $j = 0, \cdots, N-1$. Moreover, with probability at least $1 - \mathcal{O}(L)e^{-\Omega(m\epsilon^2/L)}$ over the randomness of $W_s$ for $s = 0, \cdots, L$, we have $\|q_{ij,\ell}\| \in [\|X_{ij}\|(1-\epsilon), \|X_{ij}\|(1+\epsilon)]$ for fixed $i, j$. Therefore, with probability at least $1 - \mathcal{O}(nNL)e^{-\Omega(\min(m\epsilon^2/L, \epsilon^2 d^{4b-1}, \epsilon d^{2b}))}$, we have $\Omega(d^b) = \|q_{ij,\ell}\| = \mathcal{O}(d^a)$.*

*Proof.* Consider $\frac{1}{\sigma_{t_j}}X_{ij} = \frac{e^{-\mu_{t_j}}}{\bar{\sigma}_{t_j}}x_i + \xi_{ij}$. Since $\xi_{ij} \sim \mathcal{N}(0, I)$, $\|\frac{1}{\sigma_{t_j}}X_{ij}\|^2$ follows from the noncentral $\chi^2$ distribution and $\mathbb{E}\|\frac{1}{\sigma_{t_j}}X_{ij}\|^2 = d + \|\frac{e^{-\mu_{t_j}}}{\bar{\sigma}_{t_j}}x_i\|^2$ (this includes the time variable at the $d$th dimension). By Berstein inequality,

$$\mathbb{P}\left(\left|\left\|\frac{1}{\bar{\sigma}_{t_j}}X_{ij}\right\|^2 - \mathbb{E}\left\|\frac{1}{\bar{\sigma}_{t_j}}X_{ij}\right\|^2\right| \geq t\right) \leq 2\exp\left(-c\min\left(\frac{t^2}{d}, t\right)\right)$$

$$i.e., \mathbb{P}\left(\left|\|e^{-\mu_{t_j}}x_i + \bar{\sigma}_{t_j}\xi_{ij}\|^2 - (\bar{\sigma}_{t_j}^2 d + \|e^{-\mu_{t_j}}x_i\|^2)\right| \geq \bar{\sigma}_{t_j}^2 t\right) \leq 2\exp\left(-c\min\left(\frac{t^2}{d}, t\right)\right)$$

Therefore, with probability at least $1 - \mathcal{O}(nN)e^{-\Omega(\min(\epsilon^2 d^{4b-1}, \epsilon d^{2b}))}$, $\|X_{ij}\|^2 \in [\|e^{-\mu_{t_j}}x_i\|^2 + \bar{\sigma}_{t_j}^2 d - \epsilon\bar{\sigma}_{t_j}^2 d^{2b}, \|e^{-\mu_{t_j}}x_i\|^2 + \bar{\sigma}_{t_j}^2 d + \epsilon\bar{\sigma}_{t_j}^2 d^{2b}]$ for all $i = 1, \cdots, n$ and $j = 0, \cdots, N-1$, where $\epsilon \in (0,1)$. The second part of the Lemma follows the similar proof in Lemma 7.1 of Allen-Zhu et al. [1]. The last part follows from union bound and Assumption 1. $\square$

**Lemma 5** (Upper bound). *Under the random initialization of $W_i$ for $i = 0, \cdots, L$, with probability at least $1 - \mathcal{O}(nNL)e^{-\Omega(\min(m\epsilon^2/L, \epsilon^2 d^{4b-1}, \epsilon d^{2b}))}$, we have*

$$\|\nabla_{W_\ell}\bar{\mathcal{L}}_{em}(\theta^{(0)})\|^2 = \mathcal{O}\left(md^{2a-1}N \max_j w(t_j)(t_j - t_{j-1})\bar{\sigma}_{t_j}\right)\bar{\mathcal{L}}_{em}(\theta^{(0)}).$$

*Proof.* For any $\ell = 1, \cdot, L$, we have

$$\|\nabla_{W_\ell}\bar{\mathcal{L}}_{em}(\theta)\|_F^2$$

$$= \sum_{k=1}^m \|\nabla_{(W_\ell)_k}\bar{\mathcal{L}}_{em}(\theta)\|^2$$

$$= \sum_{k=1}^m \left\|\frac{1}{n}\sum_{i=1}^n\sum_{j=1}^N w(t_j)(t_j - t_{j-1})\right.$$

$$\left. \times [(W_{L+1}D_{ij,L}W_L \cdots D_{ij,\ell}W_{\ell+1})^\top(\bar{\sigma}_{t_j}W_{L+1}q_{ij,L} + \xi_{ij})]_k\, q_{ij,\ell-1}\, \mathbb{1}_{(W_\ell q_{ij,\ell-1})_k > 0}\right\|^2$$

$$\leq \frac{N}{n}\sum_{i=1}^n\sum_{j=1}^N w(t_j)^2(t_j - t_{j-1})^2 \sum_{k=1}^m \|(W_{L+1}D_{ij,L}W_L \cdots D_{ij,\ell}W_{\ell+1})_k^\top(\bar{\sigma}_{t_j}W_{L+1}q_{ij,L} + \xi_{ij})\|^2 \cdot \|q_{ij,\ell-1}\|^2$$

$$\leq C_7 d^{2a}\frac{m}{d}\frac{N}{n}\sum_{i=1}^n\sum_{j=1}^N w(t_j)^2(t_j - t_{j-1})^2 \cdot \|\bar{\sigma}_{t_j}W_{L+1}q_{ij,L} + \xi_{ij}\|^2$$

$$\leq C_7 d^{2a}\frac{mN}{d}\max_j w(t_j)(t_j - t_{j-1})\bar{\sigma}_{t_j}\bar{\mathcal{L}}_{em}(\theta)$$

where the first inequality follows from Young's inequality; the second inequality follows from Lemma 4 and Lemma 7.4 in Allen-Zhu et al. [1]; $C_7 > 0$ □

### D.3 Proofs related to perturbation

Consider $W_i^{\text{per}} = W_i^{(0)} + W_i'$ for $i = 1, \cdots, L$ in this section. We follow the same idea in Allen-Zhu et al. [1] to consider the network value of perturbed weights at each layer. We use the superscript "per" to denotes the perturbed version, i.e.,

$$r_{ij,0}^{\text{per}} = W_0 X_{ij}, q_{ij,0}^{\text{per}} = \sigma(r_{ij,0}^{\text{per}}),$$
$$r_{ij,\ell}^{\text{per}} = W_\ell^{\text{per}} q_{ij,\ell-1}^{\text{per}}, q_{ij,\ell}^{\text{per}} = \sigma(r_{ij,\ell}^{\text{per}}), \text{ for } \ell = 1, \cdots, L$$
$$S(\theta^{\text{per}}; t_j, X_{ij}) = W_{L+1}q_{ij,L}^{\text{per}}$$

We also similarly define the diagonal matrix $D_{ij,\ell}^{\text{per}}$ for the above network.

The following Lemma measures the perturbation of each layer. The lemma differs from Lemma 8.2 in Allen-Zhu et al. [1] by a scale of $d^a$. For sake of completeness, we state it in the following and the proof can be similarly obtained.

**Lemma 6.** *Let $\omega \leq \frac{1}{C_7 L^{9/2}(\log m)^3 d^a}$ for some large $C > 1$. With probability at least $1 - \exp(-\Omega(d^a m\omega^{2/3}L))$, for any $\Delta W$ s.t. $\|\Delta W\| \leq \omega$, we have*

1. *$r_{ij,\ell}^{\text{per}} - r_{ij,\ell}$ can be decomposed to two part $r_{ij,\ell}^{\text{per}} - r_{ij,\ell} = r_{ij,\ell,1}' + r_{ij,\ell,2}'$, where $\|r_{ij,\ell,1}'\| = \mathcal{O}(\omega L^{3/2}d^a)$ and $\|r_{ij,\ell,2}'\|_\infty = \mathcal{O}(\omega L^{5/2}\sqrt{\log m}d^a m^{-1/2})$.*

2. *$\|D_{ij,\ell}^{\text{per}} - D_{ij,\ell}\|_0 = \mathcal{O}(m\omega^{2/3}L)$ and $\|(D_{ij,\ell}^{\text{per}} - D_{ij,\ell})r_{ij,\ell}^{\text{per}}\| = \mathcal{O}(\omega L^{3/2}d^a)$.*

3. *$\|r_{ij,\ell}^{\text{per}} - r_{ij,\ell}\|$ and $\|q_{ij,\ell}^{\text{per}} - q_{ij,\ell}\|$ are $\mathcal{O}(\omega L^{5/2}\sqrt{\log m}d^a)$.*

### D.4 Proofs related to the evolution of the algorithm

**Lemma 7** (Upper and lower bounds of gradient after perturbation). *Let*

$$\omega = \mathcal{O}\left(\frac{\bar{\mathcal{L}}_{em}^*}{L^9(\log m)^2 n^3 N^3 d^{\frac{1-a_0}{2}}} \cdot \frac{\min_j w(t_j)(t_j - t_{j-1})\bar{\sigma}_{t_j}}{\max\{\max_j w(t_j)(t_j - t_{j-1})\bar{\sigma}_{t_j}, \sum_j(w(t_j)(t_j - t_{j-1})\bar{\sigma}_{t_j})^2\}}\right).$$

*Consider* $\theta^{\mathrm{per}}$ *s.t.* $\|\theta^{\mathrm{per}} - \theta\| \leq \omega$, *where* $\theta$ *follows from the Gaussian initialization. Then with probability at least* $1 - \mathcal{O}(nN)e^{-\Omega(d^{2a_0-1})}$,

$$\|\nabla_{W_\ell}\bar{\mathcal{L}}_{em}(\theta^{\mathrm{per}})\|^2 = \mathcal{O}\left(md^{2a-1}N\max_j w(t_j)(t_j - t_{j-1})\bar{\sigma}_{t_j}\right)\bar{\mathcal{L}}_{em}(\theta^{\mathrm{per}}),$$

$$\|\nabla_{W_L}\bar{\mathcal{L}}_{em}(\theta^{\mathrm{per}})\|^2 = \Omega\left(\frac{md^{\frac{a_0-1}{2}}}{n^3N^2}w(t_{j^*})(t_{j^*} - t_{j^*-1})\bar{\sigma}_{t_{j^*}}\right)\min\{\bar{\mathcal{L}}_{em}(\theta), \bar{\mathcal{L}}_{em}(\theta^{\mathrm{per}})\},$$

*for* $\ell = 1, \cdots, L$.

*Proof.* Consider the following terms

$$\nabla_{(W_\ell)_k}\bar{\mathcal{L}}_{em}(\theta) = \frac{1}{n}\sum_{i=1}^{n}\sum_{j=1}^{N} w(t_j)(t_j - t_{j-1})$$
$$\times [(W_{L+1}D_{ij,L}W_L\cdots D_{ij,\ell}W_{\ell+1})^\top(\bar{\sigma}_{t_j}W_{L+1}q_{ij,L} + \xi_{ij})]_k\, q_{ij,\ell-1}\, \mathbb{1}_{(W_\ell q_{ij,\ell-1})_k>0} \quad (17)$$

$$\nabla^{\mathrm{per}}_{(W_\ell)_k}\bar{\mathcal{L}}_{em}(\theta) = \frac{1}{n}\sum_{i=1}^{n}\sum_{j=1}^{N} w(t_j)(t_j - t_{j-1})$$
$$\times [(W_{L+1}D_{ij,L}W_L\cdots D_{ij,\ell}W_{\ell+1})^\top(\bar{\sigma}_{t_j}W_{L+1}q^{\mathrm{per}}_{ij,L} + \xi_{ij})]_k\, q_{ij,\ell-1}\, \mathbb{1}_{(W_\ell q_{ij,\ell-1})_k>0}, \quad (18)$$

$$\nabla_{(W_\ell)_k}\bar{\mathcal{L}}_{em}(\theta^{\mathrm{per}}) = \frac{1}{n}\sum_{i=1}^{n}\sum_{j=1}^{N} w(t_j)(t_j - t_{j-1})$$
$$\times [(W_{L+1}D^{\mathrm{per}}_{ij,L}W^{\mathrm{per}}_L\cdots D^{\mathrm{per}}_{ij,\ell}W^{\mathrm{per}}_{\ell+1})^\top(\bar{\sigma}_{t_j}W_{L+1}q^{\mathrm{per}}_{ij,L} + \xi_{ij})]_k\, q^{\mathrm{per}}_{ij,\ell-1}\, \mathbb{1}_{(W^{\mathrm{per}}_\ell q^{\mathrm{per}}_{ij,\ell-1})_k>0} \quad (19)$$

Then

$$\|\nabla_{W_\ell}\bar{\mathcal{L}}_{em}(\theta) - \nabla^{\mathrm{per}}_{W_\ell}\bar{\mathcal{L}}_{em}(\theta)\|_F^2$$
$$= \sum_{k=1}^{m}\|\nabla_{(W_\ell)_k}\bar{\mathcal{L}}_{em}(\theta) - \nabla^{\mathrm{per}}_{(W_\ell)_k}\bar{\mathcal{L}}_{em}(\theta)\|^2$$
$$\leq \frac{N}{n}\sum_{i=1}^{n}\sum_{j=1}^{N} w(t_j)^2(t_j - t_{j-1})^2\sum_{k=1}^{m}\|(W_{L+1}D_{ij,L}W_L\cdots D_{ij,\ell}W_{\ell+1})_k^\top(\bar{\sigma}_{t_j}W_{L+1}(q_{ij,L} - q^{\mathrm{per}}_{ij,L}))\|^2 \cdot \|q_{ij,\ell-1}\|^2$$
$$\leq C_8 d^{2a}\frac{m}{d}\frac{N}{n}\sum_{i=1}^{n}\sum_{j=1}^{N} w(t_j)^2(t_j - t_{j-1})^2 \cdot \|\bar{\sigma}_{t_j}W_{L+1}(q_{ij,L} - q^{\mathrm{per}}_{ij,L})\|^2$$
$$\leq C_8' d^{2a}\frac{m}{d}N\sum_{j=1}^{N} w(t_j)^2(t_j - t_{j-1})^2(\omega L^{5/2}\sqrt{\log m}d^a)^2$$
$$\leq \tilde{C}_8\left(\frac{md^{\frac{a_0-1}{2}}}{n^3N^2}w(t_{j^*})(t_{j^*} - t_{j^*-1})\bar{\sigma}_{t_{j^*}}\right)\bar{\mathcal{L}}^*_{em}$$

where the first two inequalities follow the same as the proof of Lemma 5; the third inequality follows from Lemma 6; the last inequality follows from the definition of $\omega$.

Also, we have

$$\|\nabla^{\mathrm{per}}_{(W_\ell)_k}\bar{\mathcal{L}}_{em}(\theta) - \nabla_{(W_\ell)_k}\bar{\mathcal{L}}_{em}(\theta^{\mathrm{per}})\|$$

$$\leq \left\| \frac{1}{n}\sum_{i=1}^{n}\sum_{j=1}^{N}w(t_j)(t_j - t_{j-1}) \right.$$

$$\times [(W_{L+1}D^{\mathrm{per}}_{ij,L}W^{\mathrm{per}}_L\cdots D^{\mathrm{per}}_{ij,\ell}W^{\mathrm{per}}_{\ell+1} - W_{L+1}D_{ij,L}W_L\cdots D_{ij,\ell}W_{\ell+1})^\top (\bar{\sigma}_{t_j}W_{L+1}q^{\mathrm{per}}_{ij,L} + \xi_{ij})]_k$$

$$\left. \times q^{\mathrm{per}}_{ij,\ell-1}\,\mathbb{1}_{(W^{\mathrm{per}}_\ell q^{\mathrm{per}}_{ij,\ell-1})_k>0} \right\|$$

$$+ \left\| \frac{1}{n}\sum_{i=1}^{n}\sum_{j=1}^{N}w(t_j)(t_j - t_{j-1})[(W_{L+1}D_{ij,L}W_L\cdots D_{ij,\ell}W_{\ell+1})^\top (\bar{\sigma}_{t_j}W_{L+1}q^{\mathrm{per}}_{ij,L} + \xi_{ij})]_k \right.$$

$$\left. \times \left( q_{ij,\ell-1}\,\mathbb{1}_{(W_\ell q_{ij,\ell-1})_k>0} - q^{\mathrm{per}}_{ij,\ell-1}\,\mathbb{1}_{(W^{\mathrm{per}}_\ell q^{\mathrm{per}}_{ij,\ell-1})_k>0} \right) \right\|$$

Then

$$\|\nabla^{\mathrm{per}}_{W_\ell}\bar{\mathcal{L}}_{em}(\theta) - \nabla_{W_\ell}\bar{\mathcal{L}}_{em}(\theta^{\mathrm{per}})\|_F^2$$

$$\leq 2\frac{N}{n}\sum_{i=1}^{n}\sum_{j=1}^{N}w(t_j)^2(t_j - t_{j-1})^2 \cdot \|q^{\mathrm{per}}_{ij,\ell-1}\|^2$$

$$\times \sum_{k=1}^{m}\|(W_{L+1}D^{\mathrm{per}}_{ij,L}W^{\mathrm{per}}_L\cdots D^{\mathrm{per}}_{ij,\ell}W^{\mathrm{per}}_{\ell+1} - W_{L+1}D_{ij,L}W_L\cdots D_{ij,\ell}W_{\ell+1})^\top_k (\bar{\sigma}_{t_j}W_{L+1}q^{\mathrm{per}}_{ij,L} + \xi_{ij})\|^2$$

$$+ 2\frac{N}{n}\sum_{i=1}^{n}\sum_{j=1}^{N}w(t_j)^2(t_j - t_{j-1})^2 \cdot \|q_{ij,\ell-1}\,\mathbb{1}_{(W_\ell q_{ij,\ell-1})_k>0} - q^{\mathrm{per}}_{ij,\ell-1}\,\mathbb{1}_{(W^{\mathrm{per}}_\ell q^{\mathrm{per}}_{ij,\ell-1})_k>0}\|^2$$

$$\times \sum_{k=1}^{m}\|(W_{L+1}D_{ij,L}W_L\cdots D_{ij,\ell}W_{\ell+1})^\top_k (\bar{\sigma}_{t_j}W_{L+1}q^{\mathrm{per}}_{ij,L} + \xi_{ij})\|^2$$

$$\leq C_9(\omega^2 L^5\log md^{2a})(L^{3/2}\log m\, m^{1/2})\frac{m}{d}\frac{N}{n}\sum_{i=1}^{n}\sum_{j=1}^{N}w(t_j)^2(t_j - t_{j-1})^2 \cdot \|\bar{\sigma}_{t_j}W_{L+1}q^{\mathrm{per}}_{ij,L} + \xi_{ij}\|^2$$

$$+ C_9'(\omega^2 L^5\log md^{2a})\frac{m}{d}\frac{N}{n}\sum_{i=1}^{n}\sum_{j=1}^{N}w(t_j)^2(t_j - t_{j-1})^2 \cdot \|\bar{\sigma}_{t_j}W_{L+1}q^{\mathrm{per}}_{ij,L} + \xi_{ij}\|^2$$

$$\leq \tilde{C}_9\left( \frac{md^{\frac{a_0-1}{2}}}{n^3 N^2}w(t_{j^*})(t_{j^*} - t_{j^*-1})\bar{\sigma}_{t_{j^*}} \right)\bar{\mathcal{L}}_{em}(\theta^{\mathrm{per}})$$

where the first inequality follows from Young's inequality and the above decomposition; the second inequality follows from Lemma 7.4, 8.7 in Allen-Zhu et al. [1] (with $s = \mathcal{O}(d^a m\omega^{2/3}L)$) and Lemma 6; the last inequality follows from the definition of $\omega$.

For upper bound, we only need to consider $\nabla^{\mathrm{per}}_{(W_\ell)_k}\bar{\mathcal{L}}_{em}(\theta)$ and $\nabla_{(W_\ell)_k}\bar{\mathcal{L}}_{em}(\theta^{\mathrm{per}})$. By similar argument as Lemma 5, with probability at least $1 - \mathcal{O}(nNL)e^{-\Omega(\min(m\epsilon^2/L, \epsilon^2 d^{4b-1}, \epsilon d^{2b}))}$, we have

$$\|\nabla^{\mathrm{per}}_{(W_\ell)_k}\bar{\mathcal{L}}_{em}(\theta)\|^2 = \mathcal{O}\left( md^{2a-1}N\max_j w(t_j)(t_j - t_{j-1})\bar{\sigma}_{t_j} \right)\bar{\mathcal{L}}_{em}(\theta^{\mathrm{per}}).$$

Then

$$\|\nabla_{W_\ell}\bar{\mathcal{L}}_{em}(\theta^{\mathrm{per}})\|^2$$

$$\leq 2\|\nabla^{\mathrm{per}}_{W_\ell}\bar{\mathcal{L}}_{em}(\theta)\|_F^2 + 2\|\nabla^{\mathrm{per}}_{W_\ell}\bar{\mathcal{L}}_{em}(\theta) - \nabla_{W_\ell}\bar{\mathcal{L}}_{em}(\theta^{\mathrm{per}})\|_F^2$$

$$= \mathcal{O}\left( md^{2a-1}N\max_j w(t_j)(t_j - t_{j-1})\bar{\sigma}_{t_j} \right)\bar{\mathcal{L}}_{em}(\theta^{\mathrm{per}}).$$

Also,

$$\|\nabla_W \bar{\mathcal{L}}_{em}(\theta^{\mathrm{per}})\|^2$$
$$\geq \|\nabla_{W_L} \bar{\mathcal{L}}_{em}(\theta^{\mathrm{per}})\|^2$$
$$\geq \frac{1}{3}\|\nabla_{W_L} \bar{\mathcal{L}}_{em}(\theta)\|^2 - \|\nabla_{W_\ell} \bar{\mathcal{L}}_{em}(\theta) - \nabla_{W_\ell}^{\mathrm{per}} \bar{\mathcal{L}}_{em}(\theta)\|_F^2 - \|\nabla_{W_\ell}^{\mathrm{per}} \bar{\mathcal{L}}_{em}(\theta) - \nabla_{W_\ell} \bar{\mathcal{L}}_{em}(\theta^{\mathrm{per}})\|_F^2$$
$$= \Omega\left(\frac{md^{\frac{a_0-1}{2}}}{n^3 N^2} w(t_{j^*})(t_{j^*} - t_{j^*-1})\bar{\sigma}_{t_{j^*}}\right)\min\{\bar{\mathcal{L}}_{em}(\theta), \bar{\mathcal{L}}_{em}(\theta^{\mathrm{per}})\}.$$

$\square$

Note when interpolation is not achievable, this lower bound is always away from 0, which means the current technique can only evaluate the lower bound outside a neighbourhood of the minimizer. More advanced method is needed and we leave it for future investigation.

**Lemma 8** (semi-smoothness). *Let $\omega = \Omega$. With probability at least $1 - e^{-\Omega(\log m)}$ over the randomness of $\theta^{(0)}$, we have for all $\theta$ s.t. $\|\theta - \theta^{(0)}\| \leq \omega$, and all $\theta^{\mathrm{per}}$ s.t. $\|\theta^{\mathrm{per}} - \theta\| \leq \omega$,*

$$\bar{\mathcal{L}}_{em}(\theta^{\mathrm{per}}) \leq \bar{\mathcal{L}}_{em}(\theta) + \langle \nabla \bar{\mathcal{L}}_{em}(\theta), \theta^{\mathrm{per}} - \theta \rangle$$
$$+ \sqrt{\bar{\mathcal{L}}_{em}(\theta)}\sqrt{\sum_j w(t_j)(t_j - t_{j-1})\bar{\sigma}_{t_j}} \mathcal{O}(\omega^{1/3} L^2 \sqrt{m \log m} d^{a/2})\|\theta^{\mathrm{per}} - \theta\|$$
$$+ \sqrt{\bar{\mathcal{L}}_{em}(\theta)}\sqrt{\sum_j w(t_j)(t_j - t_{j-1})\bar{\sigma}_{t_j}} \mathcal{O}(L^2 \sqrt{m} d^a)\|\theta^{\mathrm{per}} - \theta\|^2$$

*Proof.* By definition,

$$\bar{\mathcal{L}}_{em}(\theta^{\mathrm{per}}) - \bar{\mathcal{L}}_{em}(\theta) - \langle \nabla \bar{\mathcal{L}}_{em}(\theta), \theta^{\mathrm{per}} - \theta \rangle$$
$$= \frac{1}{n}\sum_{i=1}^n \sum_{j=1}^N w(t_j)(t_j - t_{j-1})(\bar{\sigma}_{t_j} W_{L+1} q_{ij,L} + \xi_{ij})^\top W_{L+1}$$
$$\times \left(q_{ij,L}^{\mathrm{per}} - q_{ij,L} - \sum_{\ell=1}^L D_{ij,L} W_{ij,L} \cdots W_{ij,\ell+1} D_{ij,\ell}(W_{ij,\ell}^{\mathrm{per}} - W_{ij,\ell})q_{ij,\ell}\right)$$
$$+ \frac{1}{2\bar{\sigma}_{t_j}} w(t_j)(t_j - t_{j-1})\|\bar{\sigma}_{t_j} W_{L+1}(q_{ij,L}^{\mathrm{per}} - q_{ij,L})\|^2.$$

Similar to the proof of Theorem 4 in Allen-Zhu et al. [1], we obtain the desired bound by using Cauchy-Schwartz inequality. Note, in our case, due to the order of input data, we choose $s = \mathcal{O}(d^a m \omega^{2/3} L)$ in Allen-Zhu et al. [1] and therefore the bound is slightly different from theirs. $\square$

# E Proofs for sampling

In this section, we prove Theorem 2. The proof includes two main steps: 1. decomposing $\mathrm{KL}(p_\delta | q_{T-\delta})$ into the initialization error, the score estimation errors and the discretization errors; 2. estimating the initialization error and the discretization error based on our assumptions. In the following context, we introduce the proof of these two steps separately.

*Proof of Theorem 2.* **Step 1:** The error decomposition follows from the ideas in [12] of studying VPSDE-based diffusion models. According to the chain rule of KL divergence, we have

$$\mathrm{KL}(p_\delta | q_{T-\delta}) \leq \mathrm{KL}(p_T | q_0) + \mathbb{E}_{y \sim p_T}[\mathrm{KL}(p_{\delta|T}(\cdot|y) | q_{T-\delta|0}(\cdot|y))],$$

Apply the chain rule again for at across the time schedule $(T - t_j^\leftarrow)_{0 \leq j \leq N-1}$, the second term can be written as

$$\mathbb{E}_{y \sim p_T}[\mathrm{KL}(p_{\delta|T}(\cdot|y)|q_{T-\delta|0}(\cdot|y))]$$

$$\leq \sum_{j=0}^{N-1} \mathbb{E}_{y_j \sim p_{T-t_j^\leftarrow}}[\mathrm{KL}(p_{T-t_{j+1}^\leftarrow|T-t_j^\leftarrow}(\cdot|y_j)|q_{t_{j+1}^\leftarrow|t_j^\leftarrow}(\cdot|y_j))]$$

$$\leq \frac{1}{2} \sum_{j=0}^{N-1} \int_{t_j^\leftarrow}^{t_{j+1}^\leftarrow} \sigma_{T-t}^2 \mathbb{E}[\|s(\theta; T - t_j^\leftarrow, Y_{t_j^\leftarrow}) - \nabla \log p_{T-t}(Y_t)\|^2] dt$$

$$\leq \sum_{j=0}^{N-1} \int_{t_j^\leftarrow}^{t_{j+1}^\leftarrow} \sigma_{T-t}^2 \mathbb{E}[\|s(\theta; T - t_j^\leftarrow, Y_{t_j^\leftarrow}) - \nabla \log p_{T-t_j^\leftarrow}(Y_{t_j^\leftarrow})\|^2] dt$$

$$+ \sum_{j=0}^{N-1} \int_{t_j^\leftarrow}^{t_{j+1}^\leftarrow} \sigma_{T-t}^2 \mathbb{E}[\|\nabla \log p_{T-t_j^\leftarrow}(Y_{t_j^\leftarrow}) - \nabla \log p_{T-t}(Y_t)\|^2] dt,$$

where the second inequality follows from Lemma 9. Therefore, the error decomposition writes as

$$\mathrm{KL}(p_\delta|q_{T-\delta}) \lesssim \mathrm{KL}(p_T|q_0) + \sum_{j=0}^{N-1} \int_{t_j^\leftarrow}^{t_{j+1}^\leftarrow} \sigma_{T-t}^2 \mathbb{E}[\|s(\theta; T - t_j^\leftarrow, Y_{t_j^\leftarrow}) - \nabla \log p_{T-t_j^\leftarrow}(Y_{t_j^\leftarrow})\|^2] dt$$

$$+ \sum_{j=0}^{N-1} \int_{t_j^\leftarrow}^{t_{j+1}^\leftarrow} \sigma_{T-t}^2 \mathbb{E}[\|\nabla \log p_{T-t_j^\leftarrow}(Y_{t_j^\leftarrow}) - \nabla \log p_{T-t}(Y_t)\|^2] dt \qquad (20)$$

where the three terms in (20) quantify the initialization error, the score estimation error and the discretization error, respectively.

**Step 2:** In this step, we estimate the three error terms in Step 1. First, recall that $p_T = p * \mathcal{N}(0, \bar{\sigma}_T^2 I_d)$ and $q_0 = \mathcal{N}(0, \bar{\sigma}_T^2 I_d)$, hence the initialization error $\mathrm{KL}(p_T|q_0)$ can be estimated as follows,

$$\mathrm{KL}(p_T|q_0) = \mathrm{KL}(p * \mathcal{N}(0, \bar{\sigma}_T^2 I_d)|\mathcal{N}(0, \bar{\sigma}_T^2 I_d)) \lesssim \frac{\mathrm{m}_2^2}{\bar{\sigma}_T^2}, \qquad (21)$$

where the inequality follows from Lemma 10. Hence we recover the term $E_I$ in (11).

Next, since $\sigma_t$ is non-decreasing in $t$, the score estimation error can be estimated as

$$\sum_{j=0}^{N-1} \int_{t_j^\leftarrow}^{t_{j+1}^\leftarrow} \sigma_{T-t}^2 \mathbb{E}[\|s(\theta; T - t_j^\leftarrow, Y_{t_j^\leftarrow}) - \nabla \log p_{T-t_j^\leftarrow}(Y_{t_j^\leftarrow})\|^2] dt$$

$$\leq \sum_{j=0}^{N-1} \gamma_j \sigma_{T-t_j^\leftarrow}^2 \mathbb{E}[\|s(\theta; T - t_j^\leftarrow, Y_{t_j^\leftarrow}) - \nabla \log p_{T-t_j^\leftarrow}(Y_{t_j^\leftarrow})\|^2]. \qquad (22)$$

Hence, we recover the term $E_S$ in (11).

Last, we estimated the discretization error term. Our approach is motivated by analyses of VPSDEs in [8, 26]. We defines a process $L_t := \nabla \log p_{T-t}(Y_t)$. Then we can relate discretization error to quantities depending on $L_t$, and therefore bound the discretization error via properties of $\{L_t\}_{0 \leq t \leq T}$. According to Lemma 12, we have

$$\sum_{j=0}^{N-1} \int_{t_j^\leftarrow}^{t_{j+1}^\leftarrow} \sigma_{T-t}^2 \mathbb{E}[\|\nabla \log p_{T-t_j^\leftarrow}(Y_{t_j^\leftarrow}) - \nabla \log p_{T-t}(Y_t)\|^2] dt$$

$$\leq \underbrace{2d \sum_{j=0}^{N-1} \int_{t_j^\leftarrow}^{t_{j+1}^\leftarrow} \int_{t_j^\leftarrow}^{t} \sigma_{T-t}^2 \sigma_{T-u}^2 \bar{\sigma}_{T-u}^{-4} du dt}_{N_1} + \underbrace{\sum_{j=0}^{N-1} \int_{t_j^\leftarrow}^{t_{j+1}^\leftarrow} \sigma_{T-t}^2 dt \bar{\sigma}_{T-t_j^\leftarrow}^{-4} \mathbb{E}[\mathrm{tr}(\Sigma_{T-t_j^\leftarrow}(X_{T-t_j^\leftarrow}))]}_{N_2}$$

$$\underbrace{- \sum_{j=0}^{N-1} \int_{t_j^\leftarrow}^{t_{j+1}^\leftarrow} \sigma_{T-t}^2 \bar{\sigma}_{T-t}^{-4} \mathbb{E}[\mathrm{tr}(\Sigma_{T-t}(X_{T-t}))] dt}_{N_3}.$$

Since $t \mapsto \sigma_t$ is non-decreasing and $t \mapsto \mathbb{E}[\mathrm{tr}(\Sigma_{T-t}(X_{T-t}))]$ is non-increasing, we have

$$N_1 = 2d \sum_{j=0}^{N-1} \int_{T-t_{j+1}^{\leftarrow}}^{T-t_j^{\leftarrow}} \int_{T-t_{j+1}^{\leftarrow}}^{T-u} \sigma_t^2 dt \sigma_u^2 \bar{\sigma}_u^{-4} du \leq 2d \sum_{j=0}^{N-1} \gamma_j \int_{T-t_{j+1}^{\leftarrow}}^{T-t_j^{\leftarrow}} \sigma_t^4 \bar{\sigma}_t^{-4} dt,$$

$$N_2 = \sum_{j=0}^{N-1} \int_{t_j^{\leftarrow}}^{t_{j+1}^{\leftarrow}} \sigma_{T-t}^2 dt \bar{\sigma}_{T-t_j^{\leftarrow}}^{-4} \mathbb{E}[\mathrm{tr}(\Sigma_{T-t_j^{\leftarrow}}(X_{T-t_j^{\leftarrow}}))],$$

$$N_3 \leq -\sum_{j=0}^{N-1} \int_{t_j^{\leftarrow}}^{t_{j+1}^{\leftarrow}} \sigma_{T-t}^2 \bar{\sigma}_{T-t}^{-4} dt \mathbb{E}[\mathrm{tr}(\Sigma_{T-t_{j+1}^{\leftarrow}}(X_{T-t_{j+1}^{\leftarrow}}))].$$

Therefore, we obtain

$$\sum_{j=0}^{N-1} \int_{t_j^{\leftarrow}}^{t_{j+1}^{\leftarrow}} \sigma_{T-t}^2 \mathbb{E}[\|\nabla \log p_{T-t_j^{\leftarrow}}(Y_{t_j^{\leftarrow}}) - \nabla \log p_{T-t}(Y_t)\|^2] dt$$

$$\leq 2d \sum_{j=0}^{N-1} \gamma_j \int_{T-t_{j+1}^{\leftarrow}}^{T-t_j^{\leftarrow}} \sigma_t^4 \bar{\sigma}_t^{-4} dt + \sum_{j=0}^{N-1} \int_{t_j^{\leftarrow}}^{t_{j+1}^{\leftarrow}} \sigma_{T-t}^2 dt \bar{\sigma}_{T-t_j^{\leftarrow}}^{-4} \mathbb{E}[\mathrm{tr}(\Sigma_{T-t_j^{\leftarrow}}(X_{T-t_j^{\leftarrow}}))]$$

$$- \sum_{j=0}^{N-1} \int_{t_j^{\leftarrow}}^{t_{j+1}^{\leftarrow}} \sigma_{T-t}^2 \bar{\sigma}_{T-t}^{-4} dt \mathbb{E}[\mathrm{tr}(\Sigma_{T-t_{j+1}^{\leftarrow}}(X_{T-t_{j+1}^{\leftarrow}}))]$$

$$= 2d \sum_{j=0}^{N-1} \gamma_j \int_{T-t_{j+1}^{\leftarrow}}^{T-t_j^{\leftarrow}} \sigma_t^4 \bar{\sigma}_t^{-4} dt + \int_0^{t_1^{\leftarrow}} \sigma_{T-t}^2 dt \bar{\sigma}_T^{-4} \mathbb{E}[\mathrm{tr}(\Sigma_T(X_T))]$$

$$+ \sum_{j=1}^{N-1} \Big( \int_{t_j^{\leftarrow}}^{t_{j+1}^{\leftarrow}} \sigma_{T-t}^2 \bar{\sigma}_{T-t_j^{\leftarrow}}^{-4} dt - \int_{t_{j-1}^{\leftarrow}}^{t_j^{\leftarrow}} \sigma_{T-t}^2 \bar{\sigma}_{T-t}^{-4} dt \Big) \mathbb{E}[\mathrm{tr}(\Sigma_{T-t_j^{\leftarrow}}(X_{T-t_j^{\leftarrow}}))]. \tag{23}$$

The above bound depends on $\mathbb{E}[\mathrm{tr}(\Sigma_t(X_t))]$, hence we estimate $\mathbb{E}[\mathrm{tr}(\Sigma_t(X_t))]$ for different values of $t$.

First, we have

$$\mathbb{E}[\mathrm{tr}(\Sigma_t(X_t))] = \mathbb{E}[\mathbb{E}[\|X_0\|^2 | X_t] - \|\mathbb{E}[X_0 | X_t]\|^2] \leq \mathbb{E}[\|X_0\|^2] = \mathrm{m}_2^2.$$

Meanwhile,

$$\mathbb{E}[\mathrm{tr}(\Sigma_t(X_t))] = \mathbb{E}[\mathrm{tr}(\mathrm{Cov}(X_0 - X_t | X_t))] = \mathbb{E}[\mathbb{E}[\|X_0 - X_t\|^2 | X_t]] - \|\mathbb{E}[X_0 - X_t | X_t]\|^2$$

$$\leq \mathbb{E}[\|X_0 - X_t\|^2] = \bar{\sigma}_t^2 d$$

Therefore, $\mathbb{E}[\mathrm{tr}(\Sigma_t(X_t))] \leq \min(\mathrm{m}_2^2, \bar{\sigma}_t^2 d) \lesssim (1 - e^{-\bar{\sigma}_t^2})(\mathrm{m}_2^2 + d)$. Plug this estimation into (23) and we get

$$\sum_{j=0}^{N-1} \int_{t_j^{\leftarrow}}^{t_{j+1}^{\leftarrow}} \sigma_{T-t}^2 \mathbb{E}[\|\nabla \log p_{T-t_j^{\leftarrow}}(Y_{t_j^{\leftarrow}}) - \nabla \log p_{T-t}(Y_t)\|^2] dt$$

$$\lesssim d \sum_{j=0}^{N-1} \gamma_j \int_{T-t_{j+1}^{\leftarrow}}^{T-t_j^{\leftarrow}} \sigma_t^4 \bar{\sigma}_t^{-4} dt + \int_0^{t_1^{\leftarrow}} \sigma_{T-t}^2 dt \bar{\sigma}_T^{-4} \mathrm{m}_2^2$$

$$+ (\mathrm{m}_2^2 + d) \sum_{j=1}^{N-1} (1 - e^{-\bar{\sigma}_{T-t_j^{\leftarrow}}^2}) \Big( \int_{t_j^{\leftarrow}}^{t_{j+1}^{\leftarrow}} \sigma_{T-t}^2 \bar{\sigma}_{T-t_j^{\leftarrow}}^{-4} dt - \int_{t_{j-1}^{\leftarrow}}^{t_j^{\leftarrow}} \sigma_{T-t}^2 \bar{\sigma}_{T-t}^{-4} dt \Big)$$

$$\lesssim d \sum_{j=0}^{N-1} \gamma_j \int_{T-t_{j+1}^{\leftarrow}}^{T-t_j^{\leftarrow}} \sigma_t^4 \bar{\sigma}_t^{-4} dt + \mathrm{m}_2^2 \frac{\int_0^{t_1^{\leftarrow}} \sigma_{T-t}^2 dt}{\bar{\sigma}_T^4} + + (\mathrm{m}_2^2 + d) \sum_{k=1}^{N-1} (1 - e^{-\bar{\sigma}_{T-t_j^{\leftarrow}}^2}) \frac{\bar{\sigma}_{T-t_j^{\leftarrow}}^4 - \bar{\sigma}_{T-t_{j+1}^{\leftarrow}}^2 \bar{\sigma}_{T-t_{j-1}^{\leftarrow}}^2}{\bar{\sigma}_{T-t_{j-1}^{\leftarrow}}^2 \bar{\sigma}_{T-t_j^{\leftarrow}}^4},$$

where the last inequality follows from the definition of $\bar{\sigma}_t$ and integration by parts. The proof of Theorem 2 is completed. $\qquad\square$

**Lemma 9.** *Let $\{Y_t\}_{0 \leq t \leq T}$ be the solution to (2) with $f_t \equiv 0$ and $p_{t+s|s}^{\leftarrow}(\cdot|y)$ be the conditional distribution of $Y_{s+t}$ given $\{Y_s = y\}$. Let $\{\bar{Y}_t\}_{0 \leq t \leq T}$ be the solution to (11) $q_{t+s|s}(\cdot|y)$ be the conditional distribution of $\bar{Y}_{s+t}$ given $\{\bar{Y}_s = y\}$. Then for any fixed $t \in (0, \gamma_j]$, we have*

$$\mathbb{E}_{y \sim p_{t_j^{\leftarrow}}^{\leftarrow}} KL(p_{t_j^{\leftarrow}+t|t_j^{\leftarrow}}^{\leftarrow}(\cdot|y) | q_{t_j^{\leftarrow}+t|t_j^{\leftarrow}}(\cdot|y)) \leq \frac{1}{2} \sigma_{T-t}^2 \mathbb{E}[\|s(\theta; T - t_j^{\leftarrow}, Y_{t_j^{\leftarrow}}) - \nabla \log p_{T-t}(Y_t)\|^2]$$

*Proof of Lemma 9.* According to [12, Lemma 6], we have

$$\text{KL}(p^{\leftarrow}_{t_j^{\leftarrow}+t|t_j^{\leftarrow}}(\cdot|y)|q_{t_j^{\leftarrow}+t|t_j^{\leftarrow}}(\cdot|y))$$

$$\leq -2\sigma_{T-t}^2 \int p^{\leftarrow}_{t_j^{\leftarrow}+t|t_j^{\leftarrow}}(x|y)\|\nabla\log\frac{p^{\leftarrow}_{t_j^{\leftarrow}+t|t_j^{\leftarrow}}(x|y)}{q_{t_j^{\leftarrow}+t|t_j^{\leftarrow}}(x|y)}\|^2 dx$$

$$+ 2\sigma_{T-t}^2 \mathbb{E}_{p^{\leftarrow}_{t_j^{\leftarrow}+t|t_j^{\leftarrow}}(x|y)}[\langle\nabla\log p_{T-t}(x) - s(\theta; T - t_j^{\leftarrow}, Y_{t_j^{\leftarrow}}), \nabla\log\frac{p^{\leftarrow}_{t_j^{\leftarrow}+t|t_j^{\leftarrow}}(x|y)}{q_{t_j^{\leftarrow}+t|t_j^{\leftarrow}}(x|y)}\rangle]$$

$$\leq \frac{1}{2}\sigma_{T-t}^2 \mathbb{E}_{p^{\leftarrow}_{t_j^{\leftarrow}+t|t_j^{\leftarrow}}(x|y)}[\|\nabla\log p_{T-t}(x) - s(\theta; T - t_j^{\leftarrow}, Y_{t_j^{\leftarrow}})\|^2],$$

where the last inequality follows from Young's inequality. Therefore, Lemma 9 is proved after taking another expectation.

$\square$

**Lemma 10.** *For any probability distribution $p$ satisfying Assumption 3 and $q$ being a centered multivariate normal distribution with covariance matrix $\sigma^2 I_d$, we have*

$$KL(p * q|q) \leq \frac{\text{m}_2^2}{2\sigma^2}.$$

*Proof of Lemma 10.*

$$\text{KL}(p * q|q) \leq \int \text{KL}(q(\cdot - y)|q(\cdot))p(dy) = \int \text{KL}(\mathcal{N}(y, \sigma^2 I_d)|\mathcal{N}(0, \sigma^2 I_d))p(dy)$$

$$= \frac{1}{2}\int \ln(1) - d + \text{tr}(I_d) + \|y\|^2\sigma^{-2}p(dy) = \frac{\text{m}_2^2}{2\sigma^2},$$

where the inequality follows from convexity of $\text{KL}(\cdot|q)$ and the second identity follows from KL-divergence between multivariate normal distributions. $\square$

**Lemma 11.** *Let $\{X_t\}_{0\leq t\leq T}$ be the solution to (1) with $f_t \equiv 0$ and $p_{0|t}(\cdot|x)$ be the conditional distribution of $X_0$ given $\{X_t = x\}$. Define*

$$m_t(X_t) := \mathbb{E}_{X\sim p_{0|t}(\cdot|X_t)}[X], \quad \Sigma_t(X_t) = Cov_{X\sim p_{0|t}(\cdot|X_t)}(X). \tag{24}$$

*Let $\{Y_t\}_{0\leq t\leq T}$ be the solution to (2) with $f_t \equiv 0$ and $q_{0|t}(\cdot|x)$ be the conditional distribution of $Y_0$ given $\{Y_t = x\}$. Define*

$$\bar{m}_t(Y_t) := \mathbb{E}_{X\sim q_{0|t}(\cdot|Y_t)}[X], \quad \bar{\Sigma}_t(Y_t) = Cov_{X\sim q_{0|t}(\cdot|Y_t)}(X). \tag{25}$$

*Then we have for all $t \in (0, T)$,*

$$d\bar{m}_t(Y_t) = \sqrt{2}\sigma_{T-t}\bar{\sigma}_{T-t}^{-2}\bar{\Sigma}_t(Y_t)d\tilde{W}_t,$$

$$\text{and} \qquad \frac{d}{dt}\mathbb{E}[\Sigma_t(X_t)] = 2\sigma_t^2\bar{\sigma}_t^{-4}\mathbb{E}[\Sigma_t(X_t)^2].$$

*Proof of Lemma 11.* We first represent $\nabla log p_t(X_t)$ and $\nabla^2\log p_t(X_t)$ via $m_t(X_t)$ and $\Sigma_t(X_t)$. Since $\{X_t\}_{0\leq t\leq T}$ solves (1), $X_t = X_0 + \bar{\sigma}_t\xi$ with $(X_0, \xi) \sim p \otimes \mathcal{N}(0, I_d)$. Therefore, according to Bayes rule, we have

$$\nabla\log p_t(X_t) = \frac{1}{p_t(X_t)}\int \nabla\log p_{t|0}(X_t|x)p_{0,t}(x, X_t)dx$$

$$= \mathbb{E}_{x\sim p_{0|t}(\cdot|X_t)}[\bar{\sigma}_t^{-2}(X_t - x)]$$

$$= -\bar{\sigma}_t^{-2}(X_t - m_t(X_t)), \tag{26}$$

where the second identity follows from the fact that $p_{t|0}(\cdot|x) = \mathcal{N}(x, \bar{\sigma}_t^2 I_d)$. The last identity follows from the definition of $m_t(X_t)$ in Lemma 11. Similarly, according to Bayes rule, we can compute

$$
\begin{aligned}
&\nabla^2 \log p_t(X_t) \\
&= \frac{1}{p_t(X_t)} \int \nabla^2 \log p_{t|0}(X_t|x) p_{0,t}(x, X_t) dx \\
&\quad + \frac{1}{p_t(X_t)} \int \big(\nabla \log p_{t|0}(X_t|x)\big)\big(\nabla \log p_{t|0}(X_t|x)\big)^\top p_{0,t}(x, X_t) dx \\
&\quad - \frac{1}{p_t(X_t)^2} \Big(\int \nabla \log p_{t|0}(X_t|x) p_{0,t}(x, X_t) dx\Big)\Big(\int \nabla \log p_{t|0}(X_t|x) p_{0,t}(x, X_t) dx\Big)^\top \\
&= -\bar{\sigma}_t^{-2} I_d + \bar{\sigma}_t^{-4} \Sigma_t(X_t),
\end{aligned}
\tag{27}
$$

where the second identity follows from the fact that $p_{t|0}(\cdot|x) = \mathcal{N}(x, \bar{\sigma}_t^2 I_d)$ and the definition of $\Sigma_t(X_t)$ in Lemma 11.

According to Bayes rule, we have

$$
p_{0|t}(dx|X_t) \propto \exp\Big(-\frac{1}{2}\frac{\|X_t - x\|^2}{\bar{\sigma}_t^2}\Big) p(dx)
$$

and

$$
\begin{aligned}
q_{0|t}(dx|Y_t) &= Z^{-1} \exp\Big(-\frac{1}{2}\frac{\|Y_t - x\|^2}{\bar{\sigma}_{T-t}^2}\Big) p(dx) \\
&= Z_t^{-1} \exp\Big(-\frac{1}{2}\bar{\sigma}_{T-t}^{-2}\|x\|^2 + \bar{\sigma}_{T-t}^{-2}\langle x, Y_t\rangle\Big) p(dx) \\
&:= Z_t^{-1} \exp(h_t(x)) p(dx),
\end{aligned}
\tag{28}
$$

where $Z_t = \int \exp(h_t(x)) p(dx)$ is a (random) normalization constant. From the above computations, we can see that $q_{0|t}(dx|Y_t) = p_{0|T-t}(dx|X_{T-t})$ for all $t \in [0, T]$. Therefore, we have

$$
\bar{m}_t(Y_t) = \mathbb{E}_{X \sim q_{0|t}(\cdot|Y_t)}[X] = m_{T-t}(X_{T-t}), \quad \bar{\Sigma}_t(Y_t) = \mathrm{Cov}_{X \sim q_{0|t}(\cdot|Y_t)}(X) = \Sigma_{T-t}(X_{T-t}),
$$

where the identities hold in distribution. Therefore, to prove the first statement, it suffices to compute $d\bar{m}_t(Y_t)$. To do so, we first compute $dh_t(x)$, $d[h(x), h(x)]_t$, $dZ_t$ and $d\log Z_t$.

$$
dh_t(x) = \bar{\sigma}_{T-t}^{-3}\dot{\bar{\sigma}}_{T-t}\|x\|^2 dt - 2\bar{\sigma}_{T-t}^{-3}\dot{\bar{\sigma}}_{T-t}\bar{\sigma}\langle x, Y_t\rangle dt + \bar{\sigma}_{T-t}^{-2}\langle x, dY_t\rangle.
\tag{29}
$$

According to the definition of $Y_t$ and (26), we have

$$
\begin{aligned}
dY_t &= 2\sigma_{T-t}^2 \nabla \log p_{T-t}(Y_t) dt + \sqrt{2\sigma_{T-t}^2} d\tilde{W}_t \\
&= -2\sigma_{T-t}^2 \bar{\sigma}_{T-t}^{-2}(Y_t - \bar{m}_t(Y_t)) dt + \sqrt{2\sigma_{T-t}^2} d\tilde{W}_t.
\end{aligned}
$$

Therefore

$$
d[h(x), h(x)]_t = \bar{\sigma}_{T-t}^{-4}|x|^2[dY, dY]_t = 2\sigma_{T-t}^2\bar{\sigma}_{T-t}^{-4}\|x\|^2.
\tag{30}
$$

Apply (29) and (30) and we get

$$
\begin{aligned}
dZ_t &= \int \exp(h_t(x))\big(dh_t(x) + \frac{1}{2}d[h(x), h(x)]_t\big) p(dx) \\
&= \bar{\sigma}_{T-t}^{-3}\dot{\bar{\sigma}}_{T-t}\mathbb{E}_{q_{0|t}(\cdot|Y_t)}[\|x\|^2]Z_t dt - 2\bar{\sigma}_{T-t}^{-3}\dot{\bar{\sigma}}_{T-t}\langle Y_t, \bar{m}_t(Y_t)\rangle Z_t dt \\
&\quad + \bar{\sigma}_{T-t}^{-2}\langle \bar{m}_t(Y_t), dY_t\rangle Z_t + \sigma_{T-t}^2\bar{\sigma}_{T-t}^{-4}\mathbb{E}_{q_{0|t}(\cdot|Y_t)}[\|x\|^2]Z_t dt,
\end{aligned}
\tag{31}
$$

and

$$
\begin{aligned}
d\log Z_t &= Z_t^{-1} dZ_t - \frac{1}{2}Z_t^{-2}d[Z, Z]_t \\
&= -2\bar{\sigma}_{T-t}^{-3}\dot{\bar{\sigma}}_{T-t}\langle Y_t, \bar{m}_t(Y_t)\rangle dt + \bar{\sigma}_{T-t}^{-2}\langle \bar{m}_t(Y_t), dY_t\rangle - \sigma_{T-t}^2\bar{\sigma}_{T-t}^{-4}\|\bar{m}_t(Y_t)\|^2 dt.
\end{aligned}
\tag{32}
$$

If we further define $R_t(Y_t) := \frac{q_{0|t}(dx|Y_t)}{p(dx)} = Z_t^{-1}\exp(h_t(x))$. We have

$$dR_t(Y_t) = d\exp(\log R_t(Y_t)) = R_t(Y_t)d(\log R_t(Y_t)) + \frac{1}{2}R_t(Y_t)d[\log R_t(Y_t), \log R_t(Y_t)]$$

$$= -R_t(Y_t)d(\log Z_t) + R_t(Y_t)dh_t(x) + \frac{1}{2}R_t(Y_t)d[h_t(x) - \log Z_t, h_t(x) - \log Z_t] \quad (33)$$

With (29), (30), (31), (32) and (33), we have

$$d\bar{m}_t(Y_t) = d\int xR_t(Y_t)p(dx)$$

$$= \int x\Big(-d(\log Z_t) + dh_t(x) + \frac{1}{2}d[h_t(x) - \log Z_t, h_t(x) - \log Z_t]\Big)q_{0|t}(dx|Y_t)$$

$$= \sqrt{2}\sigma_{T-t}\bar{\sigma}_{T-t}^{-2}\bar{\Sigma}_t(Y_t)d\tilde{W}_t, \quad (34)$$

where most terms cancel in the last identity. Therefore, the first statement is proved. Next, we prove the second statement. We have

$$\frac{d}{dt}\mathbb{E}[\Sigma_{T-t}(X_{T-t})] = \frac{d}{dt}\mathbb{E}[\bar{\Sigma}_t(Y_t)] = \frac{d}{dt}\mathbb{E}[\Sigma_{T-t}(X_{T-t})]$$

$$= \frac{d}{dt}\mathbb{E}_{Y_t\sim p_{T-t}}\big[\mathbb{E}_{q_{0|t}(\cdot|Y_t)}[x^{\otimes 2}] - \bar{m}_t(Y_t)^{\otimes 2}\big]$$

$$= \frac{d}{dt}\mathbb{E}_{q_0}[x^{\otimes 2}] - \frac{d}{dt}\mathbb{E}[\bar{m}_t(Y_t)^{\otimes 2}]$$

$$= -\mathbb{E}[-2\bar{m}_t(Y_t)d\bar{m}_t(Y_t)^\top + d[\bar{m}_t(Y_t), \bar{m}_t(Y_t)^\top]]$$

$$= 2\bar{\sigma}_{T-t}^{-3}\dot{\bar{\sigma}}_{T-t}\mathbb{E}[\bar{\Sigma}_t(Y_t)^2]dt$$

$$= -2\sigma_{T-t}^2\bar{\sigma}_{T-t}^{-4}\mathbb{E}[\Sigma_t(X_{T-t})^2],$$

where the second last identity follows from (34) and the last identity follows from the definition of $\bar{\sigma}_t$. Last, we reverse the time and get

$$\frac{d}{dt}\mathbb{E}[\Sigma_t(X_t)] = 2\sigma_t^2\bar{\sigma}_t^{-4}\mathbb{E}[\Sigma_t(X_t)^2].$$

The proof is completed. $\qquad\square$

**Lemma 12.** *Under the conditions in Lemma 11, let $\{Y_t\}_{0\le t\le T}$ be the solution to (2) with $f_t \equiv 0$. Define $L_t := \nabla\log p_{T-t}(Y_t)$, then for any $t \in [t_j^\leftarrow, t_{j+1}^\leftarrow)$, we have*

$$\mathbb{E}[\|L_t - L_{t_j^\leftarrow}\|^2] = 2d\int_{t_j^\leftarrow}^t \sigma_{T-u}^2\bar{\sigma}_{T-u}^{-4}du + \bar{\sigma}_{T-t_j^\leftarrow}^{-4}\mathbb{E}[\text{tr}(\Sigma_{T-t_j^\leftarrow}(X_{T-t_j^\leftarrow}))] - \bar{\sigma}_{T-t}^{-4}\mathbb{E}[\text{tr}(\Sigma_{T-t}(X_{T-t}))]$$

*Proof of Lemma 12.* First, according to the definition of $L_t$ and $Y_t$, it follows from Itô's lemma that

$$dL_t = \nabla^2\log p_{T-t}(Y_t)\big(2\sigma_{T-t}^2\nabla\log p_{T-t}(Y_t)dt + \sqrt{2\sigma_{T-t}}d\tilde{W}_t\big) \quad (35)$$

$$+ \Delta\big(\nabla\log p_{T-t}(Y_t)\big)\sigma_{T-t}^2dt + \frac{d(\nabla\log p_{T-t})}{dt}(Y_t)dt \quad (36)$$

$$= \sqrt{2\sigma_{T-t}^2}\nabla^2\log p_{T-t}(Y_t)d\tilde{W}_t, \quad (37)$$

where the last step follows from applying the Fokker Planck equation of (1) with $f_t \equiv 0$, i.e., $\partial_t p_t = \sigma_t^2\Delta p_t$. Most of the terms are cancelled after applying the Fokker Planck equation. Now, for fixed $s > 0$ and $t > s$, define $E_{s,t} := \mathbb{E}[\|L_t - L_s\|^2]$. Apply Itô's lemma and (35), we have

$$dE_{s,t} = 2\mathbb{E}[\langle L_t - L_s, dL_t\rangle] + d[L]_t$$

$$= 2\mathbb{E}[\langle L_t - L_s, \sqrt{2\sigma_{T-t}^2}\nabla\log p_{T-t}(Y_t)d\tilde{W}_t\rangle] + 2\sigma_{T-t}^2\mathbb{E}[\|\nabla^2\log p_{T-t}(Y_t)\|_F^2]dt$$

$$= 2\sigma_{T-t}^2\mathbb{E}[\|\nabla^2\log p_{T-t}(Y_t)\|_F^2]dt, \quad (38)$$

where $\|A\|_F$ denotes the Frobenius norm of any matrix $A$. According to (27), we have

$$\begin{aligned}
\frac{dE_{s,t}}{dt} &= 2\sigma_{T-t}^2 \mathbb{E}\big[\|\nabla^2 \log p_{T-t}(Y_t)\|_F^2\big] = 2\sigma_{T-t}^2 \mathbb{E}\big[\|\nabla^2 \log p_{T-t}(X_{T-t})\|_F^2\big] \\
&= 2\sigma_{T-t}^2 \mathbb{E}\big[\|-\bar{\sigma}_{T-t}^{-2} I_d + \bar{\sigma}_{T-t}^{-4} \Sigma_{T-t}(X_{T-t})\|_F^2\big] \\
&= 2d\sigma_{T-t}^2 \bar{\sigma}_{T-t}^{-4} - 4\sigma_{T-t}^2 \bar{\sigma}_{T-t}^{-6} \mathbb{E}[\mathrm{tr}(\Sigma_{T-t}(X_{T-t}))] + 2\sigma_{T-t}^2 \bar{\sigma}_{T-t}^{-8} \mathbb{E}[\mathrm{tr}(\Sigma_{T-t}(X_{T-t})^2)] \\
&= 2d\sigma_{T-t}^2 \bar{\sigma}_{T-t}^{-4} - 4\sigma_{T-t}^2 \bar{\sigma}_{T-t}^{-6} \mathbb{E}[\mathrm{tr}(\Sigma_{T-t}(X_{T-t}))] - \bar{\sigma}_{T-t}^{-4} \frac{d}{dt} \mathbb{E}[\mathrm{tr}(\Sigma_{T-t}(X_{T-t}))],
\end{aligned}$$

where the last identity follows from the proof of Lemma 11. Therefore, for any $t \in [t_j^\leftarrow, t_{j+1}^\leftarrow)$, we have

$$\begin{aligned}
E_{t_j^\leftarrow, t} &= 2d \int_{t_j^\leftarrow}^t \sigma_{T-u}^2 \bar{\sigma}_{T-u}^{-4} du - 4 \int_{t_j^\leftarrow}^t \sigma_{T-u}^2 \bar{\sigma}_{T-u}^{-6} \mathbb{E}[\mathrm{tr}(\Sigma_{T-u}(X_{T-u}))] du \\
&\quad - \int_{t_j^\leftarrow}^t \bar{\sigma}_{T-u}^{-4} \frac{d}{du} \mathbb{E}[\mathrm{tr}(\Sigma_{T-u}(X_{T-u}))] \\
&= 2d \int_{t_j^\leftarrow}^t \sigma_{T-u}^2 \bar{\sigma}_{T-u}^{-4} du - 4 \int_{t_j^\leftarrow}^t \sigma_{T-u}^2 \bar{\sigma}_{T-u}^{-6} \mathbb{E}[\mathrm{tr}(\Sigma_{T-u}(X_{T-u}))] du \\
&\quad - \bar{\sigma}_{T-t}^{-4} \mathbb{E}[\mathrm{tr}(\Sigma_{T-t}(X_{T-t}))] + \bar{\sigma}_{T-t_j^\leftarrow}^{-4} \mathbb{E}[\mathrm{tr}(\Sigma_{T-t_j^\leftarrow}(X_{T-t_j^\leftarrow}))] \\
&\quad + 4 \int_{t_j^\leftarrow}^t \sigma_{T-u}^2 \bar{\sigma}_{T-u}^{-6} \mathbb{E}[\mathrm{tr}(\Sigma_{T-u}(X_{T-u}))] \\
&= 2d \int_{t_j^\leftarrow}^t \sigma_{T-u}^2 \bar{\sigma}_{T-u}^{-4} du - \bar{\sigma}_{T-t}^{-4} \mathbb{E}[\mathrm{tr}(\Sigma_{T-t}(X_{T-t}))] + \bar{\sigma}_{T-t_j^\leftarrow}^{-4} \mathbb{E}[\mathrm{tr}(\Sigma_{T-t_j^\leftarrow}(X_{T-t_j^\leftarrow}))]
\end{aligned}$$

The proof is completed. $\square$

# F   Sampling error for Gaussian data distributions

In this section, we consider a special case when the data distribution is a mixture of Gaussians, i.e.,

$$p(x) = \mathcal{N}(x; m, \sigma^2 I_d), \tag{39}$$

where $\mathcal{N}(x; m, \sigma^2 I_d)$ is the density of Gaussian random vector with mean $m$ and covariance $\sigma^2 I_d$. In this case, the score function $\nabla \log p_t(x)$ can be explicitly calculated from any $t > 0$, see Lemma 14. Therefore, the sampling process (11) can be implemented with zero score estimation error via the following piecewise SDE: for any $t \in [t_j^\leftarrow, t_{j+1}^\leftarrow)$,

$$d\bar{Y}_t = 2\sigma_{T-t}^2 \nabla \log p_{T-t_j^\leftarrow}(\bar{Y}_{t_j^\leftarrow}) dt + \sqrt{2\sigma_{T-t}^2} d\bar{W}_t. \tag{40}$$

The iterates, $(\bar{Y}_{t_j^\leftarrow})$, are all Gaussians with explicit means and covariance matrices, see Lemma 15. As a consequence, we can quantify the quantity, $\mathrm{KL}(p_\delta | q_{T-\delta})$ in Theorem 2 explicitly since both distributions are Gaussians.

**Lemma 13** (KL-divergence error for Gaussian data distribution). *Assume the data distribution has density given by* (39). *Let* $(\bar{Y}_{t_j^\leftarrow})_{j=0}^N$ *be defined by* (40) *with initial condition* $\bar{Y}_0 \sim \mathcal{N}(0, \bar{\sigma}_T^2 I_d)$. *Denote* $q_t = Law(\bar{Y}_t)$ *for all* $0 \le t \le T - \delta$. *Then*

$$KL(p_\delta | q_{T-\delta}) = \frac{d}{2}(E_\sigma - 1 - \log E_\sigma) + \|m\|^2 \frac{(\sigma^2 + \bar{\sigma}_T^2)^2}{\sigma^2 + \bar{\sigma}_\delta^2} E_\sigma, \tag{41}$$

*where* $E_\sigma$ *is a positive constant depending on the variance schedule* $(\bar{\sigma}_{T-t_j^\leftarrow})_{j=0}^N$, *given by*

$$E_\sigma^{-1} = \frac{(\sigma^2 + \bar{\sigma}_\delta^2)\bar{\sigma}_T^2}{(\sigma^2 + \bar{\sigma}_T^2)^2} + \sum_{j=0}^{N-1} \frac{(\sigma^2 + \bar{\sigma}_\delta^2)(\bar{\sigma}_{T-t_j^\leftarrow}^2 - \bar{\sigma}_{T-t_{j+1}^\leftarrow}^2)}{(\sigma^2 + \bar{\sigma}_{T-t_{j+1}^\leftarrow}^2)^2}.$$

*Proof of Lemma 13.* $p_\delta = p * \mathcal{N}(0, \bar{\sigma}_\delta^2) = \mathcal{N}(m, (\sigma^2 + \bar{\sigma}_\delta^2)I_d)$ and $q_{T-\delta} = \mathrm{Law}(\bar{Y}_{t_N^\leftarrow}) = \mathcal{N}(m_N, \Sigma_N)$ with $(m_N, \Sigma_N)$ given in Lemma 15. Therefore, we have

$$
\begin{aligned}
\mathrm{KL}(p_\delta | q_{T-\delta}) &= \mathrm{KL}(\mathcal{N}(m, (\sigma^2 + \bar{\sigma}_\delta^2)I_d) | \mathcal{N}(m_N, \Sigma_N)) \\
&= \frac{1}{2} \log \frac{\det(\Sigma_N)}{\det((\sigma^2 + \bar{\sigma}_\delta^2)I_d)} - \frac{d}{2} + \frac{1}{2} \mathrm{tr}((\sigma^2 + \bar{\sigma}_\delta^2)\Sigma_N^{-1}) + (m_N - m)^\top \Sigma_N^{-1}(m - m_N) \\
&= \frac{d}{2} \log \left( \frac{(\sigma^2 + \bar{\sigma}_\delta^2)\bar{\sigma}_T^2}{(\sigma^2 + \bar{\sigma}_T^2)^2} + \sum_{j=0}^{N-1} \frac{(\sigma^2 + \bar{\sigma}_\delta^2)(\bar{\sigma}_{T-t_j^\leftarrow}^2 - \bar{\sigma}_{T-t_{j+1}^\leftarrow}^2)}{(\sigma^2 + \bar{\sigma}_{T-t_{j+1}^\leftarrow}^2)^2} \right) - \frac{d}{2} \\
&\quad + \frac{d}{2} \left( \frac{(\sigma^2 + \bar{\sigma}_\delta^2)\bar{\sigma}_T^2}{(\sigma^2 + \bar{\sigma}_T^2)^2} + \sum_{j=0}^{N-1} \frac{(\sigma^2 + \bar{\sigma}_\delta^2)(\bar{\sigma}_{T-t_j^\leftarrow}^2 - \bar{\sigma}_{T-t_{j+1}^\leftarrow}^2)}{(\sigma^2 + \bar{\sigma}_{T-t_{j+1}^\leftarrow}^2)^2} \right)^{-1} \\
&\quad + \left( \bar{\sigma}_T^2 + (\sigma^2 + \bar{\sigma}_T^2)^2 \sum_{j=0}^{N-1} \frac{(\bar{\sigma}_{T-t_j^\leftarrow}^2 - \bar{\sigma}_{T-t_{j+1}^\leftarrow}^2)}{(\sigma^2 + \bar{\sigma}_{T-t_{j+1}^\leftarrow}^2)^2} \right)^{-1} \|m\|^2.
\end{aligned}
$$

$\square$

**Lemma 14** (Explicit score function for mixture of Gaussian target). *Assume the data distribution has density given by* (39), *then the score function is given by*

$$
\nabla \log p_t(x) = -\frac{x - m}{\sigma^2 + \bar{\sigma}_t^2}. \tag{42}
$$

*Proof.* Since the forward process (1) with $f_t \equiv 0$ is the just a process that keeps adding noise, the density $p_t$ along the process is a convolution between data density and a Gaussian density with mean zero and covariance $\bar{\sigma}_t^2 I_d$:

$$
p_t(x) = p * \mathcal{N}(\cdot\, ; 0, \bar{\sigma}_t^2 I_d)(x) = \mathcal{N}(x; m, (\sigma^2 + \bar{\sigma}_t^2)I_d).
$$

Therefore, we have

$$
\begin{aligned}
\nabla p_t(x) &= (2\pi(\sigma^2 + \bar{\sigma}_t^2))^{-d/2} \exp\left( -\frac{\|x - m\|^2}{2(\sigma^2 + \bar{\sigma}_t^2)} \right)\left( -\frac{x - m}{\sigma^2 + \bar{\sigma}_t^2} \right) \\
&= -\frac{x - m}{\sigma^2 + \bar{\sigma}_t^2} \mathcal{N}(x; m, (\sigma^2 + \bar{\sigma}_t^2)I_d).
\end{aligned}
$$

(42) follows directly from the above computations. $\square$

**Lemma 15** (Gaussian iterates along the trajectory). *Assume the data distribution has density given by* (39). *Let* $(\bar{Y}_{t_j^\leftarrow})$ *be defined by* (40) *with initial condition* $\bar{Y}_0 \sim \mathcal{N}(0, \bar{\sigma}_T^2 I_d)$. *Then for all* $0 \le j \le N$, $\bar{Y}_{t_j^\leftarrow} \sim \mathcal{N}(m_j, \Sigma_j)$ *with*

$$
m_j = \frac{\bar{\sigma}_{T-t_0^\leftarrow}^2 + \bar{\sigma}_{T-t_j^\leftarrow}^2}{\sigma^2 + \bar{\sigma}_{T-t_0^\leftarrow}^2} m, \tag{43}
$$

$$
\Sigma_j = \left( \frac{(\sigma^2 + \bar{\sigma}_{T-t_j^\leftarrow}^2)^2 \bar{\sigma}_T^2}{(\sigma^2 + \bar{\sigma}_{T-t_0^\leftarrow}^2)^2} + \sum_{l=0}^{j-1} \frac{(\sigma^2 + \bar{\sigma}_{T-t_j^\leftarrow}^2)^2(\bar{\sigma}_{T-t_l^\leftarrow}^2 - \bar{\sigma}_{T-t_{l+1}^\leftarrow}^2)}{(\sigma^2 + \bar{\sigma}_{T-t_{l+1}^\leftarrow}^2)^2} \right) I_d. \tag{44}
$$

*Proof of Lemma 15.* According to Lemma 14, (40) can be written as

$$
d\bar{Y}_t = -2\sigma_{T-t}^2 \frac{\bar{Y}_{t_j^\leftarrow} - m}{\sigma^2 + \bar{\sigma}_{T-t_j^\leftarrow}^2} dt + \sqrt{2\sigma_{T-t}^2}\, d\bar{W}_t,
$$

which implies that for any $j \in 0, 1, \cdots, N-1$:

$$
\bar{Y}_{t_{j+1}^\leftarrow} - \bar{Y}_{t_j^\leftarrow} = -2\frac{\int_{t_j^\leftarrow}^{t_{j+1}^\leftarrow} \sigma_{T-t}^2 dt}{\sigma^2 + \bar{\sigma}_{T-t_j^\leftarrow}^2}\big(\bar{Y}_{t_j^\leftarrow} - m\big) + \sqrt{2\int_{t_j^\leftarrow}^{t_{j+1}^\leftarrow}\sigma_{T-t}^2 dt}\, U_{j+1}
$$

$$
= -\frac{\bar{\sigma}_{T-t_j^\leftarrow}^2 - \bar{\sigma}_{T-t_{j+1}^\leftarrow}^2}{\sigma^2 + \bar{\sigma}_{T-t_j^\leftarrow}^2}\big(\bar{Y}_{t_j^\leftarrow} - m\big) + \sqrt{\bar{\sigma}_{T-t_j^\leftarrow}^2 - \bar{\sigma}_{T-t_{j+1}^\leftarrow}^2}\, U_{j+1},
$$

$$
\implies \quad \bar{Y}_{t_{j+1}^\leftarrow} = \Big(1 - \frac{\bar{\sigma}_{T-t_j^\leftarrow}^2 - \bar{\sigma}_{T-t_{j+1}^\leftarrow}^2}{\sigma^2 + \bar{\sigma}_{T-t_j^\leftarrow}^2}\Big)\bar{Y}_{t_j^\leftarrow} + \frac{\bar{\sigma}_{T-t_j^\leftarrow}^2 - \bar{\sigma}_{T-t_{j+1}^\leftarrow}^2}{\sigma^2 + \bar{\sigma}_{t_j^\leftarrow}^2}m + \sqrt{\bar{\sigma}_{T-t_j^\leftarrow}^2 - \bar{\sigma}_{T-t_{j+1}^\leftarrow}^2}\, U_{j+1}
$$

$$(45)$$

where $(U_j)_{j=1}^N$ are i.i.d. standard Gaussian vectors in $\mathbb{R}^d$. Since $\bar{Y}_{t_0^\leftarrow}$ is Gaussian, by induction, we prove that $\bar{Y}_{t_j^\leftarrow}$ is Gaussian for all $j = 1, \cdots, N$. Denote $\bar{Y}_{t_j^\leftarrow} \sim \mathcal{N}(m_j, \Sigma_j)$. According to (45) and the independence between $U_{j+1}$ and $\bar{Y}_{t_j^\leftarrow}$, we have

$$
m_{j+1} = \Big(1 - \frac{\bar{\sigma}_{T-t_j^\leftarrow}^2 - \bar{\sigma}_{T-t_{j+1}^\leftarrow}^2}{\sigma^2 + \bar{\sigma}_{T-t_j^\leftarrow}^2}\Big)m_j + \frac{\bar{\sigma}_{T-t_j^\leftarrow}^2 - \bar{\sigma}_{T-t_{j+1}^\leftarrow}^2}{\sigma^2 + \bar{\sigma}_{t_j^\leftarrow}^2}m,
$$

$$
\implies \quad m_{j+1} - m = \frac{\sigma^2 + \bar{\sigma}_{T-t_{j+1}^\leftarrow}^2}{\sigma^2 + \bar{\sigma}_{T-t_j^\leftarrow}^2}(m_j - m),
$$

$$
\implies \quad m_j = \frac{\sigma^2 + \bar{\sigma}_{T-t_j^\leftarrow}^2}{\sigma^2 + \bar{\sigma}_{T-t_0^\leftarrow}^2}(m_0 - m) + m = \frac{\bar{\sigma}_{T-t_0^\leftarrow}^2 + \bar{\sigma}_{T-t_j^\leftarrow}^2}{\sigma^2 + \bar{\sigma}_{T-t_0^\leftarrow}^2}m.
$$

Again, according to (45) and the independence between $U_{j+1}$ and $\bar{Y}_{t_j^\leftarrow}$, we get a relation between consecutive covariance matrices:

$$
\Sigma_{j+1} = \Big(1 - \frac{\bar{\sigma}_{T-t_j^\leftarrow}^2 - \bar{\sigma}_{T-t_{j+1}^\leftarrow}^2}{\sigma^2 + \bar{\sigma}_{T-t_j^\leftarrow}^2}\Big)^2 \Sigma_j + \big(\bar{\sigma}_{T-t_j^\leftarrow}^2 - \bar{\sigma}_{T-t_{j+1}^\leftarrow}^2\big)I_d,
$$

$$
\implies \quad \frac{\Sigma_{j+1}}{(\sigma^2 + \bar{\sigma}_{T-t_{j+1}^\leftarrow}^2)^2} = \frac{\Sigma_j}{(\sigma^2 + \bar{\sigma}_{T-t_j^\leftarrow}^2)^2} + \frac{\bar{\sigma}_{T-t_j^\leftarrow}^2 - \bar{\sigma}_{T-t_{j+1}^\leftarrow}^2}{(\sigma^2 + \bar{\sigma}_{T-t_{j+1}^\leftarrow}^2)^2}I_d,
$$

$$
\implies \quad \frac{\Sigma_j}{(\sigma^2 + \bar{\sigma}_{T-t_j^\leftarrow}^2)^2} = \frac{\Sigma_0}{(\sigma^2 + \bar{\sigma}_{T-t_0^\leftarrow}^2)^2} + \sum_{l=0}^{j-1}\frac{\bar{\sigma}_{T-t_l^\leftarrow}^2 - \bar{\sigma}_{T-t_{l+1}^\leftarrow}^2}{(\sigma^2 + \bar{\sigma}_{T-t_{l+1}^\leftarrow}^2)^2}I_d,
$$

$$
\implies \quad \Sigma_j = \Big(\frac{(\sigma^2 + \bar{\sigma}_{T-t_j^\leftarrow}^2)^2\bar{\sigma}_T^2}{(\sigma^2 + \bar{\sigma}_{T-t_0^\leftarrow}^2)^2} + \sum_{l=0}^{j-1}\frac{(\sigma^2 + \bar{\sigma}_{T-t_j^\leftarrow}^2)^2(\bar{\sigma}_{T-t_l^\leftarrow}^2 - \bar{\sigma}_{T-t_{l+1}^\leftarrow}^2)}{(\sigma^2 + \bar{\sigma}_{T-t_{l+1}^\leftarrow}^2)^2}\Big)I_d.
$$

$\square$

# G  Full error analysis

*Proof of Theorem 3.* We only need to deal with $E_S$. By applying the same schedules to training objective, we obtain

$$
E_S = \sum_{j=0}^{N-1}\frac{\sigma_{t_{N-j}}^2}{w(t_{N-j})}\cdot w(t_{N-j})(t_{N-j} - t_{N-j-1})\frac{1}{\bar{\sigma}_{t_{N-j}}}\mathbb{E}_{X_0}\mathbb{E}_\xi\|\bar{\sigma}_{t_{N-j}}s(\theta; t_{N-j}, X_{t_{N-j}}) + \xi\|^2
$$

$$
+ \sum_{j=0}^{N-1}\frac{\sigma_{t_{N-j}}^2}{w(t_{N-j})}\cdot w(t_{N-j})(t_{N-j} - t_{N-j-1})\cdot C
$$

$$
\leq \max_j \frac{\sigma_{t_{N-j}}^2}{w(t_{N-j})}\cdot(\bar{\mathcal{L}}(W) + \bar{C}).
$$

Together with

$$\bar{\mathcal{L}}(W^{(k)}) + \bar{C} \le |\bar{\mathcal{L}}(W^{(K)}) - \bar{\mathcal{L}}_{em}(W^{(K)}) + \bar{\mathcal{L}}_{em}(\theta^*) - \bar{\mathcal{L}}(\theta^*)| + |\bar{\mathcal{L}}_{em}(W^{(K)}) - \bar{\mathcal{L}}_{em}(\theta^*)|$$
$$+ |\bar{\mathcal{L}}(\theta^*) - \bar{\mathcal{L}}(\theta_{\mathcal{F}})| + |\bar{\mathcal{L}}(\theta_{\mathcal{F}}) + \bar{C}|,$$

we have the result. $\qquad\qquad\square$

## H    Proofs for Section 4.1

### H.1    Proof of "bell-shaped" curve

*Proof of Proposition 1.* Fix $x_i, \xi_{ij}, \bar{\sigma}_{t_N}$. By definition of the network $S(\theta; t_j, X_{ij})$, it is continuous with respect to $X_{ij}$.

For 1, $X_{ij} = x_i + \bar{\sigma}_{t_j}\xi_{ij}$, and thus $S(\theta; t_j, X_{ij})$ is also continuous w.r.t. $\bar{\sigma}_{t_j}$. Also, since $\bar{\sigma}_{t_j} \in [0, \bar{\sigma}_{t_N}]$, there exists $M_0 > 0$, s.t.,

$$S(\theta; t_j, x_i + \bar{\sigma}_{t_j}\xi_{ij}) \in [-M_0, M_0]^d.$$

Then for any $\epsilon_1 > 0$, there exists $\delta = \frac{\epsilon_1}{\sqrt{d}M_0} > 0$, s.t., $\forall\, 0 \le \bar{\sigma}_{t_j} < \delta_1$, we have

$$\|\bar{\sigma}_{t_j}S(\theta; t_j, x_i + \bar{\sigma}_{t_j}\xi_{ij}) + \xi_{ij}\| \ge \|\xi_{ij}\| - \|\bar{\sigma}_{t_j}S(\theta; t_j, x_i + \bar{\sigma}_{t_j}\xi_{ij})\|$$
$$\ge \|\xi_{ij}\| - \sqrt{d}M_0\bar{\sigma}_{t_j}$$
$$\ge \|\xi_{ij}\| - \epsilon_1.$$

For 2, by the positive homogeniety of ReLU,

$$S(\theta; t_j, x_i + \bar{\sigma}_{t_j}\xi_{ij}) = \bar{\sigma}_{t_j}S\left(\theta; t_j, \frac{x_i}{\bar{\sigma}_{t_j}} + \xi_{ij}\right).$$

Consider $\bar{\sigma}_{t_j} \ge M$, for some $M > 0$. Then

$$\|\bar{\sigma}_{t_j}S(\theta; t_j, x_i + \bar{\sigma}_{t_j}\xi_{ij}) + \xi_{ij}\| = \left\|\bar{\sigma}_{t_j}^2 S\left(\theta; t_j, \frac{x_i}{\bar{\sigma}_{t_j}} + \xi_{ij}\right) + \xi_{ij}\right\|$$
$$\ge \left\|\bar{\sigma}_{t_j}^2 S\left(\theta; t_j, \frac{x_i}{\bar{\sigma}_{t_j}} + \xi_{ij}\right)\right\| - \|\xi_{ij}\|$$
$$\ge M^2 \left\|S\left(\theta; t_j, \frac{x_i}{\bar{\sigma}_{t_j}} + \xi_{ij}\right)\right\| - \|\xi_{ij}\|. \qquad (46)$$

For any $y \in \mathcal{D}(M)$, where $\mathcal{D}(M) = \{y \in \mathbb{R}^d : y_s \in [(\xi_{ij})_s - |(x_i)_s|/M, (\xi_{ij})_s + |(x_i)_s|/M], \forall\, s = 1, \cdots, d\}$,

$$\|S(\theta; t_j, y)\| \ge \|S(\theta; t_j, \xi_{ij})\| - \|S(\theta; t_j, y) - S(\theta; t_j, \xi_{ij})\|. \qquad (47)$$

Since $S$ is differentiable a.e., by the fundamental theorem of calculus,

$$S(\theta; t_j, y) - S(\theta; t_j, \xi_{ij}) = \int_{\xi_{ij}}^{y} S_x'(\theta; t_j, x)dx.$$

Then

$$\|S(\theta; t_j, y) - S(\theta; t_j, \xi_{ij})\| \le \frac{1}{M} \cdot M_1, \qquad (48)$$

where $M_1 = \max_s (x_i)_s \cdot \operatorname{ess\,sup}_{x \in \mathcal{D}(M_2)} \|S_x'(\theta; t_j, x)\| < +\infty$ for some fixed $0 < M_2 < M$.

Combining (46),(47), and (48), we have

$$\|\bar{\sigma}_{t_j}S(\theta; t_j, x_i + \bar{\sigma}_{t_j}\xi_{ij}) + \xi_{ij}\| \ge M^2 \left(\|S(\theta; t_j, \xi_{ij})\| - \frac{M_1}{M}\right) - \|\xi_{ij}\|$$
$$= M^2 \left(\|S(\theta; t_j, \xi_{ij})\| - \frac{M_1}{M} - \frac{\|\xi_{ij}\|}{M^2}\right).$$

Then $\forall\, \epsilon_2 > 0$, there exists $M = \max\left\{\frac{M_1 + \sqrt{M_1^2 + 4\epsilon_2\|\xi_{ij}\|}}{2\epsilon_2}, M_2\right\} > 0$, s.t., when $\bar{\sigma}_{t_j} > M$, we have

$$\|\bar{\sigma}_{t_j}S(\theta; t_j, x_i + \bar{\sigma}_{t_j}\xi_{ij}) + \xi_{ij}\| \ge M^2(\|S(\theta; t_j, \xi_{ij})\| - \epsilon_2).$$

$\qquad\qquad\square$

## H.2 Proof of optimal rate

*Proof of Corollary 2.* If $|f(\theta^{(k)}; i, j) - f(\theta^{(k)}; l, s)| \le \epsilon$ for all $i, j, l, s$ and $k > K$, then by Lemma 1 and 7, we choose the maximum $f(\theta^{(k)}; i, j)$ for the lower bound, which is of order $\mathcal{O}(\epsilon)$ away from the other $f(\theta^{(k)}; i, j)'s$. Therefore, we can take $j^*(k) = \arg\max_j f(\theta^{(k)}; i, j)$ and absorb the $\mathcal{O}(\epsilon)$ error in constant factors. Then the result naturally follows. $\qquad\square$

## H.3 Proof of comparisons of $E_S$

Recall that the training objective of EDM is defined in the following

$$\mathbb{E}_{\bar{\sigma} \sim p_{\text{train}}} \mathbb{E}_{y,n} \lambda(\bar{\sigma}) \| D_\theta(y + n; \bar{\sigma}) - y \|^2 = \frac{1}{Z_1} \int \frac{1}{\bar{\sigma}} e^{-\frac{(\log \bar{\sigma} - P_{\text{mean}})^2}{2P_{\text{std}}^2}} \cdot \frac{\bar{\sigma}^2 + \sigma_{\text{data}}^2}{\bar{\sigma}^2 \sigma_{\text{data}}^2} \cdot \bar{\sigma}^2 \mathbb{E}_{X_0, \xi} \| \bar{\sigma} s(\theta; t, X_t) + \xi \|^2 \, d\bar{\sigma}.$$

Let $\beta_j = C_1 \beta_{\text{EDM}}$, i.e.,

$$\frac{w(t_j)}{\bar{\sigma}_{t_j}}(t_j - t_{j-1}) = C_1 \cdot e^{-\frac{(\log \bar{\sigma}_{t_j} - P_{\text{mean}})^2}{2P_{\text{std}}^2}} \cdot \frac{\bar{\sigma}_{t_j}^2 + \sigma_{\text{data}}^2}{\bar{\sigma}_{t_j}^2 \sigma_{\text{data}}^2} \cdot \bar{\sigma}_{t_j}$$

$$w(t_j) = C_1 \cdot \frac{\bar{\sigma}_{t_j}}{t_j - t_{j-1}} \cdot e^{-\frac{(\log \bar{\sigma}_{t_j} - P_{\text{mean}})^2}{2P_{\text{std}}^2}} \cdot \frac{\bar{\sigma}_{t_j}^2 + \sigma_{\text{data}}^2}{\bar{\sigma}_{t_j}^2 \sigma_{\text{data}}^2} \cdot \bar{\sigma}_{t_j}$$

**EDM.** Consider $\bar{\sigma}_t = t$ and $t_j = \left( \bar{\sigma}_{\max}^{1/\rho} - (\bar{\sigma}_{\max}^{1/\rho} - \bar{\sigma}_{\min}^{1/\rho}) \frac{N-j}{N} \right)^\rho$ for $j = 0, \cdots, N$. Then

$$w(t_j) = C_1 \cdot \frac{t_j}{t_j - t_{j-1}} \cdot e^{-\frac{(\log t_k - P_{\text{mean}})^2}{2P_{\text{std}}^2}} \cdot \frac{t_j^2 + \sigma_{\text{data}}^2}{t_j \sigma_{\text{data}}^2}$$

$$= C_1 \cdot \frac{1}{\left( \bar{\sigma}_{\max}^{1/\rho} - (\bar{\sigma}_{\max}^{1/\rho} - \bar{\sigma}_{\min}^{1/\rho}) \frac{N-j}{N} \right)^\rho - \left( \bar{\sigma}_{\max}^{1/\rho} - (\bar{\sigma}_{\max}^{1/\rho} - \bar{\sigma}_{\min}^{1/\rho}) \frac{N-j+1}{N} \right)^\rho}$$

$$\cdot e^{-\frac{\left( \log \left( \bar{\sigma}_{\max}^{1/\rho} - (\bar{\sigma}_{\max}^{1/\rho} - \bar{\sigma}_{\min}^{1/\rho}) \frac{N-j}{N} \right)^\rho - P_{\text{mean}} \right)^2}{2P_{\text{std}}^2}} \cdot \frac{\left( \bar{\sigma}_{\max}^{1/\rho} - (\bar{\sigma}_{\max}^{1/\rho} - \bar{\sigma}_{\min}^{1/\rho}) \frac{N-j}{N} \right)^{2\rho} + \sigma_{\text{data}}^2}{\sigma_{\text{data}}^2}$$

Then the maximum of $\frac{\sigma_{t_j}^2}{w(t_j)} = \frac{t_j}{w(t_j)}$ appears at $j = N$

$$\max_j \frac{\sigma_{t_j}^2}{w(t_j)} = \max_j \frac{t_j}{w(t_j)} = \frac{\bar{\sigma}_{\max} \sigma_{\text{data}}^2 e^{\frac{(P_{\text{mean}} - \log \bar{\sigma}_{\max})^2}{2P_{\text{std}}^2}}}{C_1 (\bar{\sigma}_{\max}^2 + \sigma_{\text{data}}^2)} \cdot \left( \bar{\sigma}_{\max} - \left( \bar{\sigma}_{\max}^{1/\rho} - \frac{\bar{\sigma}_{\max}^{1/\rho} - \bar{\sigma}_{\min}^{1/\rho}}{N} \right)^\rho \right)$$

**Song et al. [46].** Consider $\bar{\sigma}_t = \sqrt{t}$ and $t_j = \bar{\sigma}_{\max}^2 \left( \frac{\bar{\sigma}_{\min}^2}{\bar{\sigma}_{\max}^2} \right)^{\frac{N-j}{N}}$ for $j = 0, \cdots, N$. Then

$$w(t_j) = C_1 \cdot \frac{\sqrt{t_j}}{t_j - t_{j-1}} \cdot e^{-\frac{(\log \sqrt{t_j} - P_{\text{mean}})^2}{2P_{\text{std}}^2}} \cdot \frac{t_j + \sigma_{\text{data}}^2}{\sqrt{t_j} \sigma_{\text{data}}^2}$$

$$= C_1 \cdot \frac{1}{\bar{\sigma}_{\max}^2 \left( \frac{\bar{\sigma}_{\min}^2}{\bar{\sigma}_{\max}^2} \right)^{\frac{N-j}{N}} - \bar{\sigma}_{\max}^2 \left( \frac{\bar{\sigma}_{\min}^2}{\bar{\sigma}_{\max}^2} \right)^{\frac{N-j+1}{N}}} \cdot e^{-\frac{\left( \log \bar{\sigma}_{\max} \left( \frac{\bar{\sigma}_{\min}}{\bar{\sigma}_{\max}} \right)^{\frac{N-j}{N}} - P_{\text{mean}} \right)^2}{2P_{\text{std}}^2}} \cdot \frac{\bar{\sigma}_{\max}^2 \left( \frac{\bar{\sigma}_{\min}^2}{\bar{\sigma}_{\max}^2} \right)^{\frac{N-j}{N}} + \sigma_{\text{data}}^2}{\sigma_{\text{data}}^2}$$

Then

$$\max_j \frac{\sigma_{t_j}^2}{w(t_j)} = \max_j \frac{1}{2w(t_j)} = \frac{\bar{\sigma}_{\max} \sigma_{\text{data}}^2 e^{\frac{(P_{\text{mean}} - \log \bar{\sigma}_{\max})^2}{2P_{\text{std}}^2}}}{C_1 (\bar{\sigma}_{\max}^2 + \sigma_{\text{data}}^2)} \cdot \frac{1}{2} \left( \bar{\sigma}_{\max} - \bar{\sigma}_{\max} \left( \frac{\bar{\sigma}_{\min}^2}{\bar{\sigma}_{\max}^2} \right)^{1/N} \right)$$

# I Proofs for Section 4.2

## I.1 Proof when $E_I + E_D$ dominates.

Under the EDM choice of variance, $\bar{\sigma}_t = t$ for all $t \in [0, T]$, and study the optimal time schedule when $E_D + E_I$ dominates. First, it follows from Theorem 2 that

$$E_I + E_D \lesssim \frac{\text{m}_2^2}{T^2} + d \sum_{j=0}^{N-1} \frac{\gamma_j^2}{(T - t_j^\leftarrow)^2}$$

$$+ (\text{m}_2^2 + d)\Big( \sum_{T-t_j^\leftarrow \geq 1} \frac{\gamma_j^2}{(T - t_j^\leftarrow)^4} + \frac{\gamma_j^3}{(T - t_j^\leftarrow)^5} + \sum_{T-t_j^\leftarrow < 1} \frac{\gamma_j^2}{(T - t_j^\leftarrow)^2} + \frac{\gamma_j^3}{(T - t_j^\leftarrow)^3} \Big)$$

Based on the above time schedule dependent error bound, we quantify the errors under polynomial time schedule and exponential time schedule.

**Polynomial time schedule.** we consider $T - t_j^\leftarrow = (\delta^{1/a} + (N - j)h)^a$ with $h = \frac{T^{1/a} - \delta^{1/a}}{N}$ and $a > 1$, $\gamma_j = a(\delta^{1/a} + (N - j - \vartheta)h)^{a-1}h$ for some $\vartheta \in (0, 1)$. We have $\gamma_j/h \sim a(T - t_j^\leftarrow)^{\frac{a-1}{a}}$ and

$$E_I + E_D \lesssim \frac{\text{m}_2^2}{T^2} + \frac{da^2 T^{\frac{1}{a}}}{\delta^{\frac{1}{a}} N} + (\text{m}_2^2 + d)\Big( \frac{a^2 T^{\frac{1}{a}}}{\delta^{\frac{1}{a}} N} + \frac{a^3 T^{\frac{2}{a}}}{\delta^{\frac{2}{a}} N^2} \Big)$$

Therefore, to obtain $E_I + E_D \lesssim \varepsilon$, it suffices to require $T = \Theta(\frac{\text{m}_2}{\varepsilon^{1/2}})$ and the iteration complexity

$$N = \Omega\Big( a^2 \Big( \frac{\text{m}_2}{\delta \varepsilon^{\frac{1}{2}}} \Big)^{\frac{1}{a}} \frac{\text{m}_2^2 + d}{\varepsilon} \Big)$$

For fixed $\text{m}_2, \delta$ and $\varepsilon$, optimal value of $a$ that minimizes the iteration complexity $N$ is $a = \frac{1}{2}\ln(\frac{\text{m}_2}{\delta \varepsilon^{1/2}})$. Once we let $\delta = \bar{\sigma}_{\min}$, $T = \bar{\sigma}_{\max} = \Theta(\frac{\text{m}_2}{\varepsilon^{1/2}})$ and $a = \rho$, the iteration complexity is

$$N = \Omega\Big( \frac{\text{m}_2^2 \vee d}{d} \rho^2 \Big( \frac{\bar{\sigma}_{\max}}{\bar{\sigma}_{\min}} \Big)^{1/\rho} \bar{\sigma}_{\max}^2 \Big),$$

and it is easy to see that our theoretical result supports what's empirically observed in EDM that there is an optimal value of $\rho$ that minimizes the FID.

**Exponential time schedule.** we consider $\gamma_j = \kappa(T - t_j^\leftarrow)$ with $\kappa = \frac{\ln(T/\delta)}{N}$, we have

$$E_I + E_D \lesssim \frac{\text{m}_2^2}{T^2} + \frac{d\ln(T/\delta)^2}{N} + (\text{m}_2^2 + d)\Big( \frac{\ln(T/\delta)^2}{N} + \frac{\ln(T/\delta)^3}{N^2} \Big)$$

Therefore, to obtain $E_I + E_D \lesssim \varepsilon$, it suffices to require $T = \Theta(\frac{\text{m}_2}{\varepsilon^{\frac{1}{2}}})$ and the iteration complexity

$$N = \Omega\Big( \frac{\text{m}_2^2 + d}{\varepsilon} \ln\Big( \frac{\text{m}_2}{\delta \varepsilon^{\frac{1}{2}}} \Big)^2 \Big)$$

When $\text{m}_2 \leq O(\sqrt{d})$, the exponential time schedule is asymptotic optimal, hence it is better than the polynomial time schedule when the initilization error and discretization error dominate. Once we let $\delta = \bar{\sigma}_{\min}$, $T = \bar{\sigma}_{\max} = \Theta(\frac{\text{m}_2}{\varepsilon^{1/2}})$, the iteration complexity is

$$N = \Omega\Big( \frac{\text{m}_2^2 \vee d}{d} \ln\Big( \frac{\bar{\sigma}_{\max}}{\bar{\sigma}_{\min}} \Big)^2 \bar{\sigma}_{\max}^2 \Big).$$

Now we adopt the variance schedule in [46], $\bar{\sigma}_t = \sqrt{t}$ for all $t \in [0, T]$, it follows from Theorem 2 that

$$E_I + E_D \lesssim \frac{\text{m}_2^2}{T} + d \sum_{j=0}^{N-1} \frac{\gamma_j^2}{(T - t_j^\leftarrow)^2} + (\text{m}_2^2 + d)\Big( \sum_{T-t_j^\leftarrow \geq 1} \frac{\gamma_j^2}{(T - t_j^\leftarrow)^3} + \sum_{T-t_j^\leftarrow < 1} \frac{\gamma_j^2}{(T - t_j^\leftarrow)^2} \Big)$$

**Polynomial time schedule.** we consider $T - t_j^\leftarrow = (\delta^{1/a} + (N - j)h)^a$ with $h = \frac{T^{1/a} - \delta^{1/a}}{N}$ and $a > 1$, $\gamma_j = a(\delta^{1/a} + (N - j - \vartheta)h)^{a-1}h$ for some $\vartheta \in (0, 1)$. We have $\gamma_j/h \sim a(T - t_j^\leftarrow)^{\frac{a-1}{a}}$ and

$$E_I + E_D \lesssim \frac{\text{m}_2^2}{T} + \frac{da^2 T^{\frac{1}{a}}}{\delta^{\frac{1}{a}} N} + (\text{m}_2^2 + d)\frac{a^2 T^{\frac{1}{a}}}{\delta^{\frac{1}{a}} N}$$

Therefore, to obtain $E_I + E_D \lesssim \varepsilon$, it suffices to require $T = \Theta(\frac{\mathrm{m}_2^2}{\varepsilon})$ and the iteration complexity

$$N = \Omega\left(a^2 \left(\frac{\mathrm{m}_2^2}{\delta\varepsilon}\right)^{\frac{1}{a}} \frac{\mathrm{m}_2^2 + d}{\varepsilon}\right)$$

Once we let $\delta = \bar\sigma_{\min}^2$, $T = \bar\sigma_{\max}^2 = \Theta(\frac{\mathrm{m}_2^2}{\varepsilon})$ and $a = \rho$, the iteration complexity is

$$N = \Omega\left(\frac{\mathrm{m}_2^2 \vee d}{d} \rho^2 \left(\frac{\bar\sigma_{\max}}{\bar\sigma_{\min}}\right)^{2/\rho} \bar\sigma_{\max}^2\right).$$

Compared to exponential time schedule with the EDM choice of variance schedule, this iteration complexity is worse up to a factor $\left(\frac{\bar\sigma_{\max}}{\bar\sigma_{\min}}\right)^{1/\rho}$.

**Exponential time schedule.** we consider $\gamma_j = \kappa(T - t_j^\leftarrow)$ with $\kappa = \frac{\ln(T/\delta)}{N}$, we have

$$E_I + E_D \lesssim \frac{\mathrm{m}_2^2}{T} + \frac{d\ln(T/\delta)^2}{N} + (\mathrm{m}_2^2 + d)\frac{\ln(T/\delta)^2}{N}$$

Therefore, to obtain $E_I + E_D \lesssim \varepsilon$, it suffices to require $T = \Theta(\frac{\mathrm{m}_2^2}{\varepsilon})$ and the iteration complexity

$$N = \Omega\left(\frac{\mathrm{m}_2^2 + d}{\varepsilon} \ln\left(\frac{\mathrm{m}_2^2}{\delta\varepsilon}\right)^2\right)$$

Once we let $\delta = \bar\sigma_{\min}^2$, $T = \bar\sigma_{\max}^2 = \Theta(\frac{\mathrm{m}_2^2}{\varepsilon})$ and $a = \rho$, the iteration complexity is

$$N = \Omega\left(\frac{\mathrm{m}_2^2 \vee d}{d} \ln\left(\frac{\bar\sigma_{\max}}{\bar\sigma_{\min}}\right)^2 \bar\sigma_{\max}^2\right).$$

Compared to exponential time schedule with the EDM choice of variance schedule, this iteration complexity has the same dependence on dimension parameters $\mathrm{m}_2, d$ and the minimal/maximal variance $\bar\sigma_{\min}, \bar\sigma_{\max}$.

**Optimality of Exponential time schedule.** For simplicity, we assume $\mathrm{m}_2^2 = \mathcal{O}(d)$. Then under both schedules in [30] and [46], $E_I$s only dependent on $T$, and are independent of the time schedule. Both $E_D$s satisfy

$$E_D \lesssim d \sum_{j=0}^{N-1} \frac{\gamma_j^2}{(T - t_j^\leftarrow)^2} \lesssim \varepsilon$$

Let $\tau_j = \ln\left(\frac{T - t_j^\leftarrow}{T - t_{j+1}^\leftarrow}\right) \in (0, \infty)$. Then $\frac{\gamma_j}{T - t_j^\leftarrow} = 1 - e^{-\tau_j}$ and $\sum_{\delta < T - t_j^\leftarrow < T} \tau_j = \ln(T/\delta)$ is fixed. Since $x \mapsto (1 - e^{-x})^2$ is convex on the domain $x \in (0, \infty)$, according the Jensen's inequality, $\sum_{\delta < T - t_j^\leftarrow < T} \frac{\gamma_j^2}{(T - t_j^\leftarrow)^2}$ reaches its minimum when $\tau_j$ are constant-valued for all $j$, which implies the exponential schedule is optimal to minimize $E_D$, hence optimal to minimize $E_D + E_I$.

