# OpenReview forum: "Evaluating the design space of diffusion-based generative models"
_NeurIPS.cc/2024/Conference — NeurIPS 2024 poster_

### Official Review · Reviewer_1csf · 2024-06-16

**Soundness:** 3
**Presentation:** 1
**Contribution:** 2
**Rating:** 5
**Confidence:** 3

**Summary:**

This paper analyzes both the training and sampling errors of diffusion models. The analysis sheds light on two practically relevant design choices, which are the noise level weighting (during training) and discretization (during sampling).

**Strengths:**

- Figure 1 presents a clear qualitative picture of how to choose noise level schedules for training and sampling. In particular, this is the first theoretical work (to the best of my knowledge) to show how the sampling discretization schedule should be adapted depending on score errors arising because of imperfect training.
- The sampling theoretical analysis is thorough and is very up-to-date with the sampling convergence literature. It extends previous results from the variance-preserving to the variance-exploding case, but the extension seems straightforward.
- I am however not knowledgeable enough about the optimization literature to speak of the novelty of the training error results.

**Weaknesses:**

All the theoretical results are very hard to even parse, let alone extract intuition or guidance for practice. The authors argue that their results provide justification for the design choices of Karras et al. [31], but it seems to me that they are shoehorned to fit the chosen narrative. In other words, it is not clear to me that the exact same error bounds could not be used to justify any other schedule used in practice. I find Section 4.1 in particular to be very handwavy. I appreciate the attempt to connect theoretical results to practical choices, but I think the connection is too tenuous here.

Some claims made in the text seem unsubstantiated to me:
- The authors claim that the NTK can only be analyzed in very restricted settings (lines 103-104). I don't see why this is true, as the NTK applies to all architectures in the lazy regime.
- I don't see how Theorem 1 implies that GD has an exponential convergence (lines 223-224) as product factors depend on $s$.
- I don't see how Theorem 2 implies that the sampling error is almost linear in $d$ as $\gamma_k$ and $\sigma_t$ may depend on $d$.

Minor:
- line 140 and eq (4): $C$ should be $C_t$ as it depends on $t$
- line 150: aspectS
- line 154: by A deep ReLU
- line 176: introduce $W$ before line 174
- line 364: we maintain a fixed ~the~ total weighting

**Questions:**

- I don't understand the sentence in lines 296-299, which mentions empirical and population losses even though the paper does not tackle the generalization error. Also, I don't see how the central limit theorem applies here, as individual terms in the empirical average over the training set are not independent since the network parameters themselves depend on the training set. Could the authors please clarify?
- Do the authors believe that their work can inform practical decisions? If yes, what new predictions can be made using their theoretical results? If no (which does not necessarily imply the results are not significant), I think the paper should be more honest about this.

**Limitations:**

Contrary to what is stated in the checklist, limitations are not discussed in the paper. There is no discussion section.

- A first limitation is that only the approximation and optimization errors are studied, and generalization is not tackled. This means that the score network learns the empirical distribution of the training set, and thus reproduces training data points during sampling. This is only briefly mentioned in passing in the middle of the text but should be stated in the introduction (and potentially discussion).

- Some assumptions are questionable. First, the first and last layer are not trained and left to their random initialization. Second, and more importantly, the chosen asymptotic setting corresponds to extremely wide networks trained on very few data points (probably so that they can memorize their training set). This thus corresponds to a rather unrealistic setting.

All in all, I am divided about this paper. On the one hand, it provides a unified analysis of both training and sampling errors, which opens up understanding how to adapt the discretization schedule to score errors. On the other hand, independently of the mathematical correctness of the results (which were not checked), they seem very artificial (due to the restricted setting of the analysis in section 3, and the very qualitative arguments in section 4), and not clearly presented at all. It is thus not clear to me how they can be useful to the theory and practice communities. I recommend rejection, but I am willing to change my decision if the authors convince me that my assessment is incorrect.

---

> ### Author Rebuttal · Authors · 2024-08-07
>
> We greatly thank the reviewer for all the helpful comments. Sorry for confusions created by our initial submission. We hope the itemized responses below, can do a better job in making things clearer, and will further accomplish this goal in a revision.
>
> > All the theoretical results are very hard to even parse, let alone extract intuition or guidance for practice. The authors argue that their results provide justification for the design choices of Karras et al. [31], but it seems to me that they are shoehorned to fit the chosen narrative. In other words, it is not clear to me that the exact same error bounds could not be used to justify any other schedule used in practice. I find Section 4.1 in particular to be very handwavy. I appreciate the attempt to connect theoretical results to practical choices, but I think the connection is too tenuous here.
>
>
> We appreciate an opportunity of explanation. Our biggest contribution is the analysis of the score training process. This is in contrast to the sampling process, which, as pointed out, is by now already much better understood thanks to an entire community. Our 2nd biggest contribution, in our opinion, is the general error bound obtained by fusing both training and sampling, and it applies to all design choices. Due to the complexity of deep learning where data, architecture, nonconvex optimization, and schedules are all mingled together, the theoretical result is indeed complicated, and it might have to, if an informative rate is desired.
>
> But like the reviewer pointed out, the theoretical result may be indeed difficult to parse due to being fully *quantitative* (it could also be due to our presentation, which we now improved significantly). Understanding that, we thus also tried to illustrate some of its *qualitative* implication by connecting to the practice, whose difficulty we appreciate that the reviewer acknowledged. To that end, we simply would like to understand one thing, namely whether/why the design choice of Karras et al. [31] is good, largely due to its impressive performance and hence popularity. And our theory indeed helped us obtain answers, which can be considered as our no.3 contribution. Regarding "it is not clear to me that the exact same error bounds could not be used to justify any other schedule used in practice", our bounds indeed can be used to justify other schedules as well, but we were not able to pack analyses of more schedules due to page limit.
>
>
>
>
>
>
> > The authors claim that the NTK can only be analyzed in very restricted settings (lines 103-104). I don't see why this is true, as the NTK applies to all architectures in the lazy regime.
>
> Apology for the confusion. We feel the inconsistent understandings between us and the expert reviewer might be due to different definitions of NTK. The reviewer's interpretation of "NTK" seems to be in a broad sense: any models using infinitesimal learning rate in the lazy regime can be referred to as the NTK training regime, which is definitely true. In contrast, "NTK" considered in our original version is in a narrow sense following the initial NeurIPS'18 NTK paper, meaning that proof techniques mainly leverage the Gram matrix of the NTK kernel. Based on this definition, current literature only proved the convergence under restricted settings (scalar output, or vector output of a two-layer network with only one layer trained) due to  possibile difficulty lying in the structure of the Gram matrix. Methodologies other than the kernel technique, as discussed and used in this paper, can also reproduce many (narrow-sense) NTK results, but we would like to be precise in citing these literature and distinguishing lazy training from "NTK". More clarifications are already made in a revised version, and we appreciate the reviewer points out.
>
> > I don't see how Theorem 1 implies that GD has an exponential convergence (lines 223-224) as product factors depend on $s$.
>
> Thanks for an expert question. This $s$ dependence indicates that at each step $s$, the decay ratio between the next step and the current step can be different. However, such ratio has a uniform bound when choosing $j^*(s)= j^*=\min_j w(t_j)(t_j-t_{j-1})\bar{\sigma}\_{t_j}$, which does not depend on $s$. Therefore, the product term $\prod_{s=0}^k\left(1-C_6hw(t_{j^*(s)})(t_{j^*(s)}-t_{j^*(s)-1})\bar{\sigma}\_{t_{j^*(s)}}(\frac{m\sqrt{\log d}d^{2a-1.5-4c}}{n^2N})\right)$ is further upper bounded by $\left(1-C_6hw(t_{j^*})(t_{j^*}-t_{j^*-1})\bar{\sigma}\_{t_{j^*}}(\frac{m\sqrt{\log d}d^{2a-1.5-4c}}{n^2N})\right)^{k+1}$. Please kindly see lines 225-226 for similar illustration.
>
> > I don't see how Theorem 2 implies that the sampling error is almost linear in $d$ as $\gamma_k$ and $\sigma_t$ may depend on $d$.
>
> Theorem 2 presents a convergence result of $\text{KL}(p_\delta|q_{t_N})$ for general $\sigma_t$ and $\gamma_k$. In practice, $\sigma_t$ is usually chosen to be a function of $t$, independent of $d$, see [31, 52]. Once we determine what $\sigma_t$ to use, under the exponenital time schedule, the $d$-dependence in the sampling error is always of the form
> $$
> E_I+E_D \lesssim d \frac{\log (T/\delta)^{k+1} }{N^k},\quad k=1,2.
> $$
> Even though $T$ is chosen to be of order poly($d$), the overall sampling error is almost linear in $d$. Please refer to Appendix F.1 in the paper for a detailed discussion.
>
> > typos
>
> Sincerely appreciated. They will of course be corrected.
>
> --------
> **We would like to kindly direct the reviewer to "Comment" for the response to the rest of the review.** We sincerely apologize for exceeding the character limit but we eagerly hope to thoroughly address the reviewer's concerns.

---

> ### Author Response · Authors · 2024-08-07
> **Continued response to the reviewer**
>
> > Q1. I don't understand the sentence in lines 296-299, which mentions empirical and population losses even though the paper does not tackle the generalization error. Also, I don't see how the central limit theorem applies here, as individual terms in the empirical average over the training set are not independent since the network parameters themselves depend on the training set. Could the authors please clarify?
>
> We deeply apologize for the confusions. Yes, this paper focuses on optimization and sampling errors and does not tackle generalization error. The reason we mention the population loss and empirical loss is the following. In training, we need to optimize the empirical loss. However, when estimating the whole generation error, the population loss is necessary (see Theorem 2, the term $E_S$). Therefore, the gap between the population and empirical versions is integrated in the full error analysis (Corollary 1). We will further clarify this point in our next version.
>
> For the $\epsilon_n$, we sincerely apologize; it was our mistake, and we forgot to correct it. The "central limit theorem" part should be the law of large numbers, i.e., this $\epsilon_n$ is the statistical error that converges to 0 as $n\to \infty$. We currently do not have refined statistical error estimation, and this limitation will be clarified in a revision.
>
> > Q2. Do the authors believe that their work can inform practical decisions? If yes, what new predictions can be made using their theoretical results? If no (which does not necessarily imply the results are not significant), I think the paper should be more honest about this.
>
>
>
> Thank you for a very important question. We focused a general theory and did not explore all possible ramifications, but here is one: when the score is very well trained, it is more preferrable to use the schedules in [Song et al. 2020] to sample/generate; otherwise, the design in [Karras et al. 2022] is more preferable.
>
> This important point is already clarified in a revision. However, if the reviewer wants us to state in the limitation section that we presented a general theory but did not explore enough practical implications, we are happy to do so.
>
>
>
>
>
> > Contrary to what is stated in the checklist, limitations are not discussed in the paper. There is no discussion section.
> > * A first limitation is that only the approximation and optimization errors are studied, and generalization is not tackled. This means that the score network learns the empirical distribution of the training set, and thus reproduces training data points during sampling. This is only briefly mentioned in passing in the middle of the text but should be stated in the introduction (and potentially discussion).
>
> We sincerely apologize for not clarifying it in a separate section. In our next version, we will add a discussion section and state clearly that we do not tackle the generalization in the introduction.
>
> > * Some assumptions are questionable. First, the first and last layer are not trained and left to their random initialization. Second, and more importantly, the chosen asymptotic setting corresponds to extremely wide networks trained on very few data points (probably so that they can memorize their training set). This thus corresponds to a rather unrealistic setting.
>
> We would like to thank the reviewer for this comments.
>
> First, the assumption that "first and last layer are not trained" is equivalent to applying fixed linear transforms respectively to the input and the output of the network. This will not affect the trainability of the model if the depth $L$ and width $m$ are properly chosen. Also, it is a commonly used assumption in theoretical analysis (see e.g. [2,3]; also lines 179-180 in the paper).
>
> Second, we agree with the reviewer that "wide networks trained on very few data points" is a little bit "unrealistic". However, this is the limitation of all the current theoretical works about the convergence of neural networks due to technical difficulties. Existing works (e.g. [1]) are making efforts to get closer to practical settings but the progress is very slow. If there are important references we missed, we'd deeply appreciate it; otherwise, we sincerely hope that the reviewer could consider if we could agree about the current stage of theoretical research, and allow us to explore in a sustainable way (a 3:Reject adds one straw instead; is our work really that bad, especially in the context of the existing literature?)
>
>
>
> [1] Zou et al. An improved analysis of training over-parameterized deep neural networks. NeurIPS 2019.
>
> [2] Allen-Zhu et al. A convergence theory for deep learning via over-parameterization. ICML 2019
>
> [3] Cai et al. Neural temporal-difference learning converges to global optima. NeurIPS 2019.

---

> > ### Comment · Reviewer_1csf · 2024-08-07
> >
> > I thank the authors for their very detailed answer. This fully addresses my concerns about the validity of the results. After reading the other reviews and rebuttals, I am willing to increase my score.
> >
> > The main point that is still not clear for me is what we learn from the results in this paper.
> >
> > - This first contribution is the quantitative analysis of the learning and sampling errors. While sampling is well understood, learning is not, and the analysis in the paper suffers from the same limitations as the rest of the literature (although relaxing the very-wide assumption in the new theorem 1 above seems to be a major advance?). I suppose the main contribution lies in the novel gradient lower bound, which is relegated to the appendix, but that readers more expert than this reviewer may read and appreciate.
> >    * As a side remark, if theoretical progress is slow, then should we keep pushing in the same direction and keep submitting results with the same limitations but slightly more general assumptions every conference cycle? Or rather try to tackle these fundamental limitations with new approaches? I'm not sure that the first option is the most efficient way to drive progress in theoretical research. But this is more of a systematic issue of the field and beside the point here.
> >
> > - The second contribution is qualitative analysis of noise schedules in the literature. As the authors pointed out, this paper predicts that two different sampling schedules should be used based on the quality of the score approximation, which is an interesting hypothesis. My main qualm here still stands: the authors aim for quantitative statements, but it seems that the error analyses leave a lot of wiggle room in the epsilons and deltas to fit any desired schedule. To be clear, I see this as a negative point: it does not tell us which schedule is the best, and not even how to evaluate a given schedule beyond saying that it should have a few qualitative features. I am all in favor of qualitative analyses, which I are usually underrated in theoretical papers, but I think they should be _simple_. It would greatly strengthen the papers if the authors studied a simplified setting where the error bounds take a simple form, and it can be clearly seen by a non-expert reader what features of a sampling discretization matter. In its current state, the paper does not give _intuition_ about the problem, and readers have to trust the authors that their qualitative results are an accurate description of typical phenomena.

---

> ### Author Response · Authors · 2024-08-10
> **Response to reviewer comments**
>
> We sincerely thank the reviewer for carefully reading our rebuttal and being willing to increase the score.
>
> > This first contribution...
>
> Thank you for the comment and apologize for potential confusions. The main goal of this work is not to revamp the general methodology for analyzing neural network training, but rather to adapt existing tools to obtain a full theoretical guarantee of diffusion model's generation capability. The former is a holy grail, but the latter, we believe, is also rather important. Nevertheless, although the sampling process of diffusion model has been analyzed for ~2 years already and relatively well understood, it is only half of the generation pipeline, and no result existed that combines it with the other half, namely the score training. In fact, denoising score matching training has some significant difference from settings in which existing training analyses can directly apply, and we suspect this is why this problem has not been already solved before. Please allow us to elaborate:
>
> For the denoising score matching problem, there are a lot of special properties and additional issues. Specifically, the input and output data in the denoising score matching problem have specific structures such as:
>
> 1) The input data $X_{ij}=x_i+\bar{\sigma}\_{t_j}\xi_{ij}$ is noisy and therefore the whole model is non-interpolation, which is in contrast to the interpolation assumption in existing techniques.
> 2) The scales of the input and output data cannot be freely assumed in the theory. More precisely, the variances of the noise $\bar{\sigma}\_{t_j}$ in the input and output data $-\xi_{ij}/\bar{\sigma}\_{t_j}$ changes as $\bar{\sigma}\_{t_j}$ varies. Moreover, to favor the sampling process after training the score, the scale of input data needs to be of order $\sqrt{d}$, and the scale of outputs is at least of order $\sqrt{d}$. This is in sharp contrast to data assumptions in existing literatures. For example, in Allen-Zhu et al. [1], they assume the output data to be $o(1)$ and the input data to have order 1 scale.
>
>
> Therefore, while still leveraging the framework of existing theoretical analyses, we had to develop additional techniques to deal with the above issues. For 1, we decomposed the model into an interpolation component and a non-interpolation one in order to apply the existing techniques on the interpolation part. For 2, due to the special data structure and scalings, existing techniques, which rely on a commonly used data separability assumption, no longer suffice. Instead, we develop a new geometric technique to overcome this issue, and the only price to pay is that the order of the dimension of the input data $d$ should not be too small, which has now been significantly relaxed in the new version as stated in the global rebuttal.
>
> Hopefully our goal and contributions are better clarified by the above discussions.
>
> > The second contribution...
>
> We apologize for the confusion caused. Our full error analysis is completely quantitative, and we will now explain why our presentation on preferred schedule(s) is qualitative. This explanation could start with a question: why can't we just substitute the hyperparameter choices of a specific schedule in our error bound, get a concrete number, and then compare that number with other schedules' numbers, or even optimize that number to get better schedules?
>
> That is because our full error bound is a function of not only hyperparameters provided by a schedule, but also many other things (i.e. what the reviewer mentioned as "a lot of wiggle room"). For example, a) there is a dependence on the data distribution, which is reflected in a generalization gap term in the error bound; b) there is dependence on the network structure (for our setting, width $m$ and depth $L$); c) there is dependence on optimization hyperparameters. If all these were known, then a number could indeed be spit out, but then, we feel, it becomes a bit less interesting because we are no longer comparing general schedules per se.
>
> Therefore, we don't compare precise values, and instead adopt some qualitative arguments, so that some generic yet practical guides can still be provided. More precisely, we consider the following two general situations: $E_S\gg E_I+E_D$ and $E_S\ll E_I+E_D$.  Then we substitute the two specific time and variance schedules into the error bound, and compare either the values or the complexities of the time steps under the same error as is shown in Table 2. The schedule with smaller value or complexity will be the preferable choice. The results in Table 2 may appear informal; however, they are indeed rigorously calculated in Appendix E3 and F (see lines 844,846,853-875), and could be formalized into corollaries or propositions. Here we chose not to present the full details in Table 2 merely due to the page limit. We will clarify this part in a revision.

---

> > ### Comment · Reviewer_1csf · 2024-08-13
> >
> > Thank you for the clarifications. I have increased my score.

---

### Official Review · Reviewer_RRb9 · 2024-07-08

**Soundness:** 4
**Presentation:** 2
**Contribution:** 3
**Rating:** 6
**Confidence:** 4

**Summary:**

This work develops theoretical bounds on the training error and sample complexity of score-based diffusion models, which further imply a complete convergence analysis (by combining the both). In addition, the derived bounds shed light on the efficient hyper-parameters selection, including re-weighting functions, time discretization schemes and variance schedules, supporting former empirical works in a principled manner.

**Strengths:**

1. The problem setup is quite general, at least at the current stage.
2. This work provides a comprehensive and rigorous convergence analysis of diffusion models for *both* training and sampling. The analyzed score network also extends former theoretical literature.
3. The obtained mathematical bounds also imply beneficial insights to guide the hyper-parameters selection, which coincides with other experimental literature .

**Weaknesses:**

1. The paper is organized in a almost technical way, and some details are not easy to follow. It would be helpful to improve the readability by adding e.g., (key) notations collections and formulation illustrations. Also, there is no conclusion section.
2. The derived bounds in main theorems are too technically complex. Is it possible to simplify them to summarize the core algebraic dependence on hyper-parameters (including the data dimension $d$, sample size $n$, model capacity $(m,L)$, discretization steps $N$)?
3. There are no experiments to support the theory. Particularly, is it feasible to numerically verify the exponential convergence in Theorem 1?
4. Are there any differences on the training between over-parameterized neural networks applied to normal supervised learning, and here in the generative modeling (as the score network)? Note that this work selects the *same* feed-forward network (without biases) for *all* time steps (see the equation between Line 173 and 174), ignoring the time information in the architecture of score networks.

**Questions:**

1. Please provide more details of questions raised in the weaknesses section above.
2. The scale of model initializations is near-zero (see  Line 181-182), which seems to be one of the typical configurations in the over-parameterization regime (lazy training?). Is this setting reasonable and representative in practice? It would be better to provide (numerical) examples about this, particularly in this generative modeling case.
3. Details 1: In Line 201, "... high probability (see Lemma..)."
4. Details 2: What is the definition of $a$ in Theorem 1 (see $d^{2a-1.5-4c}$)?
5. Details 3: In Proposition 1, there seems to be a confusion on the mixed use of $S$ and $S^*$ (particularly, Point 2).
6. Details 4: In Line 297-299, it states that $\epsilon_n$ (the distance between the empirical loss and population loss at the GD outputs) can be estimated by central limit theorem. Can the authors provide more details on it (since the learned model parameters are *dependent* of data)?
7. Details 5: In Table 1, why is the exponential schedule selected as $\bar{\sigma}_t=\sqrt{t}$? In VE-SDE, $\sigma_t$ is an exponential function of $t$ (and hence $\bar{\sigma}_t^2:=2\int_0^t \sigma_s^2 ds$). Note that in this case, Assumption 2 (Point 1) may not hold.

**Limitations:**

Based on the developed theory, it is worthy to further explore new weighting functions, time discretization schemes and variance schedules to better tune diffusion models.

---

> ### Author Rebuttal · Authors · 2024-08-07
>
> We deeply thank the reviewer for all the helpful comments and sincerely appreciate the positive evaluation. Here are our itemized responses:
>
>
> > W1
>
> We sincerely appreciate your comment and have done a major revision of the paper in accordance.
>
> > W2
>
>
> Thank you for this advice. It is in general hard to simplify the dependency without choosing time and variance schedules. Therefore, in the following, we show the upper bound in our full error analysis under the time and variance schedules of [31] (a current gold standard in practices):
>
> $KL(p_\delta|q_{T-\delta}) \lesssim  \frac{\mathrm{m}_2^2}{T^2} +\frac{d a^2 T^{\frac{1}{a}} }{\delta^{\frac{1}{a}}N}+(\mathrm{m}_2^2+d)( \frac{a^2T^{\frac{1}{a}}}{\delta^{\frac{1}{a}} N}+\frac{a^3T^{\frac{2}{a}}}{\delta^{\frac{2}{a}}N^2})$
> $+\frac{C}{N}(C_1+(1-C_2 h (\frac{md^{\frac{a_0-1}{2}}}{n^3N^2}))^K),$
>
> where $\mathrm{m}_2^2$ is the second moment of the data distribution and it is of order $d$ if the data distribution is Gaussian; $a$ is chosen to be 7 in [31]; $a_0\in(1/2,1)$; $C,C_1,C_2$ are some constants; $K$ is the total number of iterations.
>
> Please see upper bounds of sampling errors under more schedules in Appendix F1. We will try to make it less technical in our next version.
>
>
>
>
>
> > W3
>
> We would like to thank the reviewer for helping us strengthen the paper. The figure in the attached pdf shows how the actual training is consistent with our theory. More precisely, the $y-$axis is in $\log$ scale; the training loss decays in a nonlinear way but it is clearly bounded by the exponential bound.
>
>
>
>
> > W4
>
> Thank you for a very sharp observation. There are indeed time information in the model of our paper compared to the previous overparameterized neural networks.
>
>
> In short, $X_{ij}$ in our theory is actually an augmented version that concatenates both the original $X_{ij}$ and $t_j$. This way we no longer need to consider separately an additional time variable. More precisely, due to the data structure $X_{ij}=x_i+\bar{\sigma}\_{t_j}\xi_{ij}$, if we take $(x_i)\_d=0$ and $(\xi_{ij})\_d=1$, then the last element of this input vector is just $\bar{\sigma}\_{t_j}$, which is a function of time. This can be seen, for example, in Table 2: practical choices of $\bar{\sigma}\_{t_j}$ include $\bar{\sigma}\_{t_j}=t_j$ and $\bar{\sigma}\_{t_j}=\sqrt{t_j}$.
>
> However, we now realize this is a source of confusion. Apology, and it will be clarified in our next version.
>
>
> > Q1
>
> Kindly see the above.
>
> > Q2
>
> Thank you for the questions. Such initialization is indeed what is used in practice. For example, in EDM [31], which is a gold standard for the design of diffusion models, they provide several choices for initialization, including Xavier and Kaiming initializations using both uniform and Gaussian distributions. When initializing the weights by Gaussian, they rescale the standard normal by $\sqrt{\frac{2}{d_{\rm in}+d_{\rm out}}}$ and $\sqrt{\frac{1}{d_{\rm in}}}$ for a weight matrix of dimension $d_{\rm out}\times d_{\rm in}$. In our paper, $d_{\rm out},d_{\rm in}$ equal to $m$ or $d$ for different layers. Therefore, our initialization of all the weight matrices ($\mathcal{N}(0,\frac{2}{m})$ and $\mathcal{N}(0,\frac{1}{d})$) matches the order of at least one of those used in practice.
>
> > Q3
>
> Apologize for the typo. This should be Lemma 4. We will add it in our next version.
>
> > Q4
>
> We deeply appreciate that you carefully checked the details and apologize for this typo and the confusion caused. It was $a=1/2$, but we now further improved our rate and this $a$ is no longer needed.
>
> > Q5
>
> We sincerely apologize for the typo. All the $S^*$ should be just $S$. We have already corrected it in a revised version.
>
> > Q6
>
> We deeply apologize for this confusion and thank the reviewer again for the careful review. This is actually a mistake that we forgot to correct. The "central limit theorem" part should be the law of large numbers, i.e., this $\epsilon_n$ is the statistical error that converges to 0 as $n\to \infty$. We currently do not have refined statistical error estimation, and this limitation will be clarified in a revision.
>
> > Q7
>
>
> The exponential schedule is the popular version from [52]; it is called so because the time schedule $(t_k)$ decays exponentially fast (see the last column of Table 1). Regarding its noise schedule, since its forward SDE is $\mathrm{d}X_t= \sqrt{2}\mathrm{d}W_t$, $\sigma_t=1$. Therefore, the variance $\bar{\sigma}_t^2=t$, which is not exponential. The Assumption 2 Point 1 is satisfied since we can always choose an appropriate scaled weight $w(t)$ in the training step.
>
> > Limitation
>
>
>
> We agree. This is actually nontrivial given the complexity (due to its dependence on many design factors) of our bound, and we hope the reviewer could consider the length and scope of a conference paper and allow us to do it properly in a future work because we'd also like to comprehensively verify everything empirically.
>
> [31] Karras et al. Elucidating the design space of diffusion-based generative models. NeurIPS 2022

---

> > ### Comment · Reviewer_RRb9 · 2024-08-11
> >
> > Thanks for the clarifications by authors. I find most of them adequate, except the following three points:
> >  - Although the current manuscript can not be refined, would you please state the plan of notations collections, formulation illustrations, and the conclusion section?
> > - Regarding the dependence on hyper-parameters of the upper bound, the dependence on model capacity $(m,L)$ and sample size $n$ seems still unclear, in the sense that the upper bound is not related to the network depth $L$, and also independent of the network width $m$ and sample size $n$ when the total GD iteration $K$ goes to infinity (see the answer of W2).
> > - On the estimation error $\epsilon_n$, the analysis is definitely more complex than a trivial application of the law of large numbers, since the data and parameters are decoupled under this setting. Can authors kindly provide at least (promising) ideas to handle this difficulty?

---

> ### Author Response · Authors · 2024-08-12
>
> We sincerely thank the reviewer for discussing with us and helping us improve the quality of our manuscript.
>
> > Although the current manuscript can not be refined, would you please state the plan of notations collections, formulation illustrations, and the conclusion section?
>
> Apologize for not having provided enough plan in the rebuttal. Here it is.
>
> For notation collections, we will add a detailed table in the appendix summarizing all our symbols for readers' easier references.
>
> For the formulation illustrations, we will add more remarks including: 1) more discussions on the special structure of this denoising score matching problem and how it is related and difference from traditional nonlinear regression problems (the former is not interpolating), 2) comparison between our assumptions and results of this model and previous overparameterized neural network models, especially weakened assumption (we do not require data separability as shown in Allen-Zhu et al. [1]), a decomposition of the non-interpolation setting in diffusion models compared to the interpolation ones used in previous models, and improved parameter dependence on the input dimension $d$ in the error bound, 3) better quantitative interpretation of our results under specific schedules with explicit parameter dependences.
>
> The conclusion section will start with the following: We provide a first full error analysis incorporating both optimization and sampling processes. For the training process, we provide a first result under a deep neural network and prove the exponential convergence into a neighborhood of minima, while for sampling, we also extend the current analysis to the variance exploding case. Moreover, based on the full error analysis, we establish quantitative understandings of the error bound under the two schedules. Consequently, we conclude a qualitative illustration of the "bell-shaped" weighting and the choices of schedules under well-trained and less-trained cases.
>
> Then we will discuss limitations and future directions. More precisely, the network architecture we used in the model is a deep ReLU network. Although being so far the most complicated arhitecture for theoretical results, it is still far from what is used in practice like U-Nets and transformers. Regarding the full error analysis, we only focus on the optimization and sampling error, and do not disect the generalization error. When bridging the theoretical results with practical designs of diffusion models, our results are mostly qualitative and we only compare two schedules under well-trained and less-trained cases.
>
> ---
>
> **We apologize for exceeding the character limit and would like to kindly direct the reviewer to our next Comment for the continuation of our reply.**

---

> ### Author Response · Authors · 2024-08-12
> **Continued response to the reviewer**
>
> > Regarding the dependence on hyper-parameters of the upper bound, the dependence on model capacity ($m,L$) and sample size $n$ seems still unclear, in the sense that the upper bound is not related to the network depth $L$, and also independent of the network width $m$ and sample size $n$ when the total GD iteration $K$ goes to infinity (see the answer of W2).
>
> Thank you for the expert questions. Here are technical explanations. In the simplified bound (for better seeing hyper-parameter dependences) that was provided in our last reply, namely
>
> $KL(p_\delta|q_{T-\delta}) \lesssim  \frac{\mathrm{m}_2^2}{T^2} +\frac{d a^2 T^{\frac{1}{a}} }{\delta^{\frac{1}{a}}N}+(\mathrm{m}_2^2+d)( \frac{a^2T^{\frac{1}{a}}}{\delta^{\frac{1}{a}} N}+\frac{a^3T^{\frac{2}{a}}}{\delta^{\frac{2}{a}}N^2})$
> $+\frac{C}{N}(C_1+(1-C_2 h (\frac{md^{\frac{a_0-1}{2}}}{n^3N^2}))^K),$
>
> the constant $C_1$ actually hides many terms:
> $$ C_1=\epsilon_n+\bar{\mathcal{L}}(\theta_\mathcal{F})+|\bar{\mathcal{L}}(\theta_\mathcal{F})+\bar{C}|, $$
> where $\epsilon_{n}=|\bar{\mathcal{L}}(\theta^{(K)})-\bar{\mathcal{L}}\_{\rm em}(\theta^{(K)})|$, $-\bar{C}$ is the true minimum of $\bar{\mathcal{L}}(\theta)$ defined in (5), and $\theta_{\mathcal{F}}=\arg\inf_{\{\theta:S(\theta)\in\mathcal{F}\}} |\bar{\mathcal{L}}(\theta)+\bar{C}|$ with $\mathcal{F}=\big\lbrace$ ReLU network function, with $d=\Omega({\rm{poly}}(\log (nN))), m=\Omega\left(\text{poly}\big(n,N,d,L,T/t_0\big)\right)\big\rbrace$. Then, $\epsilon_n$ is the statistical error, $\bar{\mathcal{L}}(\theta_\mathcal{F})$ is the estimation error, and $|\bar{\mathcal{L}}(\theta_\mathcal{F})+\bar{C}|$ is the approximation error.
>
> Regarding the depth dependence $L$, indeed it is not shown in the rate. However, it was actually reflected in the function approximation error $|\bar{\mathcal{L}}(\theta_\mathcal{F})+\bar{C}|$; for example, if $L$ is too small, this error could be large. We did not analyze the function approximation theory of neural network, which is another profound area. Since that area is extensively studied, one can just leverage the existing results, which will give $L$ dependence.
>
> Regarding the disappearing of $m$ and $n$ dependences as $K\to\infty$, here is how the dependences were hidden: 1) there is a lower bound of the width $m$ in Assumption 1; once $m$ is greater than this bound, i.e. the model is sufficiently overparametrized, it no longer has effect on the optimization error in the infinite $K$ limit. 2) Regarding the sample size $n$, its effect is mainly hidden inside the statistical error $\epsilon_n$. Please kindly see more details in the reply below.
>
> > On the estimation error $\epsilon_n$, the analysis is definitely more complex than a trivial application of the law of large numbers, since the data and parameters are decoupled under this setting. Can authors kindly provide at least (promising) ideas to handle this difficulty?
>
> We sincerely apologize for the confusion caused, which is likely due to our previous typo in the Corollary 1, and we deeply appreciate the opportunity to clarify: we did not intend to establish any estimation of the statistical error $\epsilon_n$, and our updated version actually looks like this:
>
> >> **Theorem.** Under the same conditions as updated Theorem 1 (global rebuttal) with $k=K$ and Theorem 2,
>     we have
>     $$
>         KL(p_\delta|q_{T-\delta}) \lesssim E_I+E_D+\max_k \frac{\sigma^2_{t_{N-k}}}{w(t_{N-k})} \bigg(\epsilon_{\rm{train}}+\epsilon_{n}+\bar{\mathcal{L}}(\theta_\mathcal{F})+|\bar{\mathcal{L}}(\theta_\mathcal{F})+\bar{C}|\bigg)
>     $$
>     where $E_I,E_D$ are defined in Theorem 2, $\epsilon_{\rm train}$ is defined in updated Theorem 1 (global rebuttal), $\epsilon_{n}=|\bar{\mathcal{L}}(\theta^{(K)})-\bar{\mathcal{L}}\_{\rm em}(\theta^{(K)})|$, $\bar{C}$ is defined in (5), and $\theta_{\mathcal{F}}=\arg\inf_{\{\theta:S(\theta)\in\mathcal{F}\}} |\bar{\mathcal{L}}(\theta)+\bar{C}|$ with $\mathcal{F}=\big\lbrace$ReLU network function, with $d=\Omega({\rm{poly}}(\log (nN))), m=\Omega\left(\text{poly}\big(n,N,d,L,T/t_0\big)\right)\big\rbrace$.
>
> We'd deeply appreciate it just in case the reviewer is willing to shed more light on $\epsilon_n$'s estimation. In particular, the reviewer pointed out that the analysis is more complex than a trivial application of the law of large numbers, since the data and parameters are decoupled. May we confirm if "parameters" refer to neural network parameters, or training hyperparameters? If the latter, does the decoupling matter? If the former, we thought after one step of training, the NN parameters become coupled with the data. Or, could "decoupled" be, by any chance, a typo, and "coupled" was intended instead? We thought law of large numbers is actually easier when random variables are independent from each other, and when there are coupling (i.e. correlation) it could be more tricky.
>
>
> ---
> In any case, thank you again for your time and consideration!

---

### Official Review · Reviewer_Ti47 · 2024-07-09

**Soundness:** 3
**Presentation:** 2
**Contribution:** 2
**Rating:** 5
**Confidence:** 4

**Summary:**

This work studies the training and sampling process and then achieves the end-to-end analysis for diffusion models. For the optimization process, this work uses an over-parameterized NN and gradient descent to prove the convergence rate. For the sampling process, this work provides VESDE results and explains the great performance of SOTA VESDE. After these results, this work explains the “bell shaped” weight function used in application from the theoretical perspective.

**Strengths:**

1.	This work is the first one to analyze the optimization process with a deep NN and prove the exponential convergence into a neighborhood of minima.
2.	This work makes the first step to explain the “bell shaped” weight function and the great performance of the state-of-the-art VE-based model from the theoretical perspective.

**Weaknesses:**

Weakness 1: The high-dimensional data assumption is strong. It would be better to discuss the application of high-dimensional data.

Weakness 2: The technique novelty of Theorem 1 is needed to be discussed in detail. Section 3.1 claims that this work develops a new method for obtaining a lower bound of gradient. However, it does not discuss this method on the main page. It would be helpful to discuss the technique novelty in detail.

Weakness 3: As shown in Section 3.1, this work mentions that the data separability assumption (corresponds to parameter $\delta$ in [1]) can not be directly used due to the property of diffusion models. This work replaces $\delta$ with $t_0/T$ when choosing $m$, which seems to be related to the $E_I$ term in Theorem 2. The technique challenge is unclear after replacing $\delta$ with $t_0/T$.

Weakness 4: As shown in [2], their results can be straightforwardly extended to any linear SDE, including VESDE. Hence, it would be better to discuss the technique of Theorem 2.

[1] Allen-Zhu, Z., Li, Y., & Song, Z. (2019, May). A convergence theory for deep learning via over-parameterization. In International conference on machine learning (pp. 242-252). PMLR.

[2] Benton, J., Bortoli, V. D., Doucet, A., & Deligiannidis, G. (2024). Nearly d-linear convergence bounds for diffusion models via stochastic localization.

**Questions:**

Please see the Weakness part.

Question 1. Can you relax the high-dimensional data assumption by using the low-dimensional manifold assumption [3]?

Question 2. The guarantee in [4] is a pure $W_2$ guarantee and can not be directly compared to the $KL$ guarantee in this work. Can you discuss it in detail?

Comment 1: It would be better to add a conclusion and limitation part at the end of this work.

Typo 1: In line 201, the reference Lemma is unclear.

[3] Tang, R., & Yang, Y. (2024, April). Adaptivity of diffusion models to manifold structures. In International Conference on Artificial Intelligence and Statistics (pp. 1648-1656). PMLR.

**Limitations:**

This work does not discuss the limitation and societal impace in a independent paragraph.

---

> ### Author Rebuttal · Authors · 2024-08-07
>
> We greatly thank the reviewer for all the helpful comments. We especially appreciate the comment "...the first one to analyze the optimization process with a deep NN...the first step to explain the “bell shaped” weight...". Please kindly see our itemized replies below:
> >Weakness 1
>
> We agree that this assumption is strong. Therefore, we improved our convergence theory, and the new version now uses a much more relaxed assumption, where the original version $$d=\Theta(m)$$ is replaced by
> $$d=\Omega({\text{poly}}(\log (nN))).$$
>
> Note in the old version, the input dimension is the same order as the width of the network $m$, while in the new version, it only needs $d$ to be not too small, or equivalently, the number of data points $n$ and the number of time points $N$ should not be exponential in $d$. Please also see a new version of Theorem 1 in our global rebuttal.
>
>
>
> > Weakness 2
>
> Thank you for pointing this out.
>
> Roughly speaking, we first decompose the gradient into terms based on one data point that has the largest loss value at the current iteration, and the rest $nN-1$ samples [1]. Then the gradient can be written in the form $w^\top v+w^\top u$ for some $w,v,u$. The main novelty of our proof lies in analyzing the angles between $v,u$. This allows us to show the probability of $w^\top v\ge 0,w^\top u\ge 0$ to be roughly of the order $d^{\frac{-3}{8}}$. Finally, this leads to a lower bound of the gradient by $w^\top v$. The proof is more involved but the above is the main technical innovation.
>
>
>
> > Weakness 3
>
>
> We apologize for a critical confusion: the $\delta$ used in [1] and the $t_0/T$ in our paper serve different purposes, and we did not replace $\delta$ by $t_0/T$.
>
> As the expert reviewer knows, $\delta$ is a lower bound used in the data separability assumption in [1]. However, in our paper, our proof does not require such an assumption due to our new method and the diffusion model settings, and therefore there is no counterpart of this variable. More precisely, in [1], they require the output data to be in $o(1)$; in this case, the input data have to be well separated so that the gradient is locally strongly convex (obtaining the lower bound). In contrast, denoising diffusion models require Gaussian outputs that cannot be small, which help contribute to the strongly convexity and relax the requirement on the input data.
>
>
> Regarding $t_0/T$, it adds no extra technical challenges and is merely due to the weighted $\ell_2$ loss for diffusion models, which depends on different time points and variances, compared to the standard $\ell_2$ loss. Such weighting will enter the constant factors of different bounds and consequently cause the dependency in the width $m$.
>
>
>
> > Weakness 4
>
> Results in [2] indeed can be extended to VESDE, for the sampling part where discretization error is analyzed. However, we respectfully think the detailed calculations still need to be done, and several not-so-straightforward steps we encountered included:
> 1) bounding the initialization error: [2] considered the VPSDE and uses KL-decay along the OU-process which is no longer true for VESDE. Therefore we instead handle the initialization error by the convexity property of KL divergence (Lem.10).
> 2) bounding the discretization error: we considered a general time-schedule, which makes it harder to bound the term $N_2+N_3$ in line 758-759. We utilize the asymptotic estimation $\min(\mathrm{m}_2^2, \bar{\sigma}\_t^2 d)\lesssim(1-e^{-\bar{\sigma}\_t^2})(\mathrm{m}_2^2 +d)$ to obtain the upper bound.
>
> Thanks to the reviewer, we will include more details of these in a revised version.
>
>
>
>
> > Question 1
>
> We deeply appreciate the reviewer for proposing a nice possibility to improve our result and will carefully evaluate it. Meanwhile, as indicated in the reply to Weakness 1, we have already improved our results without additional assumptions.
>
> > Question 2
>
> The reviewer mentioned [4] but didn't give the reference. We are guessing that the reviewer might mean [24 in our manuscript]. $W_2$ is in general not comparable to KL unless if we assume $p_{T-\delta}$ in Thm.2 satisfies Talagrand inequality (implying $W_2^2\le C\cdot\text{KL}$). The comparison to [24] is actually not fair as [24] assumes strongly log-concavity, which is much much stronger than our situation, where the data distribution can be arbitrarily non-log-concave and multimodal. In the updated version, we compare to [6] (Cor.1), whose convergence is established in TV for compact supported data distributions. In this case, we can see our result has better dimension dependence by Pinsker's inequality $2\text{TV}^2\le \text{KL}$.
>
>
> > Comment 1
>
> Thank you very much for this comment. We will add it to our next version.
>
> > typos
>
> We sincerely appreciate!
>
>
>
> **Additional References**
>
> [5] Eldan. Taming correlations through entropy-efficient measure decompositions with applications to mean-field approximation. PTRF'19.
>
> [6] Yang et al. The Convergence of Variance Exploding Diffusion Models under the Manifold Hypothesis.

---

> > ### Comment · Reviewer_Ti47 · 2024-08-12
> >
> > Thanks for your careful response. The significant improvement of the dependence of $d$ is helpful. However, the proof sketch of the improved theorem is not provided. It would be better to provide an intuitive proof sketch, which is necessary to check the correctness.
> >
> > Furthermore, I think it would be better to discuss the limitation in the rebuttal phase and add an independent limitation paragraph in the next version.

---

> ### Author Response · Authors · 2024-08-12
>
> We deeply thank the reviewer for reading our rebuttal and discussing with us.
>
> > The significant improvement of the dependence of  is helpful. However, the proof sketch of the improved theorem is not provided. It would be better to provide an intuitive proof sketch, which is necessary to check the correctness.
>
> The main improvement is on proving the lower bound of the gradient. Therefore, we will provide a proof sketch for it, and the rest of the proof follows the similar framework as Allen-Zhu et al [1].
>
>
> We first decompose the gradient of the $k$th row of $W_L$ $\nabla\_{(W_L)\_k}\bar{\mathcal{L}}\_{\rm em}(\theta)=\underbrace{\frac{1}{n}w(t_{j^*})(t_{j^*}-t_{j^*-1}){(W_{L+1})^k}^\top(\bar{\sigma}\_{t_{j^*}}W_{L+1} q_{i\^\* j\^\*,L}+\xi_{i\^\* j\^\*})q_{i\^\* j\^\*,L-1} {1}\_{(W_L q_{i\^\* j\^\*,L-1})\_k>0}}\_{\nabla_1}$
>     $+\underbrace{\frac{1}{n}\sum_{(i,j)\ne (i^*,j^*)}w(t_j)(t_j-t_{j-1}){(W_{L+1})^k}^\top(\bar{\sigma}\_{t_j}W_{L+1}q_{ij,L}+\xi_{ij})\,q_{ij,L-1} {1}\_{(W_L q_{ij,L-1})\_k>0}}_{\nabla_2}$
>
> where $(i^*,j^*)$ indicates the sample index with the largest loss value.
>
>
> Then we first fix $(q_{ij,L-1})\_s=1$, and prove that the index set of both $(q\_{i^\*j^\*,L})\_s> 0$ and $\sum\_{(i,j)\ne (i^*,j^*)}w(t_j)(t_j-t_{j-1})\bar{\sigma}\_{t_j}{1}
> \_{(W_Lq_{ij,L-1})\_k>0}(q_{ij,L})_s>0$ is order $m$ with high probability.
>
> Next, we conditioned on the index set we've found, then we can decouple each element of $\nabla_{(W_L)_k}\bar{\mathcal{L}}\_{\rm em}$ with high probability. We then prove that with high probability, the event $(\nabla_1)_s> 0$ and $(\nabla_2)_s>0$ has probability at least of order $d^{(a_0-1)/2}$ where $a_0\in(1/2,1)$.
>
> Now, we deal with $(q_{ij,L-1})\_s$ and prove that if the above results hold for $(q_{ij,L-1})_s=1$, then there exists an index set with cardinality of order $m/(nN)$ such that $(\nabla_1)_s> 0$ and $(\nabla_2)_s>0$ also hold in this index set.
>
> In the end, combining all the steps above yields the lower bound.
>
>
> > Furthermore, I think it would be better to discuss the limitation in the rebuttal phase and add an independent limitation paragraph in the next version.
>
> We thank the reviewer for the suggestion. We will add a limitation paragraph to our paper. Below is a quick summary:
>
> The network architecture we used in the model is a deep ReLU network. Although being so far the most complicated arhitecture for theoretical results, it is still far from what is used in practice like U-Nets and transformers. Regarding the full error analysis, we only focus on the optimization and sampling error, and do not disect the generalization error. When bridging the theoretical results with practical designs of diffusion models, our results are mostly qualitative and we only compare two schedules under well-trained and less-trained cases. Finer analysis techniques are needed for providing a quantitative demonstration of the parameters in diffusion models, as well as for motivating more designs of weightings and schedules. We will leave them for future exploration.

---

> > ### Comment · Reviewer_Ti47 · 2024-08-13
> >
> > Thanks for the detailed discussion on the technique novelty and limitation. Since the rebuttal addresses my concerns, I will raise my score to $5$.

---

### Official Review · Reviewer_5uDw · 2024-07-13

**Soundness:** 4
**Presentation:** 4
**Contribution:** 3
**Rating:** 7
**Confidence:** 3

**Summary:**

This paper provides a full error analysis (considering both training and sampling) for a class of score-based SDE diffusion models, and using the results to understand why and when certain time and variance schedules are preferred.

**Strengths:**

- The theoretical contributions of the paper are excellent
- Overall the paper is well organized and the notations are clear

**Weaknesses:**

- The stated theorems are quite dense and are not easy to parse. The paper would benefit from having informal version of these results highlighting the key qualitative features
- Only an SDE based sampler (the exponential integrator scheme) is considered, while comparisons with other samplers (such as the Probability Flow ODE) are missing

**Questions:**

- In Remark 1, it was stated that the KL divergence can be explicitly computed when the data distribution is Gaussian, would you provide such result in the appendix?

Minor remarks:
- typo in line 16: variance -> various
- missing lemma number in line 201

**Limitations:**

Yes

---

> ### Author Rebuttal · Authors · 2024-08-07
>
> We sincerely thank the reviewer for all the helpful advice and comments and deeply appreciate the positive evaluation.
>
> >Weakness 1: The stated theorems are quite dense and are not easy to parse...
>
> We greatly appreciate the comment and will add more intuitive explanation or informal versions of results in our next version.
>
> >Weakness 2: Only an SDE based sampler (the exponential integrator scheme) is considered, while comparisons with other samplers (such as the Probability Flow ODE) are missing.
>
> Probability Flow ODE ([1]) is indeed another implementation. It reverses the diffusion as an ODE. Because of that, common tools like Girsanov no longer applies, and its analysis requires different approaches. However, there are existing anlyses (e.g., [2,3]) that assumes score is already learned within $\epsilon$ error, and these approaches can be used to replace the sampling component of our full analysis. We also note so far there is no verdict on which one is better, even just for the sampling part.
>
> In this work we simply chose to analyze the VE-SDE implementation as it was used in many celebrated papers [4,5]. But we agree that other sampling strategies could also be interesting future investigations!
>
> >Question 1: In Remark 1, it was stated that the KL divergence can be explicitly computed when the data distribution is Gaussian, would you provide such result in the appendix?
>
> When the target distribution is a Gaussian with mean $m$ and covariance $\sigma^2 I_d$, the score function can be eaxctly computed explicitly. In this case, $p_\delta$ and $q_{T-\delta}$ in Theorem 2 are both Gaussian and $\text{KL}(p_\delta|q_{T-\delta})$ satisfies:
> $$
> \text{KL}(p_\delta|q_{T-\delta})=\frac{d}{2}\log M -\frac{d}{2} +\frac{d}{2M}+ \frac{\lVert m\rVert^2}{(\sigma^2+\bar{\sigma}_T^2)^2 M },
> $$
>
> with $M=\frac{\bar{\sigma}\_T^2+\sum_{k=0}^{N-1} \frac{(\sigma^2+\bar{\sigma}\_\delta^2)^2}{(\sigma^2+\bar{\sigma}\_{T-t_{k+1}}^2)^2}(\bar{\sigma}\_{T-t_k}^2-\bar{\sigma}\_{T-t_{k+1}})}{\sigma^2+\bar{\sigma}\_\delta^2}$. We will add the detailed computation in the appendix of our updated manuscript.
>
> [1]  Y. Song et al. “Score-based generative modeling through stochastic differential equations”. In: International Conference on Learning Representations. 2021
>
> [2] Chen, Sitan, et al. "The probability flow ode is provably fast." Advances in Neural Information Processing Systems 36 (2024).
>
> [3] Li, Gen, et al. "Towards faster non-asymptotic convergence for diffusion-based generative models." arXiv preprint arXiv:2306.09251 (2023).
>
> [4] Karras, Tero, et al. "Elucidating the design space of diffusion-based generative models." Advances in neural information processing systems 35 (2022): 26565-26577.
>
> [5] Song, Yang, et al. "Score-based generative modeling through stochastic differential equations." arXiv preprint arXiv:2011.13456 (2020).

---

> > ### Comment · Reviewer_5uDw · 2024-08-13
> > **Thank you for the rebuttal**
> >
> > I am satisfied with the rebuttal and will keep my score.

---

### Author Rebuttal · Authors · 2024-08-07

We would like to thank all the reviewers for the helpful comments. We have improved our convergence theory, and the new version now uses a much more relaxed assumption, where the original version $$d=\Theta(m)$$ is replaced by
$$d=\Omega({\text{poly}}(\log (nN))).$$

Note in the old version, the input dimension is the same order as the width of the network $m$, while in the new version, it only needs $d$ to be not too small, or equivalently, the number of data points $n$ and the number of time points $N$ should not be exponential in $d$.

All the other assumptions remain unchanged and we obtain the following new version of the theorem:
> **Theorem 1.** Define a set of indices to be $\mathcal{G}^{(s)}=\{(i,j)|f(\theta^{(s)};i,j)\ge f(\theta^{(s)};i',j')\text{ for all }i',j'\}$.
 Then given Assumption 1 and 2, for any $\epsilon_{\rm train}>0$, there exists some $M(\epsilon_{\rm train})=\Omega\left(\text{poly}\big(n,N,d,L,T/t_0,\log(\frac{1}{\epsilon_{\rm train}})\big)\right)$, s.t., when $m\ge M(\epsilon_{\rm train})$,  $h =\Theta(\frac{nN}{m\min_j w(t_j)(t_j-t_{j-1}) \bar\sigma_{t_j} })$, and $k=\mathcal{O}(d^{\frac{1-a_0}{2}}n^2N\log(\frac{d}{\epsilon_{\rm train}}))$, with probability at least $1-\mathcal{O}(nN)\exp(-\Omega(d^{2a_0-1}))$, we have
    $$
      \bar{\mathcal{L}}\_{\rm em}(\theta^{(k)})\le\prod_{s=0}^{k-1}\left(1-C_5 h \ w(t_{j^*(s)})(t_{j^*(s)}-t_{j^*(s)-1})\bar{\sigma}\_{t_{j^*(s)}} \left(\frac{md^{\frac{a_0-1}{2}}}{n^3N^2}\right)\right)\bar{\mathcal{L}}\_{\rm em}(\theta^{(0)})
       $$
    where the universal constant $C_5>0$, $a_0\in(\frac{1}{2},1)$,  and $(i^*(s),j^*(s))=\arg\max_{(i,j)\in\mathcal{G}^{(s)}}w(t_{j})(t_j-t_{j-1})\bar{\sigma}\_{t_{j}}$. Moreover, when $K=\Theta(d^{\frac{1-a_0}{2}}n^2N\log(\frac{d}{\epsilon_{\rm train}}))$,
    $$
        \bar{\mathcal{L}}\_{\rm em}(\theta^{(K)})\le \epsilon_{\rm{train}}.
   $$

This bound also has improved dependency on $d$, namely $d^{\frac{a_0-1}{2}}\in(d^{-1/4},1)$ as opposed to the old version, which is $d^{2a-1.5-4c}\sqrt{\log d}\in (d^{-1}\sqrt{\log d}, d^{-9/10}\sqrt{\log d})$.

---

### Comment · Area_Chair_uWEN · 2024-08-12
**Discussion**

Please participate to the discussion. Thanks

---

### Decision · Program_Chairs · 2024-09-25

**Decision:**

Accept (poster)

**Comment:**

This paper presents an analysis of diffusion models, taking into account both the training and sampling processes. The authors derive non-asymptotic convergence bounds for denoising score matching under gradient descent and obtain a refined sampling error analysis for variance exploding models. This unified analysis provides insights into the design choices of diffusion models, including the selection of noise distribution, loss weighting functions, and time and variance schedules.

The paper has perhaps potential to guide practical applications. However, the results presented therein are complex and it will be difficult for readers to understand their practical implications. While the paper aims to connect theoretical results with practical design choices, it somewhat lacks in concrete examples and guidelines. This is reinforced by the lack of empirical results. Additionally the authors rely on strong assumptions which may not capture the practice of diffusion models.

So while I recommend the paper be accepted, the authors are strongly encouraged to enhance the clarity and accessibility of the presentation. This could include providing more intuitive explanations of the theoretical results and their practical implications, offering informal versions of the theorems, summarizing key features etc. A more concrete discussion of practical implications should be included. and the authors should demonstrate how their theoretical findings can help guiding the selection of noise distributions, weighting functions, and time and variance schedules, providing specific examples and applications. Presenting detailed experiments to validate these guidelines would improve significantly the paper's impact.